# PHF6-mediated transcriptional control of NSC via Ephrin receptors is impaired in the intellectual disability syndrome BFLS

Dilan Rasool[1,2,3,4], Audrey Burban[1,2,4,5], Ahmad Sharanek[1,2,4,5], Ariel Madrigal[6,7], Jinghua Hu[8], Keqin Yan[8], Dianbo Qu[1,2], Anne K Voss [9,10], Ruth S Slack[1,2], Tim Thomas [9,10], Azad Bonni[11,12], David J Picketts[1,8,13], Vahab D Soleimani [3,4,6,13], Hamed S Najafabadi [6,7✉] & Arezu Jahani-Asl [1,2,3,4,5,8,14✉]

## Abstract

**The plant homeodomain zinc-finger protein, PHF6, is a transcriptional regulator, and PHF6 germline mutations cause the X-linked intellectual disability (XLID) Börjeson-Forssman-Lehmann syndrome (BFLS). The mechanisms by which PHF6 regulates transcription and how its mutations cause BFLS remain poorly characterized. Here, we show genome-wide binding of PHF6 in the developing cortex in the vicinity of genes involved in central nervous system development and neurogenesis. Characterization of BFLS mice harbouring PHF6 patient mutations reveals an increase in embryonic neural stem cell (eNSC) self-renewal and a reduction of neural progenitors. We identify a panel of Ephrin receptors (EphRs) as direct transcriptional targets of PHF6. Mechanistically, we show that PHF6 regulation of EphR is impaired in BFLS mice and in conditional *Phf6* knock-out mice. Knockdown of *EphR-A* phenocopies the PHF6 loss-of-function defects in altering eNSCs, and its forced expression rescues defects of BFLS mice-derived eNSCs. Our data indicate that PHF6 directly promotes Ephrin receptor expression to control eNSC behaviour in the developing brain, and that this pathway is impaired in BFLS.**

**Keywords** Neural Stem Cells; Intellectual Disability; BFLS; PHF6; Ephrin Receptors
**Subject Categories** Chromatin, Transcription & Genomics; Molecular Biology of Disease; Neuroscience

## Introduction

The plant homeodomain zinc finger protein 6, PHF6, is a transcriptional regulator (Liu et al, 2014) that is highly conserved in vertebrates with high expression during the early stages of corticogenesis (Cheng et al, 2018; Voss et al, 2007). PHF6 is found in a complex with different components of the Polymerase associated factor 1 (PAF1) complex to promote neuronal migration in the developing cerebral cortex (Jahani-Asl et al, 2016; Zhang et al, 2013) suggesting a role for PHF6 in transcriptional elongation. The PAF1 complex, has also been shown to regulate promoter proximal pausing of RNA polymerase II (Chen et al, 2015). Whether and how PHF6 may be involved in transcriptional elongation and polymerase pausing has remained to be investigated.

Germline mutations in *Phf6* causes the X-linked intellectual disability (XLID), Börjeson-Forssman-Lehmann syndrome (BFLS), characterized by impairments in cognitive function, epileptic-like seizures, and behavioural disturbances (Lower et al, 2002), in addition to endocrine defects (McRae et al, 2020). Multiple mutations on the *Phf6* gene within the X chromosome have been identified in BFLS patients (Berland et al, 2010; Carter et al, 2009; Lower et al, 2002; Turner et al, 2004). Although prior research has established that loss of PHF6 function impairs the migration of newly born neurons, the involvement of PHF6 in the regulation of different aspects of neural development remains unexplored.

Neurogenesis is outlined as a process in which new neurons are generated from neural stem cells (NSCs). This process is comprised of proliferation and fate specification of NSCs, migration of newborn neurons, and maturation of these neurons (Urbán and Guillemot, 2014). A number of XLID genes appear to impair

[1]Department of Cellular and Molecular Medicine, University of Ottawa, 451 Smyth Road, Ottawa, ON K1H 8M5, Canada. [2]University of Ottawa, Brain and Mind Research Institute, 451 Smyth Road, Ottawa, ON K1H 8M5, Canada. [3]Department of Medicine, Division of Experimental Medicine, McGill University, 1001 Decarie Boulevard, Montréal, QC H4A 3J1, Canada. [4]Lady Davis Institute for Medical Research, Jewish General Hospital, 3755 Chemin de la Côte-Sainte-Catherine, Montréal, QC, H3T 1E2, Canada. [5]Gerald Bronfman Department of Oncology, McGill University, 5100 de Maisonneuve Blvd. West, Montréal, QC, H4A 3T2, Canada. [6]Department of Human Genetics, McGill University, 3640 Rue University, Montréal, QC H3A 0C7, Canada. [7]McGill Genome Centre, Dahdaleh Institute of Genomic Medicine, 740 Dr Penfield Avenue, Montréal, QC H3A 0G1, Canada. [8]Regenerative Medicine Program and Cancer Therapeutics Program, Ottawa Hospital Research Institute, Ottawa, ON K1H 8L6, Canada. [9]Walter and Eliza Hall Institute of Medical Research, Melbourne, VIC 3052, Australia. [10]Department of Medical Biology, The University of Melbourne, Melbourne, VIC 3052, Australia. [11]Roche Pharma Research and Early Development (pRED), Roche Innovation Center, F. Hoffmann-La Roche Ltd., Basel, Switzerland. [12]Department of Neuroscience, Washington University School of Medicine, St. Louis, MO 63110, USA. [13]Departments of Biochemistry, Microbiology and Immunology, University of Ottawa, Ottawa, ON K1H8M5, Canada. [14]Ottawa Institutes of System Biology, University of Ottawa, Health Sciences Campus, 451 Smyth Road, Ottawa, ON K1H 8M5, Canada. ✉E-mail: hamed.najafabadi@mcgill.ca; arezu.jahani@uottawa.ca

neurogenesis via altering NSC fate (Bustos et al, 2018; Kim et al, 2016; Luo et al, 2010; May et al, 2015; Selvan et al, 2018; Telias et al, 2015), raising the question of whether *Phf6* mutations impact the NSC pool in the developing brain.

Ephrin receptors (EphR), the largest family of receptor tyrosine kinases (RTK) (Kullander and Klein, 2002), are highly expressed in the developing brain and play crucial roles in the regulation of proliferation, apoptosis, cell adhesion, cell fate specification, and neurogenesis (Gerstmann and Zimmer, 2018; Kullander and Klein, 2002; Park, 2013). EphRs are classified as either A- or B-type of receptors according to sequence homology, and require binding to membrane-bound ephrin ligands for signal transduction (Committee, 1997). EphA members have been studied in the contexts of axon guidance, neural stem cell proliferation during development, embryogenesis, and neuroblast migration to the olfactory bulbs via forward signalling mechanisms (North et al, 2009; Park, 2013; Todd et al, 2017). EphB members have also been reported to alter hippocampal progenitor cells and cell proliferation (Calò et al, 2005; Genander and Frisén, 2010; He et al, 2005).

In the present study, we characterize global PHF6 regulation of the genome in the developing cortex and show a position-dependent role for PHF6 in the regulation of transcription as an activator or repressor. We employ several genetic mouse models including BFLS patient mouse models and *Phf6* knock-out (KO) models to establish a role for PHF6 in altering eNSCs. Importantly, we report several members of EphRs as direct transcriptional targets of PHF6, with the EphA family members involved in the regulation of neurogenic processes. Our data suggests that these receptors could represent a therapeutically exploitable target for BFLS and other XLID disorders with impaired neurogenesis.

## Results

### Genome-wide analysis of PHF6 targets in the developing brain

To begin to examine the function of PHF6 as a transcriptional regulator in the embryonic brain, we performed ChIP-seq analysis of PHF6 in the developing cortex of mouse embryos at embryonic day 17 to 18 (E17-18). We identified 2467 PHF6 binding sites at $P$-value $< 10^{-5}$ (Dataset EV1, Figs. 1 and EV1). These binding sites occurred in various genomic regions, including the proximal region of transcription start sites (TSS'), gene bodies, and intergenic regions (Fig. 1A). Compared to what would be expected from the random distribution of binding sites across the genome, we observed significant enrichment upstream of the TSS as well as in the 5' untranslated region (UTR) of protein-coding genes (Fig. 1A). Particularly, PHF6 sites are strongly enriched in the 1 kb region around the TSS, with the highest density immediately downstream of the TSS (Fig. 1B). This pattern suggests a role of PHF6 in regulating gene expression.

Follow-up analysis revealed that PHF6-bound regions significantly overlap $(CA)_n$-microsatellite repeats, as revealed by motif analysis of the top 1000 PHF6 sites (Figs. 1C and EV1B). These microsatellites are specifically located at the centre of PHF6 sites (Figs. 1D and EV1B), suggesting that they are associated with PHF6 binding. Among the top 1000 PHF6 peaks, 609 overlap a $(CA)_n$

repeat on either DNA strand. In comparison, we observed an overlap of only 67 between $(CA)_n$ repeats and shuffled peak coordinates (Fisher's exact test $P < 2.2e-16$) (Dataset EV2). An unbiased analysis of the distribution of all genomic $(CA)_n$ repeats revealed that they are largely enriched near genes involved in developmental processes, including central nervous system development, neurogenesis, and neuron differentiation (Fig. 1E, Dataset EV3). Function enrichment analysis of PHF6 sites also revealed the same trend (Dataset EV4), with many Gene Ontology (GO) terms such as forebrain development and regulation of neurogenesis commonly found among the most enriched terms for both PHF6 sites and $(CA)_n$ microsatellite repeats (Fig. 1F, Dataset EV5). These results suggest that $(CA)_n$ repeats are specifically enriched near neural development genes and are bound by PHF6.

Next, we profiled the genome-wide pattern of gene deregulation by analysis of *Phf6* knockdown (KD) and control cortical progenitors following their isolation at embryonic day 14 (E14) and expansion for 5 days in culture. RNA-seq analysis (Dataset EV6) revealed that PHF6 functions as a transcriptional activator or repressor (Fig. 2A). In addition, enrichment analysis, performed separately on upregulated and downregulated genes, revealed that a large panel of genes involved in nervous system development are downregulated in the *Phf6* KD group (FDR < 0.02) (Fig. 2B). A number of significant PHF6-differentially expressed genes were found to have peaks within the $+/-2$ kb vicinity of the TSS (Fig. 2C,D).

To further understand the role of PHF6 in the regulation of transcription, we employed Pol II occupancy data (Liu et al, 2017), and examined the association between PHF6 binding and Pol II occupancy in neural progenitor cells. Interestingly, we observed that TSS' with a PHF6 site within 300 bp tend to be depleted of Pol II, compared to genes with a PHF6 site between 300–1000 bp of the TSS (Fig. 2E). This pattern suggests that the binding of PHF6 within the immediate vicinity of TSS might have a negative effect on the recruitment of Pol II to the TSS. To examine this prediction, we analyzed the association of PHF6 peaks and PHF6 differentially regulated genes. We found that PHF6 inhibition led to an overall increase in the expression of 65% of genes with a PHF6 site at or immediately downstream of the TSS (Fig. 2D,F). These observations suggest a position-dependent role for PHF6 in regulating transcription, which may provide mechanistic insight into the dual role of PHF6 as a transcriptional activator and repressor.

### *Phf6* knockdown in primary eNSC cultures alters eNSC expansion

Our data on functional annotation of PHF6 binding sites suggest that PHF6 regulates neurogenesis (Fig. 1F). Interestingly, previous studies in hematopoietic stem cells (HSCs) showed that PHF6 can restrict the self-renewal capacity of HSCs (McRae et al, 2019; Miyagi et al, 2019). These findings led us to investigate whether PHF6 regulates cell proliferation or self-renewal. To begin with, we subjected PHF6-GFP or control GFP-expressing neuroblastoma (N2A) cell lines to KI67 staining and found that PHF6 significantly suppressed the proliferation of these cells (Fig EV2A–C). Next, we induced the KD of *Phf6* via a pool of siRNA in primary E14 eNSC cultures followed by limiting dilution assay (LDA). Compared with eNSCs transfected with non-targeting siRNA control, we found a significant increase in eNSC neurosphere numbers upon KD of

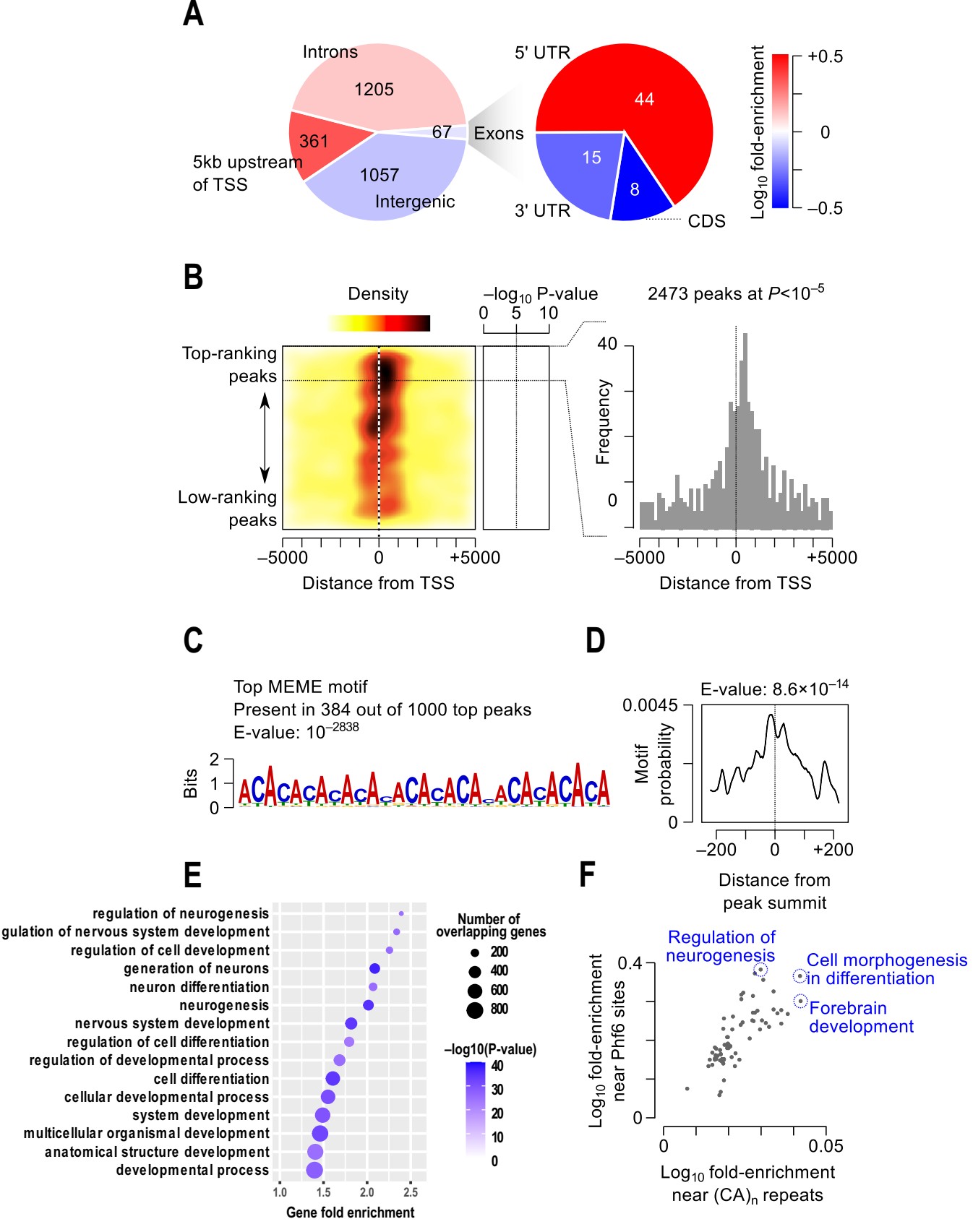

**Figure 1.   Genomic distribution of PHF6 binding sites in the developing cortex.**

(A) The numbers of PHF6 sites that overlap different genomic regions are shown in the pie chart. The right pie chart shows a breakdown of sites that overlap exonic regions. The colour gradient, shown on the right, represents the logarithm of enrichment of PHF6 sites in each region relative to random expectation. Only PHF6 sites with $P < 10^{-5}$ are included in the charts. (B) The heatmap on the left shows the distribution of PHF6 sites relative to TSS'. The peaks are sorted by ascending order of their $P$-values (shown in the middle) from the top to the bottom. The colour gradient depicts the frequency of PHF6 sites relative to the position of the nearest TSS, also shown for top-ranking PHF6 sites using the histogram on the right. (C–F) PHF6 binds to $(CA)_n$-microsatellite repeats. (C) The sequence logo depicts the top motif identified by MEME-ChIP [PMID: 21486936]. (D) The distribution of the $(CA)_n$ motif relative to the peak summits is shown, as revealed by CentriMo [PMID: 22610855]. (E) Dot plot representation of the GO terms that are enriched near PHF6 sites. Only the top 15 terms with the most significant p-values are shown. The x-axis shows the fold-enrichment of the term, while the dot size and colour represent the number of PHF6 targets that overlap the GO term and the hypergeometric p-value, respectively. (F) Each dot in the scatterplot represents a GO term that is significantly enriched in both the GREAT analysis of $(CA)_n$ simple repeats and the GREAT analysis of PHF6 sites. The x- and y-axes reflect the logarithm of the hypergeometric fold-enrichment of the terms. The GO terms with the largest enrichment are highlighted. $n = 6$ mouse cortices were pooled for each PHF6 ChIP and IgG control ChIP, where $n$ represents an independent biological sample.

*Phf6* (Fig. 3A,B). Consistent with this observation, immunoblotting analyses of neurospheres following 7 days in culture, revealed upregulation of stem cell markers, SOX2 and NESTIN in *Phf6* KD relative to the control cells (Fig. 3C). Our data suggest that PHF6 restricts stem cell self-renewal in primary neurosphere cultures.

### *Phf6* conditional knock-out mice exhibit alterations in eNSC processes

We next set out to characterize the role of PHF6 in stemness using a genetic mouse model in which we induced genetic deletion of *Phf6* via breeding *Phf6*[loxP/loxP] with *Nestin-CreERT2*[+] mice followed by tamoxifen administration at E14 for 24–48 h to delete *Phf6* exons 4 and 5 in the Nestin expressing cells (Fig. 3D). Extreme limiting dilution assay (ELDA) (Hu and Smyth, 2009; Rasool et al, 2022) and LDA analyses of eNSCs obtained from *Phf6*[-/Y] / *Nestin-CreERT2*[+] *(Phf6* KO) and *Phf6*[loxP/Y] / *Nestin-CreERT2*[-] (control) mice revealed a significant increase in self-renewal (Fig. 3E), sphere number (Fig. 3F), and sphere diameter (Fig. 3G,H) in *Phf6* KO eNSCs. Importantly, a significant increase in the expression of the stemness markers, *Nestin* and *Sox2*, (Fig. 3I) and an increase in EdU incorporation (Fig. 3J) was observed in the eNSCs of *Phf6*[-/Y] / *Nestin-CreERT2*[+] mice. Our data shows that the genetic deletion of *Phf6* promotes the self-renewal of eNSCs, suggesting that PHF6 loss-of-function may restrict eNSC commitment to differentiated progenies in the developing brain.

In parallel, we employed a second *Phf6* KO mouse model wherein *Phf6*[loxP/loxP] were bred with *Nestin-Cre*[+] mice to induce deletion of *Phf6* from the mouse central and peripheral nervous system at E11.5 (Tronche et al, 1999), the onset of *Nestin* gene expression, thus producing a highly efficient KO model (Fig. EV3A,B). We subjected the brain sections from *Phf6*[-/Y] / *Nestin-Cre*[+] (KO) and *Phf6*[loxP/Y] / *Nestin-Cre*[-] (Ctl) mice at post-natal day 0 (P0) to Nissl staining and found a notable decline in neuron density within the forebrain and midbrain sections of *Phf6*[-/Y] / *Nestin-Cre*[+] brains compared to *Phf6*[loxP/Y] / *Nestin-Cre*[-] controls (Fig. EV3C). Our data suggest that the deletion of *Phf6* induces a decline in neuron density. Taken together, these results support a model whereby PHF6 may restrict eNSC self-renewal and promotes eNSC commitment to newly born neurons.

### BFLS patient mouse models exhibit alterations in stemness markers and eNSC self-renewal

The R324X mutation is the most recurrent BFLS patient-mutation occurring at exon 10 (C.1024 C > T), impairing the ePHD2 domain,

whereby PHF6 is proposed to function as a truncated protein (Ahmed et al, 2021; Chao et al, 2010; Crawford et al, 2006; Gecz et al, 2006; Jahani-Asl et al, 2016; Lower et al, 2004, 2002; Todd et al, 2015). Another BFLS patient point mutation (m) in *Phf6* is wherein cysteine-99 is replaced with phenylalanine (C99F) at nt.296 G > T impairing the function of the PHD1 domain. To investigate whether impairment in eNSC fate specification may underlie BFLS pathogenesis, we employed both BFLS mouse models, R342X and C99F-m. Analysis of mRNA expression in E14 cerebral cortices revealed a consistent increase in the expression of both *Nestin* and *Sox2* in BFLS relative to wild-type control mice (Fig. 3K,L). We thus conducted additional analysis in eNSCs of R342X mice and found a significant increase in their self-renewal, neurosphere number, and proliferation relative to eNSCs of the wild-type control mice (Fig. 3M–P). Taken together, our findings demonstrate that similar to *Phf6*[-/Y] / *Nestin-CreERT2*[+] mice, BFLS patient mouse models exhibit alterations in eNSC expansion.

### PHF6 target analysis: Identification of Ephrin Receptors

To identify downstream effectors of PHF6 function in the regulation of neurogenesis, we first analyzed the candidate target genes with their expression significantly deregulated based on the RNA-seq analysis with particular focus on druggable targets (e.g., Receptors, Kinases). These analyses revealed a host of candidate genes that could serve as PHF6 targets to regulate neurogenesis (Dataset EV6). We focused on members of the EphR family (*EphA4/7* and *EphB1/2*) given that EphRs are the largest family of RTKs highly expressed in the developing brain (Barquilla and Pasquale, 2015; Darling and Lamb, 2019; Lisabeth et al, 2013). Importantly, EphRs have been shown to play different roles in regulating neuronal development (Aoki et al, 2004; del Valle et al, 2011; Stuckmann et al, 2001; Wilkinson, 2014). Prior to validation of EphRs as viable targets of PHF6 in the context of BFLS, we conducted additional gene expression analysis using public databases. First, via querying single-cell RNA-seq data [Data ref: (Di Bella et al, 2021)] of the developing mouse brain, we found that *Phf6*, *EphA4/7*, and *EphB1/2* are expressed in the developing brain of mice ranging from embryonic day 10 (E10) to postnatal mice at day 4 (P4) (Figs. 4A–F and EV4A–E). Furthermore, Pearson correlation analysis between *Phf6*, *EphA4/7*, and *EphB1/2* revealed a positive correlation between *Phf6* and *EphR* expression in different cell types, in particular progenitors and migrating neurons (Figs. 4G,H and EV4A–E). Second, we analyzed the RNA-seq data of the human ventral frontal cortex (VFC) [Data ref: (BrainSpan

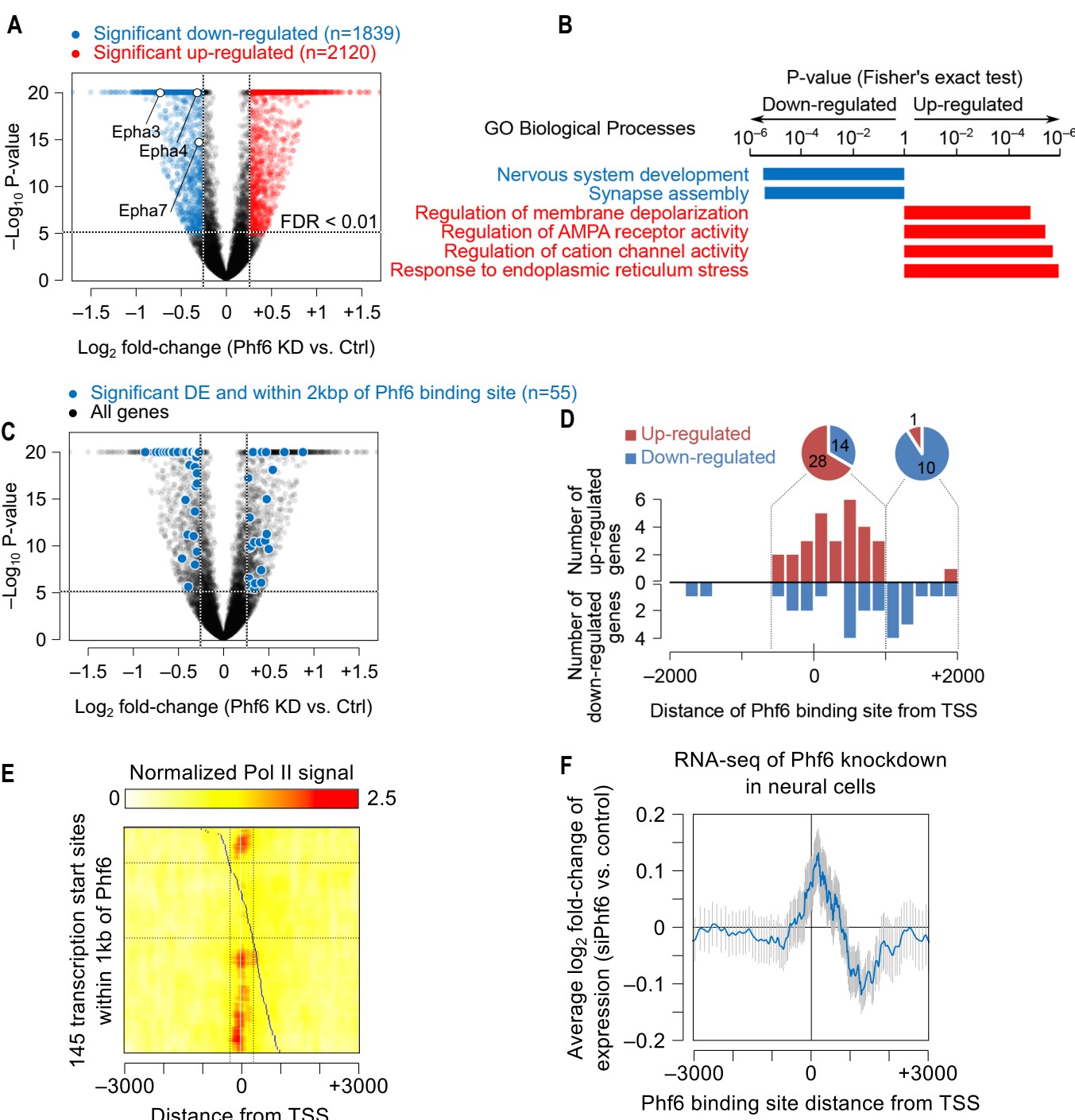

**Figure 2. Position-dependent effect of PHF6 on transcription.**

(A,B) *Phf6* KD and control cortical progenitors were subjected to mRNA-seq analysis ($n = 3$). Plots represent differentially regulated candidate target genes (A), and functional annotation of downregulated versus upregulated genes (B). GO term enrichment analysis was performed using CPDB (Kamburov et al, 2011). (C,D) PHF6 peak-gene associations within $+/-$ 2Kb of TSS and the effect of *Phf6* KD ($n = 3$) on expression is presented. (E) PolII signal near the TSS of the PHF6-bound genes is shown using the colour gradient in the heatmap. The rows represent the genes, sorted based on the position of the PHF6 site. The PHF6 binding sites are depicted in blue. The vertical dotted lines delineate the $+/-300$ bp region around the TSS'. The horizontal dotted lines delineate the genes with a PHF6 site within this $+/-300$ bp region. (F) The expression changes in *Phf6* KD cells as a function of the binding position of PHF6. Each data point shows the average for 50 genes that have PHF6 binding, with the binding site location relative to the TSS shown on the *x*-axis. Data information: Error bars represent ± SEM. mRNA-seq raw reads were mapped to mm10 genome using HISAT2 (Kim et al, 2015), followed by duplicate read removal using samtools. Gene-level read counts were obtained by HTSeq (Anders et al, 2015), using gene annotations from GENCODE (release M9). Genes with a minimum of 150 reads in at least one sample were retained. Gene set analysis was performed using ConsensusPathDB (Kamburov et al, 2011). *n* represents an independent biological sample.

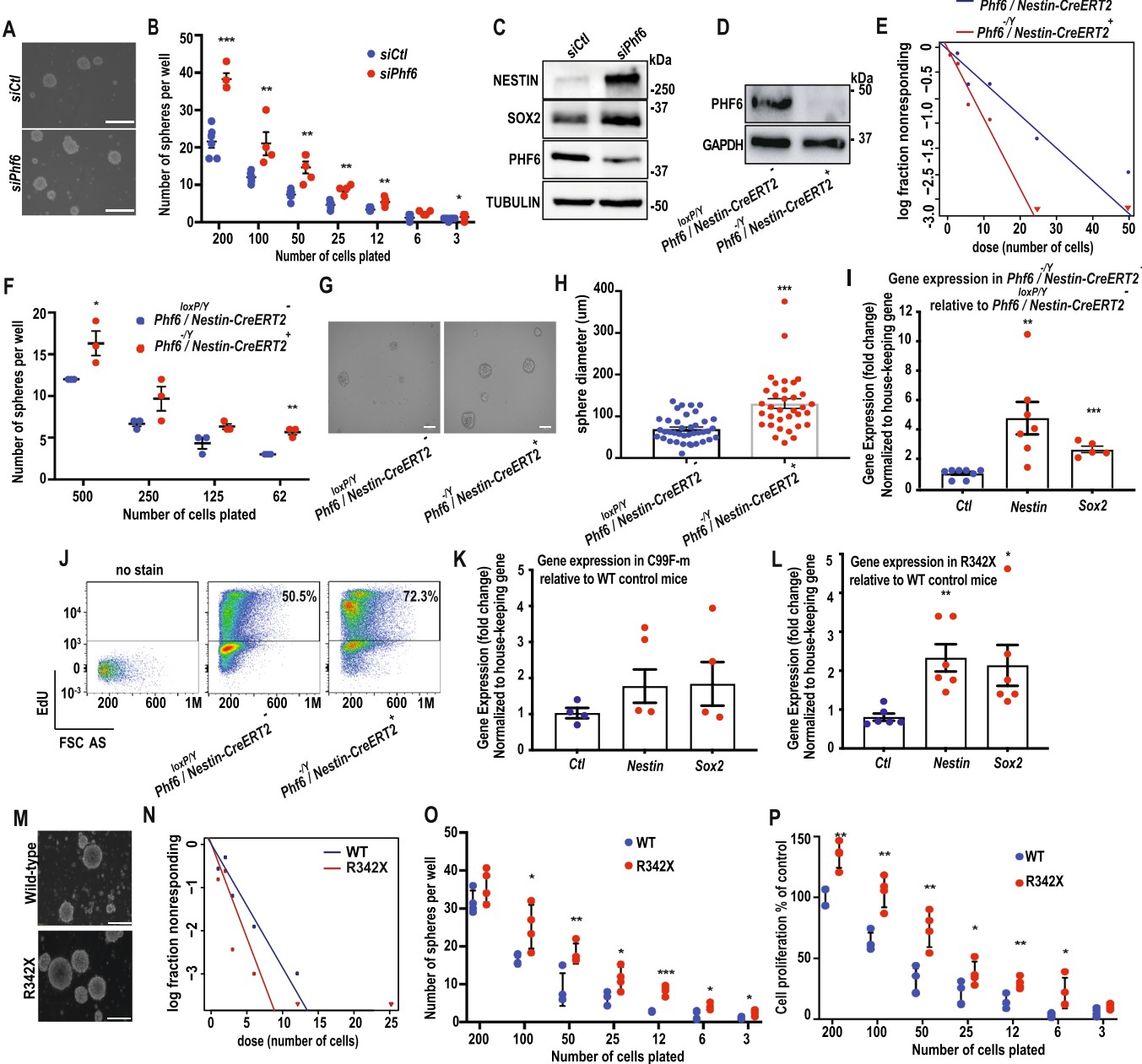

**Figure 3. PHF6 suppresses self-renewal of eNSCs.**

(A–C) eNSC were isolated and cultured from WT mice at E14 and *Phf6* KD was induced using an siRNA approach. Samples were analyzed using a limiting dilution assay (LDA) (A,B) and immunoblotting (C) using antibodies indicated on the blot. (D–J) eNSCs were cultured from *Phf6⁻/Y / Nestin-CreERT2⁺* and control *Phf6loxP/Y / Nestin-CreERT2⁻* mouse brains at ~E15 and were subjected to immunoblotting analysis (D), ELDA (E) (*p* = 0.00686), LDA (F), sphere diameter (G,H) (*p* < 0.0001), RT-qPCR analysis using *Nestin* and *Sox2* primers (I), and 5-ethynyl-2'-deoxyuridine (EdU) analysis (J). (K,L) eNSCs were cultured from C99F (K), R342X (L) and corresponding wild-type control mice. mRNA expression of *Nestin* and *Sox2* were analyzed by RT-qPCR. (M–P) eNSC were cultured from R342X mice and wild-type control mice and were subjected to ELDA (M,N) (*p* = 0.0211), LDA (O), and alamarBlue analysis (P) 7 days post-plating. Scale bar represents 100 µm. Data information: Data are presented as mean ± SEM. **p* < 0.05, ***p* < 0.01, ****p* < 0.001 (two-tailed unpaired student *t*-test). Representative plots of *n* > 3 independent replicates are shown in (A,C–E,G,J,M,N), data in panels (B,F,H,I,K,L,O,P) are plotted with *n* > 3 mean +/− SEM. *n* represents an independent biological sample. Source data are available online for this figure.

Atlas of the Developing Human Brain, 2011)] and found a similar trend in the expression of *EPHR* genes across development, and their correlation with *PHF6* expression (Fig EV4F–J). We, thus, asked if EphR expression levels are altered in *Phf6* KO and BFLS mice. Via subjecting eNSCs from *Phf6loxP/Y / Nestin-CreERT2⁻* and

*Phf6⁻/Y / Nestin-CreERT2⁺* mice to RT-qPCR and immunoblotting analyses, we observed a significant decrease in both mRNA and protein expression of each of the identified EphR upon genetic deletion of *Phf6* (Fig 5A,B). Independently, we also subjected E14 brain tissue from R342X and C99F-m mice to RT-qPCR and

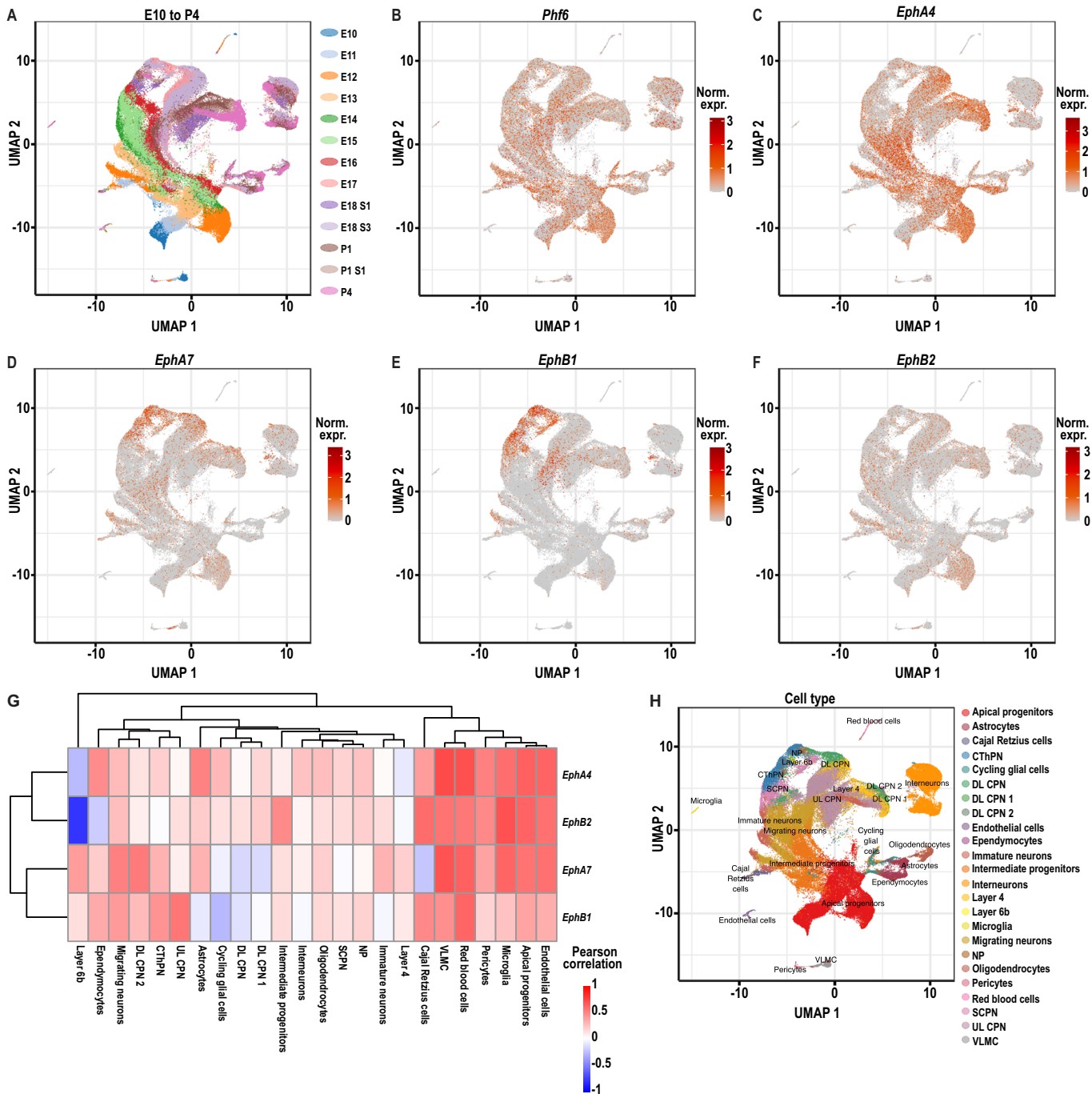

**Figure 4. Cell type specific co-expression analysis of *EphR* and *Phf6* in mouse cerebral cortex.**

(A–F) Low-dimensional representation of single cells from mouse cerebral cortex, based on UMAP embedding of single-cell RNA-seq data [Data ref: (Di Bella et al, 2021)] are shown. Cells are coloured based on animal age (A), or the expression of *Phf6* (B), *EphA4* (C), *EphA7* (D), *EphB1* (E), or *EphB2* (F). (G) Heatmap representation of the Pearson correlation coefficients between *Phf6* and *EphR* across various cell types are shown. Correlation values were calculated using imputed gene expression profiles after applying MAGIC (Van Dijk et al, 2018). (H) UMAP embedding of cells are coloured by cell type. UMAP coordinates and cell type annotations are from [Data ref: (Di Bella et al, 2021) (GEO GSE153164)].

immunoblotting analyses. The results revealed downregulation of *EphR* mRNA and protein expression in both C99F and R342X mice with a more profound impact in R342X mice (Figs. 5C,D and EV5A,B), confirming that the expression of *EphRs* is altered in BFLS mice harbouring PHF6 patient mutations.

## PHF6 directly occupies the gene regulatory regions of EphR to alter their expression

We next set out to investigate if the identified *EphRs* are direct PHF6 targets. Our ChIP-seq data revealed robust and significant

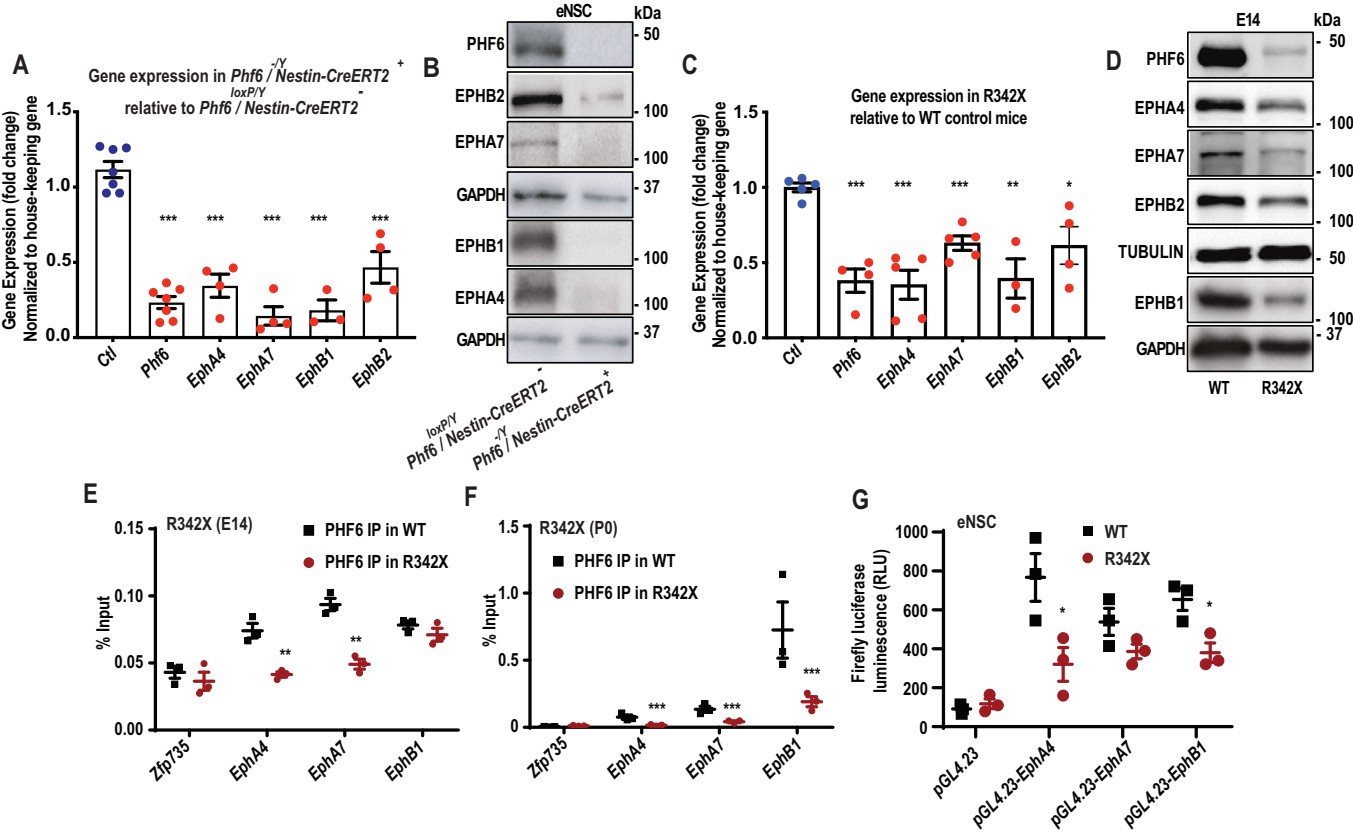

**Figure 5. EphR are direct PHF6 targets.**

(A,B) eNSCs were cultured from *Phf6^-/Y^ / Nestin-CreERT2^+^* and control *Phf6^loxP/Y^ / Nestin-CreERT2^-^* at ~E15 and mRNA and protein expression of EphR were analyzed by RT-qPCR (A) and immunoblotting (B). (C,D) mRNA and protein of brain tissue obtained from E14 R342X and wild-type control mice were analyzed as described in (A,B). (E,F) Cerebral cortical tissues were isolated from WT and R342X mice at E14 (E) or at P0 (F). Samples were subjected to ChIP-qPCR using a PHF6 antibody. *Zfp735* loci was used as negative control for the PCR. (G) Dual luciferase reporter assay was performed in WT or R342X eNSC cultures 48 h following electroporation with pGL4.23-*EphA4*, pGL4.23-*EphA7*, pGL4.23-*EphB1* or pGL4.23-basic reporter plasmids. RLU Relative luminescence units. Data information: Data are presented as mean ± SEM. *$p < 0.05$, **$p < 0.01$, ***$p < 0.001$. Two-tailed unpaired student *t*-test (A,C,G), one-way ANOVA (E,F). Representative data of $n > 3$ independent replicates are shown in panels (B,D). Data in panels (A,C,E–G) are plotted with $n > 3$ mean ± SEM. *n* represents an independent biological sample. Source data are available online for this figure.

binding of PHF6 to the promoter of *EphA4* with a *p*-value of 1.8E−08 (Dataset EV1). ChIP-seq data also revealed peaks associated with the TSS of *EphA7* and *EphB1* although the *p*-values did not reach the cut off values for significance (*p*-value for *EphA7*: 7.2E−04; *p*-value for *EphB1*, 1.4E−04) (Dataset EV1, Fig. EV1). We designed ChIP-qPCR experiments to specifically investigate the possibility of PHF6 binding to *EphA4* but also examined *EphA7/EphB1* (Fig. EV1) due to their significant deregulated expression in a PHF6-dependent manner (Dataset EV6). To begin with, ChIP-qPCR experiments were conducted in PHF6-overexpressing N2A cell lines using a ChIP-grade PHF6 antibody. We established PHF6 enrichment on the *EphR* genes that we examined in N2A cells expressing PHF6-GFP relative to GFP control (Fig. EV5C). We further assessed the functional consequences of PHF6 binding to *EphR* via loss- and gain-of-function studies. We conducted a firefly luciferase assay in N2A cell lines expressing PHF6-GFP or GFP control (Fig. EV5D). The cells were electroporated with either the control pGL4.23-basic reporter plasmid (pGL4.23), or the luciferase reporter plasmids harbouring the promoters of different *EphR* genes, including pGL4.23-*EphA4*, pGL4.23-*EphA7*, and pGL4.23-*EphB1*, together with a Renilla expression plasmid, and were

subjected to a dual luciferase assay after 48 h. Cells expressing PHF6-GFP showed increased reporter activity for EphR regulatory regions (Fig. EV5D). Second, we induced the KD of *Phf6* via a pool of siRNA (Fig. EV5E) and subjected the cells to a firefly luciferase assay. Our data revealed significant downregulation of *EphR* promoter activity in *Phf6* KD cells. Importantly, parallel immuno-blotting and RT-qPCR analyses revealed significant deregulation of the EphR protein and mRNA expression levels in a PHF6-dependent manner (Fig. EV5F,G).

To further assess if PHF6 direct regulation of *EphR* might be perturbed in the patient mouse models, we conducted ChIP assays in either E14 or P0 whole brain tissue of R342X, as well as luciferase assay in primary eNSCs and found that PHF6 regulation of *EphR* is consistently impaired in R342X mice (Fig. 5E–G). Similarly, the ChIP assay revealed that the binding of PHF6 to *EphA4* and *EphB1* promoters were significantly attenuated in whole brain tissue of E14 C99F mice relative to the wild-type control (Fig. EV5H). The specificity of the PHF6 antibody used for ChIP was also confirmed in IP and ChIP-PCR experiments using *Phf6^loxP/Y^ / Nestin-CreERT2^-^* and *Phf6^-/Y^ / Nestin-CreERT2^+^* eNSCs (Fig. EV5I,J).

## Knockdown of *EphA* phenocopies PHF6 loss-of-function

We have established that PHF6 directly binds to gene regulatory elements of *EphR* to upregulate their expression. Mice harbouring *Phf6* deletion, or BFLS patient mutations exhibit altered NSC self-renewal and deregulated *EphR* expression, raising the question of whether knockdown of either *EphA4/7* or *EphB1* can phenocopy the PHF6 mutant induced eNSC phenotype in BFLS.

EphA4 and EphA7 are involved in NSC regulation and neural development. EphA4 has been studied in axon guidance and neural circuit formation, whereas EphA7 plays a key role in apoptosis and cortical patterning (Depaepe et al, 2005; Kania and Klein, 2016; Klein, 2012).

We employed an siRNA approach in primary E14 WT eNSCs to induce the KD of each of these receptors followed by ELDA analysis to assess eNSC self-renewal (Fig. 6A–D). Our results revealed a significant increase in eNSC self-renewal in both *EphA4* and *EphA7* KD cells with the most profound impact in the *EphA4* KD cells (Fig. 6A,B). Although there was a similar trend with *EphB* KD in eNSCs, no significant changes in self-renewal were induced upon the knockdown of *EphB* family of receptors (Fig. 6C,D). To investigate the impact of *EphR* KD on stemness, we also subjected whole protein lysates to immunoblotting using SOX2 and NESTIN antibodies (Figs. 6E,F and EV5K,L). We found that KD of *EphA* members and *EphB1*, but not *EphB2*, induced an increase in the protein expression levels of SOX2 and NESTIN (Figs. 6E,F and EV5K,L). Our studies demonstrate that although PHF6 regulates the gene expression of several *EphR* family members, KD of *EphA4* induces the most significant phenotype on eNSC self-renewal.

## EphA- family of receptors rescues the R342X-induced eNSC alterations

In view of our observations that *EphA4* KD most closely phenocopies the *Phf6* mutant-induced eNSC phenotype, we next assessed if forced expression of EphA4 alters eNSC expansion. We generated an EphA4 plasmid fused with a GFP tag on the C-terminus. E14 WT eNSCs were cultured and electroporated with *EphA4*-GFP (pLVX.*EphA4*-GFP), or control GFP plasmid (pLVX.GFP) followed by ELDA and immunoblotting analysis (Fig. 6G–J). Our results showed that the expression of *EphA4* induced a significant decline in eNSC self-renewal (Fig. 6G), the protein expression of both SOX2 and NESTIN (Fig. 6I), stem cell frequency (SCF) (Fig. 6H), and eNSC sphere size (Fig. 6J). We thus aimed to examine if the EphA- family of receptors can rescue the PHF6-mutant induced phenotype using the R342X mouse model (Fig. 6K–P). Forced expression of *EphA4*-GFP and *EphA7*-GFP was induced in eNSC cultures from the R342X mouse brain, and efficient electroporation of EphA4 and EphA7-GFP plasmids were confirmed by immunoblotting (Fig. 6K,L). LDA and ELDA analysis revealed that both *EphA4* (Fig. 6M,N) and *EphA7* (Fig. 6O,P) rescue the R342X-induced eNSC phenotype. In particular, EphA4 more profoundly decreased eNSC self-renewal and SCF in R342X eNSC (Fig. 6M,N). These findings assert the potential for the EphA- family of receptors, specifically EphA4, in ameliorating the PHF6-mutant induced eNSC phenotype.

## BFLS and PHF6-mutant mouse brains display imbalances in stem cell population

We have established that PHF6 patient mutations alter eNSC fate in BFLS, prompting us to characterize the eNSCs, in their niche, in the developing brain. We analyzed the whole-brain lysates of C99F-m (Fig. 7A) and R342X (Fig. 7B) at E14 by immunoblotting analysis. Similar to results from primary eNSCs, we found a marked increase in the expression of stem cell markers, in BFLS mouse brains (Figs. 7A,B and EV3H), and a decrease in protein expression of mature cell-type markers including oligodendrocytes (OLIG2), astrocytes (GFAP), as well as progenitor cells (ASCL1) (Fig. EV3I). We next analyzed the stem cell marker, SOX2 (Fig. 7C), and the progenitor cell marker, TBR2 (Fig. 7D), via immunohistochemical analysis of R342X E14 coronal brain sections (Fig. 7F). Percent population of both cell types were imaged and quantified in the ventricular zone (VZ) and subventricular zone (SVZ), which are regions of high stem cell density. We observed a reverse correlation between SOX2 positive (SOX2+) and TBR2 positive (TBR2+) cells in their neurogenic niches, whereby BFLS mice exhibited a higher percentage of SOX2+ cells and an attenuated number of TBR2+ cells (Fig. 7C,D,F), with no significant differences observed in the percentage of merged SOX2 + /TBR2+ cells (Fig. 7E,F). A similar trend of increased SOX2+ cells was noted in the *Phf6*$^{-/Y}$ / *Nestin-Cre*$^+$ brains (Fig EV3D).

The changes in the proportion of SOX2+ and TBR2+ cells suggest altered cell populations manifesting a disproportionate number of neural stem versus progenitor cells in BFLS, which may contribute to disease pathogenesis. In parallel studies, Nissl staining analyses revealed that similar to *Phf6* KO mice, R342X brain sections exhibited a decrease in neuronal density throughout the cortex (Fig. EV3J), suggesting the possibility of impaired neuronal migration. We thus set out to analyze the impact of *Phf6* deletion on cortical layer neurons via subjecting *Phf6*$^{loxP/Y}$ / *Nestin-Cre*$^-$ and *Phf6*$^{-/Y}$ / *Nestin-Cre*$^+$ brain sections at P0 to immunohistochemical analysis using antibodies to SATB2$^+$, CTIP2$^+$, and TBR1$^+$ to quantify neuronal numbers in cortical layers II-V, layer V, and layer VI, respectively (Fig. EV3E,F). Our results revealed a shift of SATB2+ neurons away from the apical cerebral cortex plate in *Phf6*$^{-/Y}$ / *Nestin-Cre*$^+$ mice, with no significant changes in the number of SATB2+, CTIP2+, and TBR1+ neurons. To quantify the migration patterns of SATB2+ neurons in the cerebral cortex influenced by loss of *Phf6*, a grid consisting of 10 equivalent bins was applied to the image of P0 cerebral cortex to equally divide the cortical wall spanning from the basal of ventricle zone to the pial surface into ten bins. The ten bins were marked sequentially from apical to basal, with bin 1 covering the most superficial (i.e., apical) layer, and bin 10 covering the deepest (i.e., basal) layer. Neurons within each bin were counted and a significant decline in SATB2+ neurons in bin 1 of the cerebral cortex was observed in *Phf6*$^{-/Y}$ / *Nestin-Cre*$^+$ mice (Fig. EV3E,F), suggesting impairment in the ability of SATB2+ neurons to migrate to superficial layers of the developing cerebral cortex in *Phf6* KO mice. This finding is consistent with the attenuation of neuron density (Fig. EV3C) and suggests that PHF6 is involved in regulating the process of radial neuronal migration during the establishment of cortical lamination.

Together, we report that PHF6 alters the mechanisms that regulate NSC fate in the developing brain, and that loss-of-function of PHF6 in BFLS results in an imbalance in the number of uncommitted stem cells and neural progenitors which may contribute to BFLS pathogenesis.

# Discussion

In the present study, we report the discovery of a PHF6/EphR transcriptional pathway in the regulation of NSCs in the developing

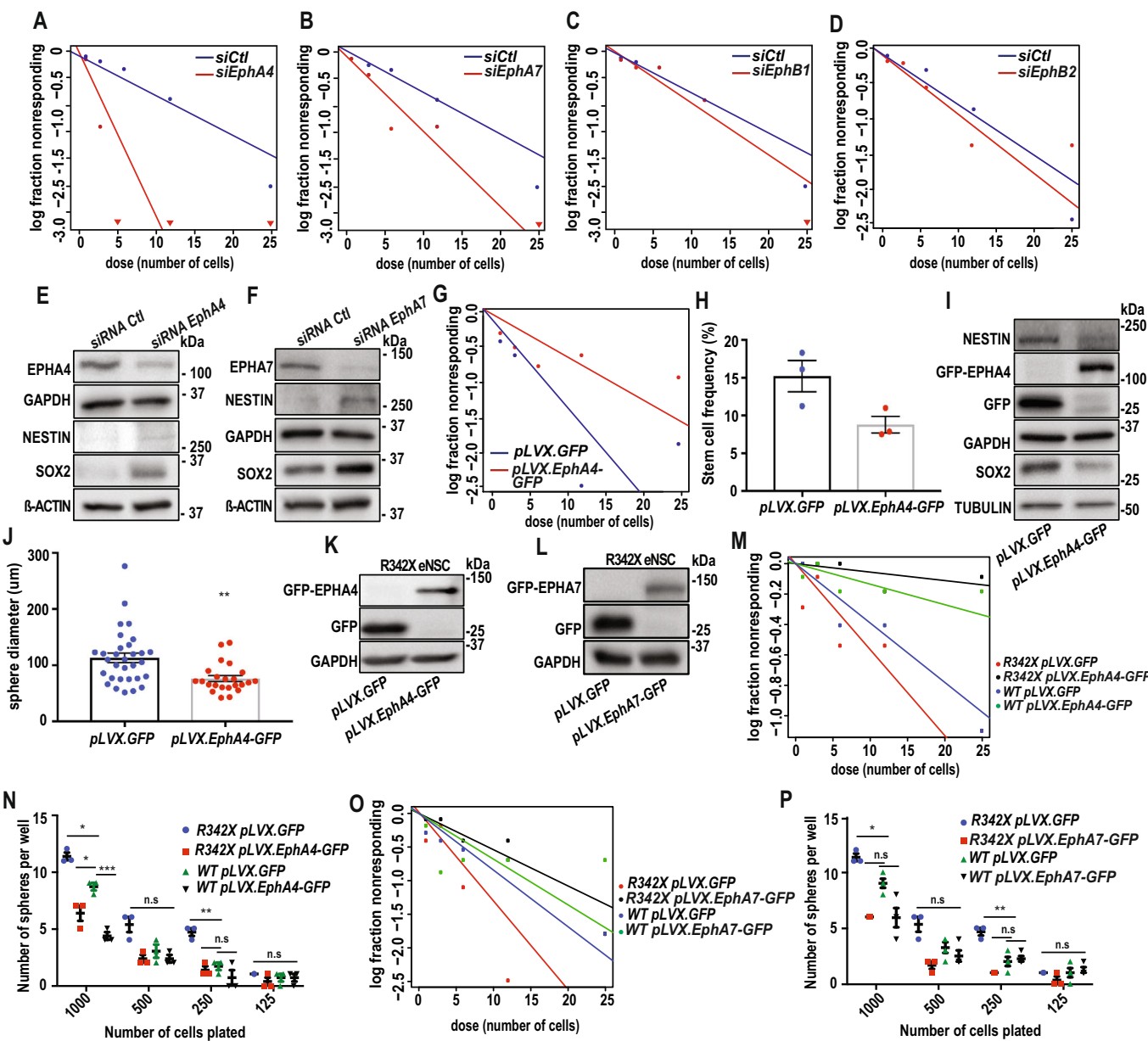

**Figure 6. EphA-family of receptors rescues the eNSC phenotype in R342X mice.**

(A–D) WT eNSCs cultured at E14 were electroporated with siRNA targeting each of the *EphR* followed by self-renewal analysis. ELDA plots are presented for *EphA4* (A) ($p > 0.00001$), *EphA7* (B) ($p = 0.0219$), *EphB1* (C) ($p = 0.426$), and *EphB2* (D) ($p = 0.569$). (E,F) Protein expression of each EPHR, SOX2 and NESTIN were analyzed by immunoblotting. B-ACTIN was used as loading control. (G–J) E14 WT eNSCs were electroporated with pLVX.GFP and pLVX.*EphA4*-GFP constructs followed by ELDA (G) ($p = 0.00355$), and stem cell frequency analysis (H) ($p = 0.0527$), immunoblotting using EPHA4, NESTIN, SOX2, and GFP antibodies (I), and sphere diameter analysis (J) ($p = 0.0017$). (K–P) R342X and WT eNSCs cultured at E14 were electroporated with pLVX.GFP, pLVX.*EphA4*-GFP, and pLVX.*EphA7*-GFP and samples were subjected to immunoblotting analysis with EPHA4, EPHA7, and GFP antibodies (K,L), ELDA (M,O), and sphere analysis (N,P) following 7 days in culture [$p = 0.00264$ (M) and $p = 0.00255$ (O)]. Data information: Data are presented as mean ± SEM. *$p < 0.05$, **$p < 0.01$, ***$p < 0.001$. (H,J) two-tailed unpaired student *t*-test, (N,P) One-way ANOVA with Tukey's multiple comparisons test. Representative data of $n > 3$ independent replicates are shown in panels (A–G,I,K–M,O). Data in panels (H,J,N,P) are plotted with $n > 3$ mean $+/-$ SEM. n represents an independent biological sample. Source data are available online for this figure.

brain. To begin with, mapping PHF6 sites of occupancy in the developing mouse cortex led to the identification of PHF6-bound regions, enriched near genes involved in central nervous system development and neurogenesis. Via a combination of gene expression profiling and PHF6 sites of occupancy, we established a dual function for PHF6 as both a transcriptional activator and

repressor, depending on its binding pattern to the genome. Importantly, we established that PHF6 regulates neurogenesis via altering eNSC fate. Mechanistically, we report that members of EphRs including *EphA4, EphA7, EphB1*, and *EphB2* serve as downstream targets of PHF6. EphRs play crucial roles in the proper formation of the brain (Gerstmann and Zimmer, 2018; Kullander

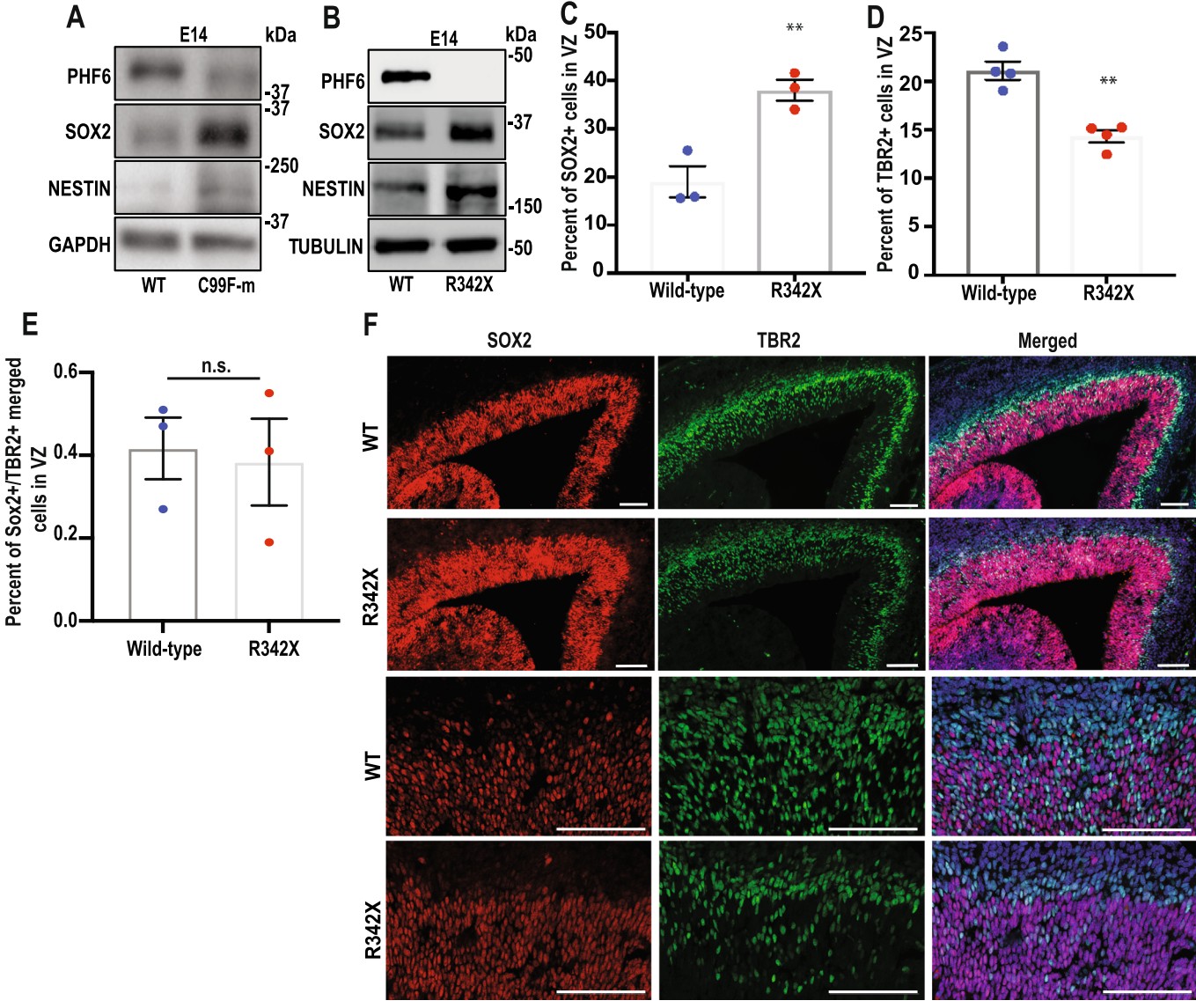

**Figure 7. BFLS patient mouse models exhibit imbalance in the percent population of stem cell and neural progenitors.**

(A,B) Protein expression of PHF6, SOX2, and NESTIN in C99F-m (A) or R342X (B) E14 brains were analyzed with immunoblotting. GAPDH or TUBULIN were used as loading controls. (C–F) E14 brains were sectioned at a thickness of 8 μm and were subjected to staining using SOX2 and TBR2 antibodies. DAPI was used as a nuclei marker. Percentage of SOX2+ (C) (p = 0.0084), TBR2+ (D) (p = 0.001), and SOX2+/TBR2+ merged (E) cells were quantified using FIJI software. Representative images are shown (F). Scale bar represents 100 μm. Data information: Data are presented as mean ± SEM. *p < 0.05, **p < 0.01, ***p < 0.001 for panels (C–E), two-tailed unpaired student t-test (n > 3 independent replicates). Data in (A,B) represents 3 biological replicates (n = 3 mice). Source data are available online for this figure.

and Klein, 2002; Park, 2013). We show that PHF6 directly binds the gene regulatory regions of the identified EphRs to upregulate their expression. Importantly, characterization of BFLS mice including R342X and C99F revealed that EphRs are significantly impacted in BFLS. Furthermore, we generated a conditional *Phf6* KO mouse and confirmed our observations from the BFLS mice whereby impaired NSC pool and deregulation of EphRs resulted from *Phf6* genetic deletion. Finally, we report that although EphA and EphB members function downstream of PHF6, EphA members play the most profound roles in altering eNSC fate. Our results suggest that EphA-receptors could serve as a potential therapeutic target for BFLS. These studies not only shed mechanistic insights on BFLS

and XLID but opens up new avenues of research for impaired NSC processes in other neurodevelopmental disorders of cognition.

There are contradicting reports on the binding of PHF6 to either histones or double-stranded DNA (dsDNA) (Liu et al, 2014; Soto-Feliciano et al, 2017; Todd and Picketts, 2012; Xiang et al, 2019). Our study suggests that PHF6 directly binds DNA to regulate transcription in the developing brain. In particular, we find enrichment of (CA)n repeats in PHF6 peak summits, consistent with a previous study in T-cell acute lymphoblastic leukaemia (T-ALL) where PHF6 was also shown to bind (CA)n repeats (Binhassan, 2020). However, whether PHF6 regulation of the genome could also be epigenetically encoded in the context of BFLS

pathogenesis remains a subject for future studies. In investigating the pattern of PHF6 binding to the genome, we found enrichment in the 5' UTR and TSS consistent with previous studies in B-cell leukemia where PHF6 was shown to bind to the TSS, the 5' UTR (Soto-Feliciano et al, 2017), and enhancer regions in a model of acute myeloid leukemia (AML) (Pawar et al, 2021). Notably, consistent with our findings in stem cell regulation, other groups have also reported a role for PHF6 in cell differentiation (Pawar et al, 2021) and lineage specification (Soto-Feliciano et al, 2017) in leukemia myeloid cell models.

Our analyses suggesting that PHF6 functions as a transcriptional activator or repressor depending on its binding pattern, could also describe the association of PHF6 with the PAF1 complex (Jahani-Asl et al, 2016; Zhang et al, 2013), as the PAF1 complex can either occupy the promoter and gene body of actively transcribed genes and associates with Pol II to promote transcriptional elongation (Pokholok et al, 2002; Wood et al, 2003), or PAF1 also appears to regulate promoter-proximal pausing of Pol II in mammalian cells (Chen et al, 2015). Mechanistically, we present a model that can help explain the dual role of PHF6 in the regulation of gene expression as an activator or repressor, depending on its binding pattern to the gene bodies downstream of the TSS to promote transcriptional elongation, or to the TSS to halt Pol II recruitment and transcription. However, we found that this pattern applies to 65% of candidate genes identified in our screen. How other factors or co-factors enhance or suppress PHF6's role in the regulation of gene expression requires further investigation.

In the present study, we employed a combination of genome-wide studies, conditional *Phf6* KO mice, and BFLS patient mouse models to characterize the mechanisms by which PHF6 regulates gene expression and NSCs in the developing brain. We report a role of PHF6 in the regulation of eNSC fate in the developing brain whereby PHF6 loss-of-function leads to an imbalance of proper fate commitment of NSCs. However, GO term analysis also revealed the upregulation of cation channels (Fig. 2B). Cation channels are vital for action potential generation and propagation, synaptic transmission, and overall neuronal communication and functioning (Chen and Lui, 2019). The upregulation of cation channel activities might represent a compensatory mechanism to enhance neuronal function or to accelerate certain aspects of neuronal maturation given the developmental delays observed in BFLS.

EphA4 is of particular importance amongst the EphRs in the context of stem cell processes and is a widely studied receptor of the ephrin family. High expression of EphA4 is present in hippocampal endothelial cells, mature astrocytes, neurons, and neural progenitor cells (Deininger et al, 2008; Goldshmit et al, 2006; North et al, 2009; Todd et al, 2017; Tremblay et al, 2009). Single-cell studies further proved that EphA4 is expressed in neuroblasts (Todd et al, 2017). Previously, overexpression of *EphA4* in neural progenitor cells in the cortex was shown to cause a decrease in stem cell frequency (North et al, 2009), specifically through ephrinB1-initiated signalling. However, another recent study showed that inhibition of *EphA4* via an antagonist that blocks EphA4 forward signalling, increased proliferation of hippocampal precursor cells (Zhao et al, 2019). In yet another recent study, EphA4 activity via ephrinA1 and VEGFR2 was shown to play a role in neural stem and progenitor cell (NSPC) differentiation (Chen et al, 2020). These results suggest that EphA4 functions in a cell-type and stimuli-dependent manner to confer different outcomes.

Previous studies suggest that EphRs play important roles in cell fate specification (Aoki et al, 2004; Vazin et al, 2009; Wilkinson, 2014). The upstream regulators of EphR remain largely unknown. Here we identify PHF6 as a key upstream regulator of EphR expression and function. Specifically, our data suggest that EphA family members profoundly alter the fate of NSCs suggesting its potential as a therapeutic target to rescue PHF6 loss-of-function in BFLS. Although the EphB family members also appear to serve as PHF6 targets, we did not observe a significant phenotype in the regulation of eNSC with EphB1 or EphB2. It remains to be investigated whether the EphB family members are involved in the regulation of other aspects of neural development such as neuronal morphogenesis and migration in the context of BFLS.

## Methods

### Mice generation, housing, and genotyping

All animal experiments were approved by the Animal Care Committee (ACC) at the University of Ottawa in Ottawa, Ontario, Canada, and McGill University in Montreal, Quebec, Canada. Mice were maintained in regular housing conditions with standard access to food and drink in a pathogen-free facility. The R342X mouse model was generated using CRISPR/Cas9 and functions as a truncated PHF6 protein (Chao et al, 2010; Crawford et al, 2006; Gecz et al, 2006; Jahani-Asl et al, 2016; Lower et al, 2004; Lower et al, 2002; Todd et al, 2015) This strain was generated through the breeding of R342X female heterozygous (HET) mice with C57BL6/J WT (B6 WT) male mice. Hemizygous (HEMI) males were used as experimental mice, and B6 WT males were used as a control. The C99F-m mouse model was generated using CRISPR/Cas9 where cysteine-99 is replaced with phenylalanine (C99F) at nt.29 G > T (Cheng et al, 2018). This strain was generated through breeding C99F-m female HET mice with B6 WT male mice. HEMI males were used as experimental mice, and B6 WT males were used as control.

The *Phf6*$^{-/Y}$ / *Nestin-CreERT2*$^+$ mouse strain (KO) is generated by a brain-specific deletion of *Phf6* via breeding *Phf6*$^{fl/fl}$ female mice (McRae et al, 2019) with Nestin-CreERT2$^+$ male mice and inducing the Cre recombinase via oral gavage of Tamoxifen (Sigma-Aldrich, T5648) in pregnant dams at E14 and embryos collected 24–48 h later. *Phf6*$^{-/Y}$ / *Nestin-CreERT2*$^+$ were characterized and compared to *Phf6*$^{loxP/Y}$ / *Nestin-CreERT2*$^-$ control mice subjected to tamoxifen administration and used as control in all analyses. Male mice were used throughout.

The *Phf6*$^{-/Y}$ / *Nestin-Cre*$^+$ mouse strain was generated by breeding *Phf6*$^{fl/fl}$ female mice with *Nestin-Cre*$^+$ male mice to generate *Phf6*$^{-/Y}$/*Nestin-Cre*$^+$ KO males and *Phf6*$^{-/Y}$/ *Nestin-Cre*$^-$ control littermates. Here, the *Phf6* gene was deleted from the mouse central and peripheral nervous system from E11.5 (Tronche et al. 1999), which is the onset of *Nestin* gene expression.

For genotyping, mouse tissue (tail or ear clipping) was first lysed in alkaline lysis buffer (25 mM NaOH, 0.2 mM EDTA pH 12) and then placed in a heat block at 95 °C for 30 min. The samples were then neutralized using an equal volume of neutralization buffer (40 mM Tris-HCl pH 5.0).

For genotyping of C99F-m and R342x, the PCR reaction mixture was set up as follows using Klentaq Thermostable DNA

Polymerase Thermus aquaticus, recombinant, E. coli (Jena Bioscience, #PCR-217L); 2.5 μL 10x PCR buffer, 0.2 μL 25 mM dNTP, 6.5 μL Betaine, 1 μL 10 μM forward primer, 1 μL 10 μM reverse/mutation primer, 0.2 μL Klentaq enzyme, 12.6 μL RNAse-free $H_2O$, 1 μL DNA for a total mix of 25 μL per PCR tube.

For genotyping of $Phf6^{fl/fl}$, the PCR reaction mixture was set up as follows using a 2x Green PCR Master-Mix high performing (ZmTech Scientific, #S2100G); 7.5 μL 2x Green PCR Master-Mix, 0.4 μL 10 μM forward primer, 0.4 μL 10 μM reverse primer, 5.7 μL RNAse-free $H_2O$, 1 μL DNA for a total mix of 15 μL per PCR tube.

For genotyping of Nestin-CreERT2, the PCR reaction mixture was set up as follows using a 2x Green PCR Master-Mix high performing (ZmTech Scientific, #S2100G); 7.5 μL 2x Green PCR Master-Mix, 1.5 μL 0.5 μM oIMR1084 primer, 1.5 μL 0.5 μM oIMR1085 primer, 1.5 μL 0.5 μM oIMR7338 primer, 1.5 μL 0.5 μM oIMR7339 primer, 0.975 μL 6.5% glycerol, 1 μL DNA for a total mix of 15.5 μL per PCR tube.

The genotyping samples were PCR amplified in a Bio-Rad T100 Thermal Cycler using the following program for C99F-m, R342X, $Phf6^{Loxp/Loxp}$: 1. 95 °C for 2 min, 2. 95 °C for 30 s, 3. 60 °C for 30 s, 4. 72 °C for 30 s, 5. repeat steps 2–4 33x, and 6. 72 °C for 4 min.

The Nestin-CreERT2 genotyping samples were PCR amplified using the following program: 1. 94 °C for 2 min, 2. 94 °C for 20 s, 3. 65 °C for 15 s (−0.5 °C per cycle), 4. 68 °C for 10 s, 5. Repeat steps 2–4 10 times, 6. 94 °C for 15 s, 7. 60 °C for 15 s, 8. 72 °C for 10 s, 9. Repeat steps 6–8 28 times, and 10. 72 °C for 2 min.

The Nestin-Cre genotyping samples were PCR amplified using the following program: 1. 94 °C for 2 min, 2. 94 °C for 20 s, 3. 60 °C for 20 s, 4. 72 °C for 25 s, 5. Repeat steps 2–4 35 times, and 6. 72 °C for 2 min.

The PCR-amplified products were run on a 3% agarose gel at 100 V for 40 min for C99F-m, R342X, Nestin-CreERT2, and Nestin-Cre. The $Phf6^{fl/fl}$ PCR amplified products were run on a 3% agarose gel at 100 V for 60 min. See primers listed in Table EV1.

## Induction of Cre recombinase in $Phf6^{fl/fl}$/ Nestin-CreERT2 mice

Pregnant dames (gestation day E14) were given an oral gavage of one 0.1 mL dose of Tamoxifen (Sigma-Aldrich, T5648) at a concentration of 20 mg/mL using a 1 mL syringe and a 22-gauge feeding needle (Instech Solomon, #FTP-22-25-5).

## Immunoblotting

Protein lysates were obtained from whole brain tissue harvested in RIPA lysis buffer containing protease and phosphatase inhibitors (ThermoFisher Scientific, A32959). The concentration of proteins was analyzed by the Bradford Assay (Bio-Rad) with BSA standard. PVDF membranes were activated in Methanol for 5 min and then blocked in 5% BSA in TBST. Membranes were probed with anti-PHF6 (NOVUS, NB100-68262, 1:1000), anti-EphA4 (Thermo-Fisher, 37-1600, 1:500) or (Santa Cruz, sc-365503, 1:100), anti-EphA7 (ThermoFisher, BS-7034R, 1:500) or (R&D Systems, MAB1495, 1:100), anti-EphB1 (Abcam, ab129103, 1:1000), anti-EphB2 (Abcam, ab252935, 1:500), anti-SOX2 (Abcam, ab97959, 1:250), anti-NESTIN (Santa Cruz, sc-23927, 1:100) or (R&D Systems, MAB2736, 1:500), anti-GFP (Abcam, ab1218, 1:1000), anti–GAPDH (Cell Signaling, 2118S, 1:5000), anti-beta-Actin

(Sigma-Aldrich, a5316, 1:2000), alpha-Tubulin (Abcam, 9074, 1:5000), overnight at 4 °C, followed by HRP-conjugated secondary antibody, anti-rabbit IgG HRP (Bio-Rad, 1706515) or anti-mouse IgG HRP (Bio-Rad, 1706516) for 2 h at room temperature. Proteins were visualized with ECL (Bio-Rad), and signals were detected with a Chemidoc imaging system (Bio-Rad).

## Immunoprecipitation (IP)

80 μg of total cell extracts from $Phf6^{loxP/Y}$ / Nestin-CreERT2⁻ or $Phf6^{-/Y}$ / Nestin-CreERT2⁺ eNSCs were employed for immunoprecipitation (IP), using either 1 μg of IgG or PHF6 antibody (NOVUS, NB100-68262, 1:1000). For input, 4 μg of total cell lysates from both $Phf6^{loxP/Y}$ / Nestin-CreERT2⁻ and $Phf6^{-/Y}$ / Nestin-CreERT2⁺ eNSCs were utilized.

## Quantitative real-time PCR

RNA was isolated from cells and whole brain tissue with Trizol (Invitrogen) according to the manufacturer's instructions. Reverse transcription of RNA was performed using 5x All-In-One RT MasterMix cDNA synthesis (Abm, G492). Quantitative real-time PCR was performed using SsoAdvanced™ Universal SYBR®Green Supermix (Bio-Rad, 1725271). Samples were incubated at 25 °C for 10 min, followed by incubation at 42 °C for 15 min, and finally 85 °C for 5 min to inactivate the reaction. See primers listed in Table EV1.

## Immunofluorescence staining of tissue

Mouse brains were fixed in 4% paraformaldehyde (PFA) for 24 h, followed by 24 h of 15% sucrose fixation, and another 24 h of 30% sucrose fixation before being snap frozen in OCT on dry ice. 8 μm frozen sections were cut using a cryostat. Antigen retrieval was performed on sections prior to blocking by submerging slides in a slide holder with Dako Target Retrieval Solution (Agilent, S1699) and heating in a beaker of water for 20 min at 95–98 °C. Sections were then cooled for 15 min and blocked in 20% donkey serum, 0.1% Triton-X, 0.1% Tween in PBS, for 20 min at room temperature. We applied the SOX2 (1:250) antibody (Abcam, ab97959) and the TBR2 antibody (1:50) (ThermoFisher, 14-4875-82) overnight at 4 °C in a humid chamber. Secondary antibodies (1:500); Anti-rabbit IgG, Alexa Fluor® 647 Conjugate (Cell Signaling, 4414 S), Anti-rat IgG Alexa Fluor® 488 Conjugate (Cell Signaling, 4416 S), and DAPI (1:1000 of 1 μg/ml) (ThermoFisher, D1306) were applied for 45 min at room temperature in a humid chamber. Slides were mounted with ProLong Gold Antifade Mountant (ThermoFisher, P36934) with a #1.5 coverslip. Images were obtained with a laser scanning confocal microscope (ZEISS LSM 800) at 20× objective. Detection wavelengths were as follows: DAPI detection 400–605, TBR2 (AF488) 400–650, SOX2 (AF647) 645–700, and all with a detector gain of 650 V.

For PHF6 and coronal layer marker immunofluorescent brain section staining, brains were fixed with 4% PFA and equilibrated in 30% sucrose solution at 4 °C until the brains sank to the bottom of the vail. Brains were immersed in 50% OCT (VWR) solution diluted by 30% sucrose for overnight at 4 °C. Brains were transferred to the cryomold (VWR) filled with 50% OCT and flash frozen in liquid nitrogen. Frozen brains were stored at −80 °C. Brain blocks were subjected to cryosection at the thickness of 12 μm

and mounted onto SuperFrost slides (Fisher Scientific). Sections were then washed three times with PBS and 0.1% Tween-20 detergent (PBST), then antigen retrieval in Citrate buffer (0.1 M, pH 6.0) by microwave boiling for 10 min and blocked in 10% horse serum/PBST for 30 min at room temperature. After blocking, sections were subjected to the following primary antibody for PHF6 immunofluorescence (overnight at 4 °C): rabbit anti-PHF6 (1:150, Sigma-HPA001023), or for coronal layer markers; mouse anti-SATB2 (1:200, Abcam-ab51502), rat anti-CTIP2 (1:200, Abcam-ab18465), and rabbit anti-TBR1 (1:200, Abcam-ab31940). The next day, after washing three times in PBST, sections were incubated in secondary antibody for 1 h at room temperature: anti-rabbit 555 Alex Fluor (1:500, Invitrogen A21206), anti-mouse 488 Alexa Fluor (1:500, Invitrogen A21202), or anti-rat 647 Alexa Fluor (1:500, Invitrogen A21247). Nuclei were counterstained by incubating sections in Hoechst 33342 dye (ThermoFisher Scientific) for 15 min at room temperature. Finally, slides were mounted onto coverslips (Fisher Scientific) in DAKO Fluorescence Mounting Medium (Agilent Technologies).

## Histology staining

For Nissl staining, brain sections were rehydrated by 10 min submersions in 95% ethanol, followed by 1 min submersion in 70% ethanol and 1 min submersion in 50% ethanol. Sections were rinsed in tap water and then in distilled water. After washing, sections were stained in 0.25% Cresyl Violet Stain Solution in distilled water for 5 min, followed by a quick wash in distilled water. Sections were quickly differentiated in 70% ethanol with 1% acetic acid for 10 s to 1 min and checked under the microscope. Sections were then dehydrated via two 5-min submersions in 100% ethanol. Finally, slides were cleared by three 5-min submersions in xylene and mounted onto coverslips (Fisher Scientific) with Permount Mounting Medium (Fisher Chemical). Stained slides were air-dried overnight in the fume hood at room temperature. Immunofluorescent images were acquired by using Zeiss Axiovert Observer Z1 epifluorescent/light microscope equipped with an AxioCam cooled-colour camera (Zeiss) or SP8 confocal microscope (Leica). Nissl-stained slides were scanned by a Zeiss AxioScan Z1.

## Embryonic neural stem cell culture

Embryonic NSCs (eNSCs) were obtained by whole brain culturing of E14 mice (Burban and Jahani-Asl, 2022; Nasser et al, 2018) (excluding cerebellum). Pregnant mice were euthanized, uterine horns were removed, and embryos were placed in cold 1× HBSS. Brain tissue was cut into small pieces and placed in 15 mL falcon tubes containing 1 mL cold 1× HBSS. Tissue was allowed to settle to the bottom, HBSS was replaced with 1 mL fresh HBSS for washing, and then replaced once more with 1 mL stem cell media (SCM) containing 1:1 DMEM-F12 (Wisent, 319-005-CL) (ThermoFisher, 31765035), 50 units/mL penicillin-streptomycin (Wisent, 450-201-EL), 1× B-27 supplement (Invitrogen, 17504044), 2 µg/mL Heparin (Stemcell Technologies, 07980), 20 ng/mL mEGF (Cell Signaling, 5331SC), 12.5 ng/mL bFGF (Abbiotec, 600182). The tissue was mechanically dissociated 15× with P1000 then an additional 15× with P200. The lysate was then plated in 6 mL of SCM and left in the incubator for 6–7 days until spheres grew to 40–200 µm in size, replenishing with 2 mL SCM media at day 4.

## Analysis of self-renewal and proliferation

For the limiting dilution assay (LDA), NSCs were dissociated to single-cell suspension using Accumax. Single cells were counted and plated in a 96-well plate at different cell doses per well, in triplicates. Spheres were counted 7 days post-plating.

For the extreme limiting dilution assay (ELDA), NSCs were dissociated to single-cell suspension using Accumax. Single cells were counted and plated in a 96-well plate at different cell doses per well with a minimum of 12 wells/cell dose (Rasool et al, 2022). 7 days post-plating, the presence or absence of spheres in each well was recorded and analyzed with http://bioinf.wehi.edu.au/software/elda/33 (Hu and Smyth, 2009).

For cell viability, NSCs were dissociated to single-cell suspension using Accumax. Single cells were counted and seeded at a density of 200 cells/well, in a 96-well plate. Cell viability was evaluated 7 days post-plating using alamarBlue (Thermo Fisher Scientific, #DAL1100) according to the manufacturer's protocol. 10% resazurin was added to the cells in each well and incubated for 4 h at 37 °C. Fluorescence was read using a fluorescence excitation wavelength of 560 nm and an emission of 590 nm.

Representative images of spheres were taken with the 10× objective lens of an Olympus IX83 microscope with an X-Cite 120 LED from Lumen Dynamics, and an Olympus DP80 camera.

## 5-ethynyl-2′-deoxyuridine (EdU) proliferation assay

eNSCs were dissociated into single-cell suspension using Accumax, counted and plated at a density of $1 \times 10^6$ cells. Cells were incubated with 10 µM EdU upon plating. Following 22 h in culture, eNSCs were fixed, permeabilized, and stained using the Click-iT EdU proliferation kit (Thermo Fisher Scientific, #C10337) according to the manufacturer's protocol. Fluorescence was analyzed by flow cytometry (BD FACS CantoII & Sony SH800). Data were analyzed using the FlowJo software. The number of cells that had incorporated EdU was defined as the ratio of EdU-positive cells over total number of cells.

## Chromatin immunoprecipitation (ChIP)

PBS containing protease inhibitors (Thermo Fisher Scientific, #A32959) was used as cell washing buffer prior to fixation. Cross-linking was done via 1% formaldehyde in PBS for 10 min and quenched with 0.125 M glycine in PBS for 5 min at room temperature (RT). Washing, fixation, and quenching was done in 15 mL tubes while rotating at RT. Post-quenching, cells were washed twice with PBS containing protease inhibitors. Cells were then pelleted by spinning at $150 \times g$ for 10 min at 4 °C. Cell pellets were dissolved in ChIP lysis buffer (40 mM Tris-HCl, pH 8.0, 1.0% Triton X-100, 4 mM EDTA, 300 mM NaCl) containing protease inhibitors. Chromatin fragmentation was performed through water bath sonication (BioRuptor) at 4 °C, creating an average length of 500 base pairs (bp) of product. Cell lysates were spun down at 12,000 G for 15 min, followed by dilution of supernatant (1:1) in ChIP dilution buffer (40 mM Tris-HCl, pH 8.0, 4 mM EDTA, protease inhibitors).

Immunoprecipitation (IP) was done using a PHF6 antibody (Novus Biological, NB100-68262), rabbit IgG antibody (Cell Signaling, #3900 S). Antibody-protein-DNA complexes were

collected, washed, and then eluted. Reverse cross-linking was done as described in Soleimani et al, 2013 (Soleimani et al, 2013). Immunoprecipitated DNA was analyzed by qPCR, and the binding enrichment was expressed as a percentage of the input.

## Dual-luciferase reporter assay

The PHF6 binding regions (based on ChIP-seq peaks) were cloned into the pGL4.23 (Promega) vector to generate the *EphA4*, *EphA7* and *EphB1* luciferase reporter genes by digesting the plasmid and the annealed primer pair using EcoRV (NEB, #R0195L) and KpnI (NEB, #R3142) then ligating them with T4 DNA ligase (NEB, #M0202L). The constructs were confirmed by DNA sequencing. Cells were electroporated with the *EphA4*-pGL4.23, *EphA7*-pGL4.23, *EphB1*-pGL4.23 or the empty pGL4.23. Luciferase assays were performed 48 h after transfection with the Dual-Luciferase Reporter Assay system (Promega, #E1910) with a GloMax Luminometer (Promega). In all experiments, cells were electroporated with a Renilla firefly reporter control and the firefly luminescence signal was normalized to the Renilla luminescence signal. See primers listed in Table EV1.

## siRNA

Transient KD of *Phf6* and *EphA4/A7/B1/B2* using an siRNA approach was performed with ON TARGET-plus SMART pool mouse *Phf6* siRNA (Dharmacon, #L-058690-01-0005), mouse *EphA4* siRNA (Sino Biological, #MG50575-M), mouse *EphA7* siRNA (Sino Biological, #MG50587-M, mouse *EphB1* siRNA (Sino Biological, #MG50479-M), mouse *EphB2* siRNA (Santa Cruz, #sc-39950), and ON TARGET-plus non-targeting pool (Santa Cruz, #sc-36869). siRNA (100 nM) were nucleofected into eNSCs ($10^6$ cells) and cultured in eNSC media at 37 °C in a humidified atmosphere of 5% $CO_2$.

## Leveraging published sequencing datasets

Single cell RNA-seq data from the mouse cerebral cortex was obtained [Data ref: (Di Bella et al, 2021)]. Log normalized counts, cell type annotation and UMAP coordinates were retrieved from the original publication and used to generate UMAP plots. For the correlation analysis, MAGIC (Van Dijk et al, 2018) was applied to obtain imputed gene expression. Correlation values were obtained on the imputed gene expression after applying MAGIC.

Normalized RPKM (Reads per Kilobase Million) values of RNA-seq data were obtained from the Allen Brain Atlas BrainSpan dataset [Data ref: (BrainSpan Atlas of the Developing Human Brain, 2011)] and data from the ventral frontal cortex (VFC) was taken. The average RPKM values was calculated per developmental time. All plots were generated using R (version 4.0.0).

## ChIP-seq data processing

ChIP-seq was performed by pooling the cortex of three mice (*n* = 3) prior to sequencing. ChIP-seq data were processed as previously described (Hernandez-Corchado and Najafabadi, 2022). Briefly, raw reads were aligned to the mouse genome assembly version mm10 with bowtie2 (version 2.3.4.1) using the "--very-

sensitive-local" mode. Duplicate reads were removed using samtools (version 1.9) (Danecek et al, 2021). ChIP-seq peaks were identified using MACS (version 1.4) (Feng et al, 2012; Zhang et al, 2008) with a permissive p-value threshold of 0.001, using "--nomodel" option. Fragment size was specified using "--shiftsize" argument, with the fragment length obtained by cross-correlation analysis using phantompeakqualtools (Landt et al, 2012). Peak-TSS distances were calculated using bedtools (Quinlan and Hall, 2010) only for peaks that passed *p*-value threshold of $10^{-5}$, with TSS coordinates obtained from GENCODE (Frankish et al, 2019) (release M9).

## Identifying Pol II occupancy and its intersection with PHF6 data

Pol II occupancy data were obtained from GEO (accession number GSM2442441) (Liu et al, 2017). The bedGraph file representing Pol II occupancy was directly downloaded from GEO, converted to bigWig, and overlaid on gene TSS coordinates using bwtool (Pohl and Beato, 2014).

## mRNA-seq

Cortical progenitors were established from the cortex of wild-type E14 mice and subjected to electroporation with *Phf6* siRNAs (*n* = 3) and non-targeting control siRNA (*n* = 3). Cell were subjected to mRNA-Seq analysis following 5 days in culture. mRNA-seq raw reads were mapped to mm10 genome using HISAT2 (Kim et al, 2015), followed by duplicate read removal using samtools. Gene-level read counts were obtained by HTSeq (Anders et al, 2015), using gene annotations from GENCODE (release M9). Genes with a minimum of 150 reads in at least one sample were retained. Gene set analysis was performed using ConsensusPathDB (Kamburov et al, 2011).

## Quantification and statistical analysis

Statistical analysis was performed with the aid of GraphPad software 7. Two-tailed unpaired student t-tests were used to compare two conditions (normal distribution). One-way ANOVA was used for analyzing multiple groups (normal distribution). Data are shown as mean with standard error of mean (mean ± SEM). *p*-values of equal or less than 0.05 were considered significant and were marked with one asterisk (*). *p*-values of less than 0.01 are denoted by **, and *p* values of less than 0.001 are denoted by ***. All data presented are from 3 or more independent biological (n) replicates (*n* ≥ 3), unless otherwise noted in corresponding figure legends, thus no additional statistical methods were used to predetermine sample size. Randomization was used to allocate animals to experimental groups, following genotyping. The researchers were blind to treatment groups for all quantifications as well as imaging analysis. Only male mice were included in this study. Methods of statistical analysis and *p*-values employed are reported in corresponding figure legends.

## Graphics

Synopsis graphic was created with BioRender.com.

## Data availability

The datasets produced in Figs. 1, 2 are available in the following database for both datasets at Gene Expression Omnibus (GEO GSE247838). Figure 4 data are available at (GEO GSE153164) and the Single Cell Portal: https://singlecell.broadinstitute.org/single_cell/study/SCP1290/molecular-logic-of-cellular-diversification-in-the-mammalian-cerebral-cortex.

## Peer review information

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

## Acknowledgements

This work was supported by grants from NSERC (RGPIN-2016-00605) and CIHR (PJG-185800) to AJ-A. AJ-A and HSN hold Canada Research Chairs. DJP was supported by a CIHR grant (PJT-159619), and HSN is supported by Compute Canada resource allocations and NSERC grant (RGPIN-2018-05962). AB was supported by an FRQS fellowship. AMA was supported by a training scholarship from FRQS. We thank staff at the University of Ottawa Animal Care and Veterinary Services. The authors acknowledge the Cell Biology and Image Acquisition Core (RRID: SCR_021845) funded by the University of Ottawa, Ottawa, Natural Sciences and Engineering Research Council of Canada, and the Canada Foundation for Innovation. We thank the team at the Histology/Imaging/Staining services provided by the Louise Pelletier HCF (RRID: SCR_021737) at the University of Ottawa. The authors acknowledge the use of instruments and expertise from the uOttawa Flow Cytometry Core Facility, which is funded by the University of Ottawa, Natural Sciences and Engineering Research Council of Canada, and the Canadian Foundation for Innovation. The Authors acknowledge and thank Aldo Hernández Corchado for help with uploading datasets.

## Author contributions

**Dilan Rasool**: Software; Investigation; Visualization; Methodology; Writing—original draft; Writing—review and editing. **Audrey Burban**: Software; Investigation; Visualization; Methodology; Writing—review and editing. **Ahmad Sharanek**: Software; Investigation; Visualization; Methodology; Writing—review and editing. **Ariel Madrigal**: Software; Visualization; Methodology. **Jinghua Hu**: Visualization; Methodology. **Keqin Yan**: Visualization; Methodology. **Dianbo Qu**: Visualization; Methodology; Writing—review and editing. **Anne K Voss**: Resources. **Ruth S Slack**: Resources. **Tim Thomas**: Resources. **Azad Bonni**: Resources. **David J Picketts**: Supervision; Funding acquisition; Writing—review and editing. **Vahab D Soleimani**: Resources; Visualization; Methodology; Writing—review and editing. **Hamed S Najafabadi**: Software; Supervision; Funding acquisition; Investigation; Visualization; Methodology; Writing—original draft; Writing—review and editing. **Arezu Jahani-Asl**: Conceptualization; Resources; Supervision; Funding acquisition; Investigation; Visualization; Methodology; Writing—original draft; Project administration; Writing—review and editing.

## Disclosure and competing interests statement

The authors declare no competing interests.

# Expanded View Figures

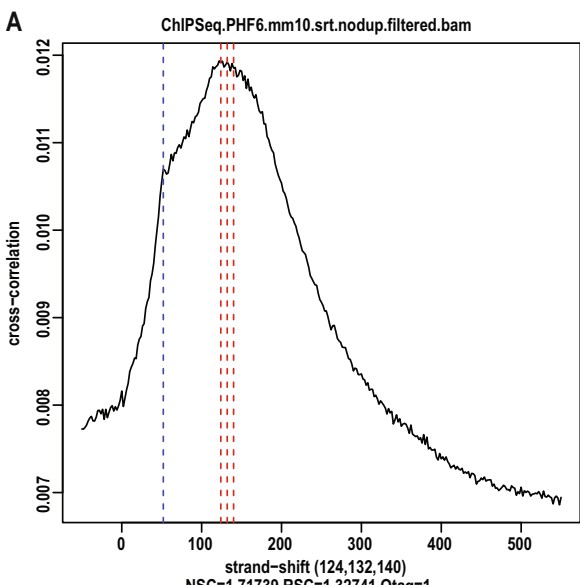

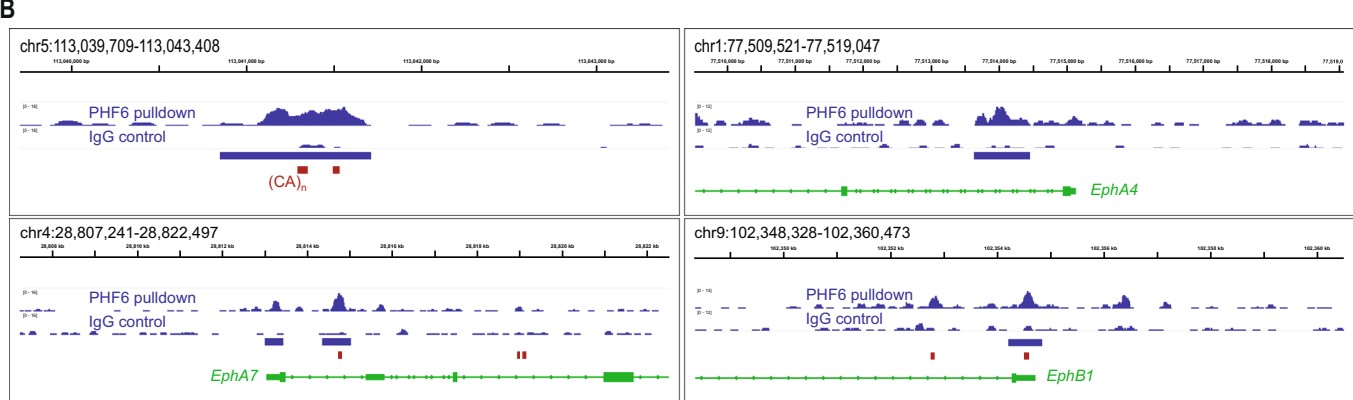

**Figure EV1.  PHF6 ChIP-Seq analysis.**

(A) PHF6 ChIP-seq cross-correlation analysis was conducted using cross-correlation metrics as described in Landt et al, (Landt et al, 2012). (B) Example ChIP-seq tracks for PHF6 pull-down and IgG control. (CA)n repeats are demarcated with red boxes, while the blue boxes represent the identified PHF6 peak.

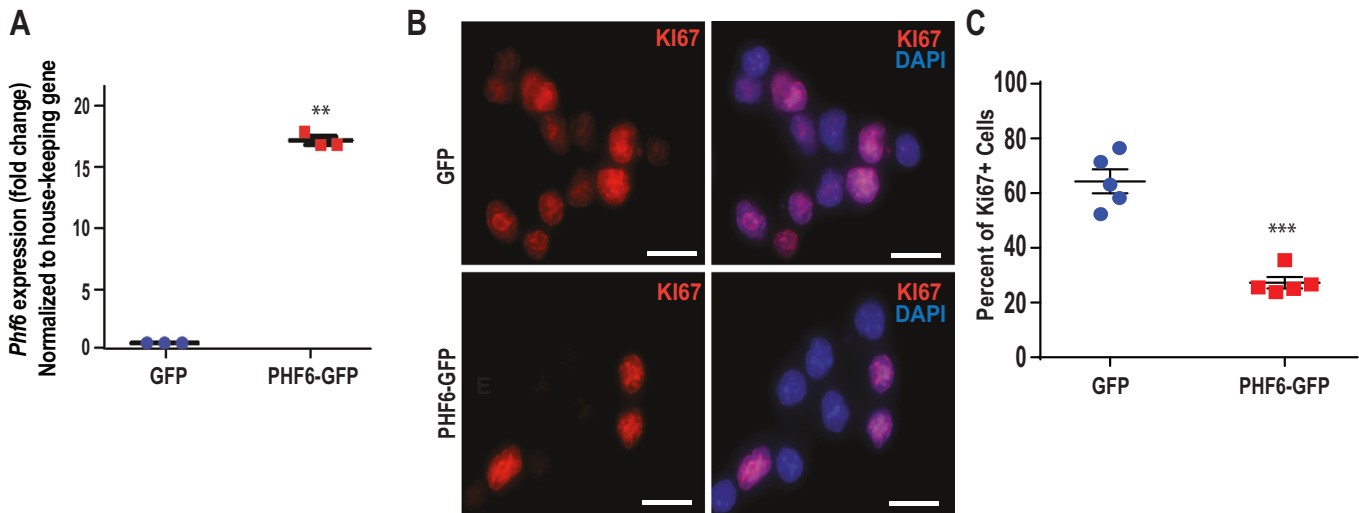

**Figure EV2.  PHF6 regulation of proliferation in neuroblastoma (N2A) cells.**

(A–C) N2A cells were transfected with *Phf6* (PHF6-GFP) or GFP-expressing control (GFP) constructs. (A) Gene expression was assessed by RT-qPCR ($n = 3$). (B) Samples were subjected to KI67 staining for assessment of proliferation ($n > 3$, representative image shown). Scale bar represents 20 µm. (C) Quantification of percent KI67 positive cells are shown ($n > 3$). Data information: Data are presented as mean ± SEM. *$p < 0.05$, **$p < 0.01$, ***$p < 0.001$ (two-tailed unpaired student *t*-test). *n* represents an independent biological sample.

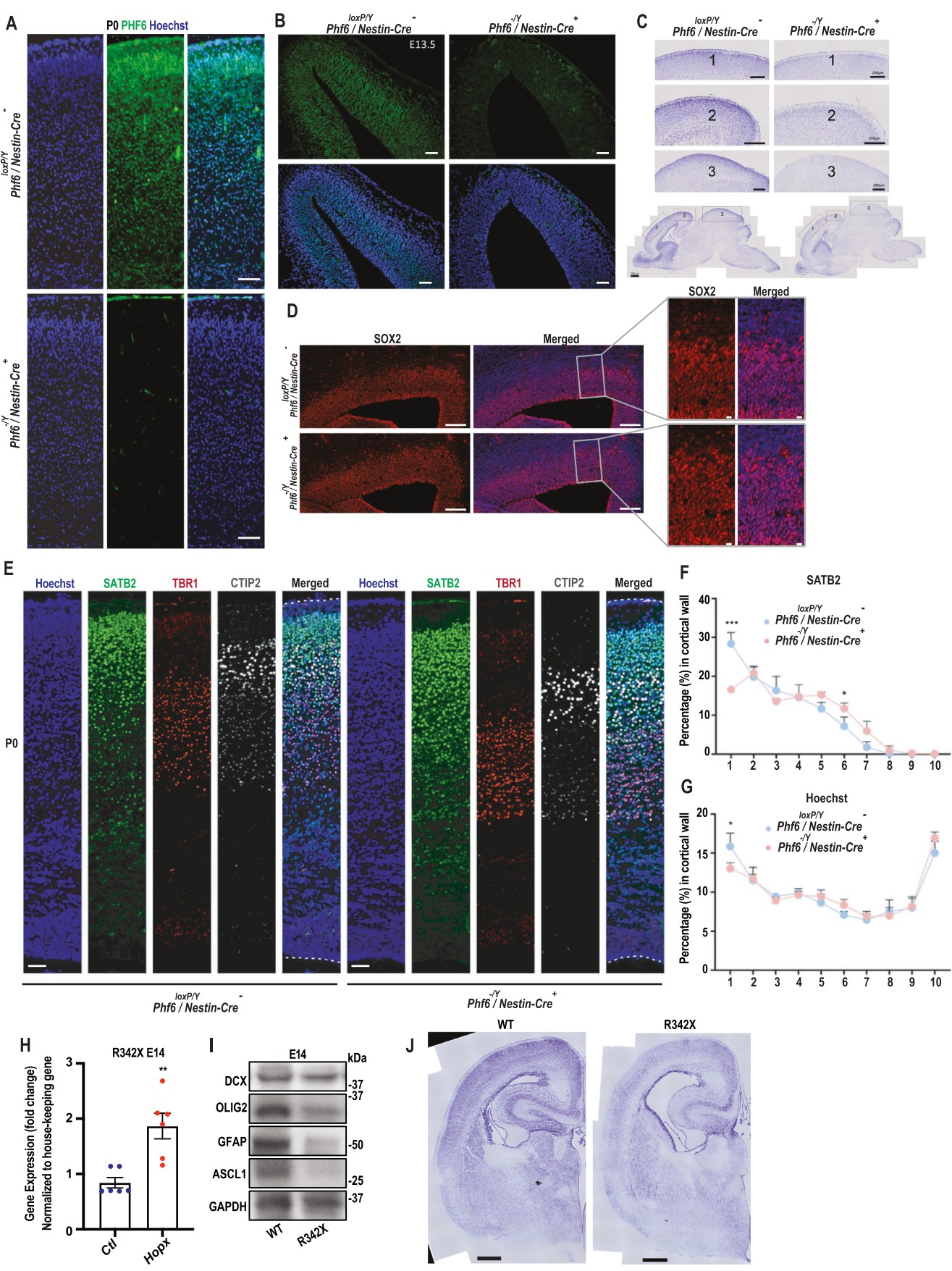

**Figure EV3. Characterization of *Phf6/Nestin-Cre* and BFLS mouse brain development.**

(A,B) Immunofluorescence (IF) staining of coronal sections from P0 (A) and E13.5 (B) for $Phf6^{-/Y}$ / $Nestin-Cre^+$ and $Phf6^{loxp/Y}$ / $Nestin-Cre^-$ male mice using a PHF6 antibody (green) in the cerebral cortex. Nuclei were counterstained by Hoechst. Scale bars represent 50 μm. (C) $Phf6^{-/Y}$ / $Nestin-Cre^+$ and $Phf6^{loxp/Y}$ / $Nestin-Cre^-$ male mice were collected at P0 and subjected to Nissl staining with sagittal sections shown. Scale bars represent 500 μm in lower magnification and 250 μm in higher magnification photomicrographs. (D) IF staining of coronal sections from ~E15 male mice using a SOX2 antibody is shown. Scale bar represents 100 μm at lower magnification and 10 μm at higher magnification. (E) IF staining of coronal sections from P0 using cortical layer markers: SATB2 (green, layer II-V), TBR1 (red, layer VI), and CTIP2 (grey, layer V). Nuclei were counterstained by Hoechst. The cortical wall spanning from the basal of ventricle zone to the pial surface was equally divided into ten bins, the bin 1 covers the most superficial layer and bin 10 covers the deepest layer. (F) Comparative analysis of SATB2+ neurons in each segment of P0 male mice ($n = 3$). (G) Comparative analysis of Hoechst+ nuclei in each segment of P0 male mice ($n=3$). Scale bars represent 50 μm. (H,I) mRNA and protein of E14 R342X and wild-type control mice were subjected to RT-qPCR for *Hopx* expression ($n > 3$) (H) ($p = 0.0021$), and immunoblotting analysis of cell type-specific markers (I) ($n = 3$, representative blots shown). (J) R342X and WT mice were collected at P0 and subjected to Nissl staining ($n = 2$, representative image shown). Coronal sections are shown. Scale bars represent 500 μm. Data information: Data are presented as mean ± SEM. *$p < 0.05$, **$p < 0.01$, ***$p < 0.001$. two-tailed unpaired student *t*-test (H). two-way ANOVA with multiple comparisons (F,G). *n* represents an independent biological sample. Source data are available online for this figure.

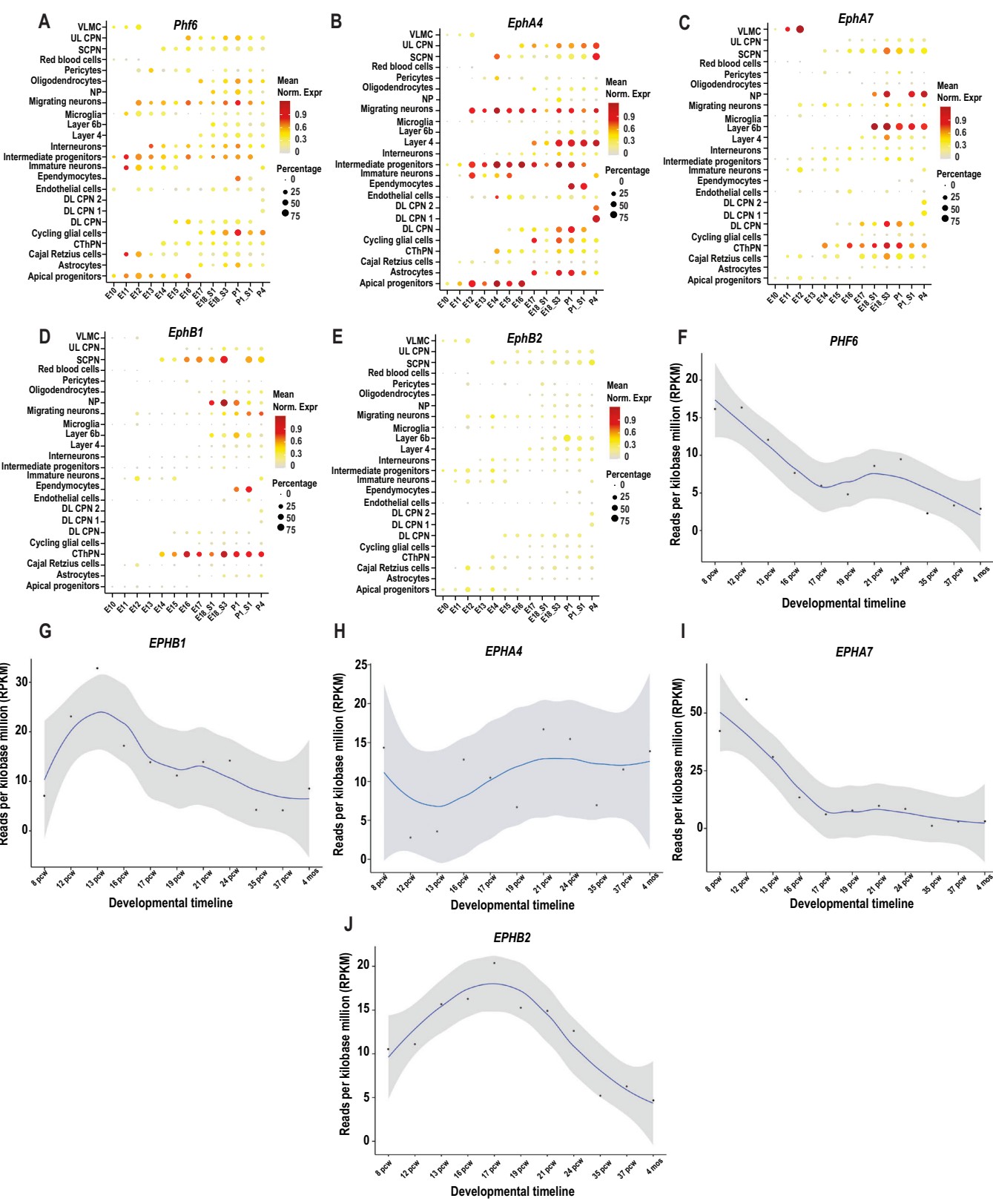

◀   **Figure EV4.   Analysis of *Phf6* and *EphR* mRNA expression across development.**

(**A–E**) Dot plots showing expression of *Phf6* (**A**), *EphA4* (**B**), *EphA7* (**C**), *EphB1* (**D**), and *EphB2* (**E**) in the mouse cerebral cortex during development where the colour of each dot represents the mean normalized expression values per cell type for a given timepoint. The size of the circle represents the percentage of cells expressing each gene. Single cell mouse RNA-seq data was obtained from GEO GSE153164 [Data ref: (Di Bella et al, 2021)]. (**F–J**) Analysis of *PHF6* and *EPHR* expression in the human cortex. Average reads per kilobase million (RPKM) values over human developmental time (post-conceptual weeks; pcw) for gene analysis of *PHF6* (**F**), *EPHB1* (**G**), *EPHA4* (**H**), *EPHA7* (**I**), and *EPHB2* (**J**) are shown. Gene analysis was taken from publicly available RNA-seq data taken from the human ventral frontal cortex (VFC) of the Allen Brain Atlas BrainSpan dataset [Data ref: (BrainSpan Atlas of the Developing Human Brain, 2011)].

   

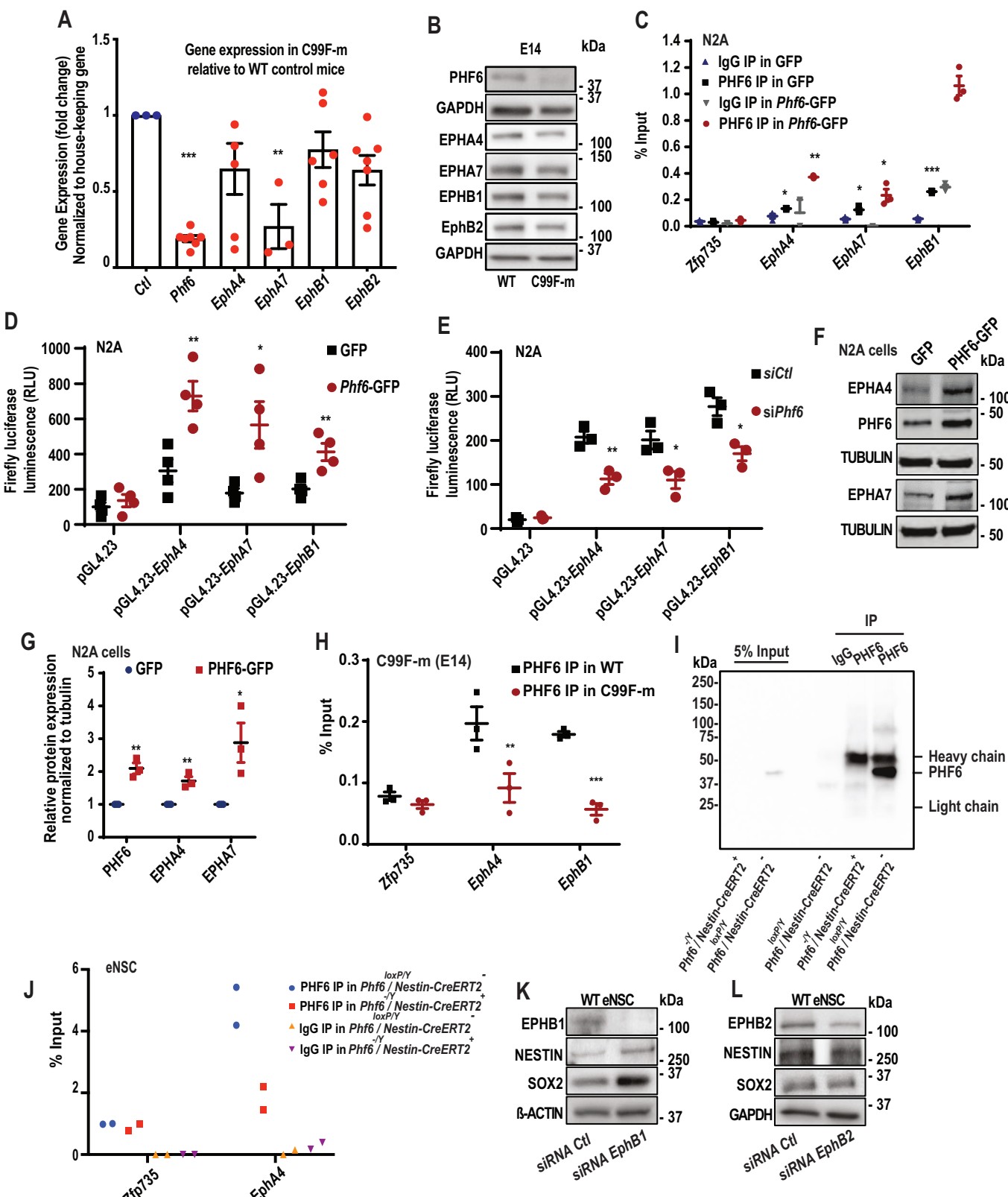

**Figure EV5. Analysis of PHF6 and EphR expression in BFLS mice.**

(A,B) mRNA and protein of E14 C99F-m and wild-type control mice were subjected to RT-qPCR and immunoblotting analysis ($n \geq 3$). (C) GFP or PHF6-GFP expressing N2A cells were subjected to ChIP using an antibody to PHF6 or IgG control followed by PCR analysis using primers to *EphA4*, *EphA7* and *EphB1*. *Zfp* locus was used as control ($n = 3$). (D) GFP or PHF6-GFP- expressing cells were electroplated with a luciferase reporter plasmid driven by a promoter containing 583 bp of the *EphA4* gene (pGL4.23-*EphA4*), 550 bp of the *EphA7* gene (pGL4.23-*EphA7*) or 709 bp of the *EphB1* gene (pGL4.23-*EphB1*). The pGL4.23-basic reporter plasmid (pGL4.23) was used as a control. Renilla expression plasmid was used as an internal control for all samples. RLU Relative luminescence unit. Dual luciferase reporter assay was performed 48 h following electroporation ($n = 3$). (E) N2A cells were electroporated with siRNA against *Phf6* (si*Phf6*) or control siRNA (siCtl) followed by dual luciferase reporter assay at 48 h ($n = 3$). (F) EPHA4, EPHA7 and PHF6 levels were analyzed by immunoblotting in PHF6-GFP- expressing N2A cells. TUBULIN was used as a loading control. (G) Densitometric quantification of PHF6, EPHA4 and EPHA7 protein level normalized to TUBULIN is shown ($n = 3$). (H) E14-Cerebral cortical tissues from WT and C99F-m mice were subjected to ChIP-PCR analysis, as described in panel (C). (I) eNSCs cultured from *Phf6*$^{-/Y}$ / *Nestin-CreERT2*$^{+}$ and control *Phf6*$^{loxP/Y}$ / *Nestin-CreERT2*$^{-}$ ~E15 mouse brains were subjected to immunoprecipitation (IP) using PHF6 antibody or IgG as control followed by immunoblotting analysis using a PHF6 antibody. (J) eNSCs from *Phf6*$^{-/Y}$ / *Nestin-CreERT2*$^{+}$ and control *Phf6*$^{loxP/Y}$ / *Nestin-CreERT2*$^{-}$ mouse brains at ~E15, were subjected to ChIP-PCR using a PHF6 antibody. *Zfp735* loci was used as control for the PCR ($n = 2$). (K,L) Protein expression of EPHB1 (K), EPHB2 (L), SOX2 and NESTIN were analyzed by immunoblotting in *EphB1* and *EphB2* knockdown (KD) cells. Loading controls of ß-ACTIN and GAPDH were used ($n = 2$). Data information: Data are presented as mean ± SEM. *$p < 0.05$, **$p < 0.01$, ***$p < 0.001$. [(C,H) one-way ANOVA, (A,D,E,G) two-tailed unpaired student *t*-test]. *n* represents an independent biological sample.

