## [Peer Review File · EMBO Reports]

PHF6-mediated transcriptional control of NSC via Ephrin receptors is impaired in the intellectual disability syndrome BFLS

Dilan Rasool, Audrey Burban, Ahmad Sharanek, Ariel Aguirre, Jinghua Hu, Keqin Yan, Dianbo Qu, Anne Voss, Ruth Slack, Tim Thomas, Azad Bonni, David Picketts, Vahab Soleimani, Hamed Najafabadi, and Arezu Jahani-Asl

Corresponding author(s): Arezu Jahani-Asl (Arezu.Jahani@uottawa.ca) , Hamed Najafabadi (hamed.najafabadi@mcgill.ca)

Review Timeline:

Submission Date:	13th Jun 23
Editorial Decision:	12th Jul 23
Revision Received:	20th Nov 23
Editorial Decision:	15th Dec 23
Revision Received:	17th Jan 24
Accepted:	22nd Jan 24

Editor: Esther Schnapp

Transaction Report:

Dear Dr. Jahani-Asl

Thank you for the submission of your manuscript to EMBO reports. We have now received the full set of referee reports that is pasted below.

As you will see, the referees acknowledge that the findings are potentially interesting. However, they also all raise a number of concerns and point out that the manuscript provides insufficient details on several materials and methods, which precludes a solid interpretation of the experimental evidence provided. Your manuscript is a borderline case, and all concerns would need to be satisfactorily addressed in order to proceed with this study here.

Also, and as referee 1 notes, all larger datasets should be deposited in a repository and made accessible to reviewers, and latest upon the publication of your study also to the scientific community.

Please let me know if you would like to discuss any aspects of the revisions or if you have any questions or comments. We can also do this in a video chat, if you like.

Taken together, given the potential interest of your findings, and that all concerns can potentially be addressed, I would like to invite you to revise your manuscript with the understanding that the referee concerns must be fully addressed and their suggestions taken on board. Please address all referee concerns in a complete point-by-point response. Acceptance of the manuscript will depend on a positive outcome of a second round of review. It is EMBO reports policy to allow a single round of major revision only and acceptance or rejection of the manuscript will therefore depend on the completeness of your responses included in the next, final version of the manuscript.

We realize that it is difficult to revise to a specific deadline. In the interest of protecting the conceptual advance provided by the work, we recommend a revision within 3 months (12th Oct 2023). Please discuss the revision progress ahead of this time with us if you require more time to complete the revisions.

- 1) A data availability section providing access to data deposited in public databases is missing. If you have not deposited any data, please add a sentence to the data availability section that explains that.
- 2) Your manuscript contains statistics and error bars based on $n=2$. Please use scatter blots in these cases. No statistics should be calculated if $n=2$.

5) a complete author checklist, which you can download from our author guidelines <https://www.embopress.org/page/journal/14693178/authorguide>. Please insert information in the checklist that is also

reflected in the manuscript. The completed author checklist will also be part of the RPF.

6) Please note that all corresponding authors are required to supply an ORCID ID for their name upon submission of a revised manuscript (<<https://orcid.org/>>). Please find instructions on how to link your ORCID ID to your account in our manuscript tracking system in our Author guidelines <<https://www.embopress.org/page/journal/14693178/authorguide#authorshipguidelines>>

7) Before submitting your revision, primary datasets produced in this study need to be deposited in an appropriate public database (see <https://www.embopress.org/page/journal/14693178/authorguide#datadeposition>). Please remember to provide a reviewer password if the datasets are not yet public. The accession numbers and database should be listed in a formal "Data Availability" section placed after Materials & Method (see also <https://www.embopress.org/page/journal/14693178/authorguide#datadeposition>). Please note that the Data Availability Section is restricted to new primary data that are part of this study. * Note - All links should resolve to a page where the data can be accessed. *
If your study has not produced novel datasets, please mention this fact in the Data Availability Section.

12) All Materials and Methods need to be described in the main text. We would encourage you to use 'Structured Methods', our new Materials and Methods format. According to this format, the Materials and Methods section should include a Reagents and Tools Table (listing key reagents, experimental models, software and relevant equipment and including their sources and relevant identifiers) followed by a Methods and Protocols section in which we encourage the authors to describe their methods using a step-by-step protocol format with bullet points, to facilitate the adoption of the methodologies across labs. More information on how to adhere to this format as well as downloadable templates (.doc or .xls) for the Reagents and Tools Table can be found in our author guidelines: <<https://www.embopress.org/page/journal/17444292/authorguide#textformat>>. An example of a Method paper with Structured Methods can be found here: <<https://www.embopress.org/doi/10.15252/msb.20178071>>.

We would also welcome the submission of cover suggestions, or motifs to be used by our Graphics Illustrator in designing a

cover.

I look forward to seeing a revised form of your manuscript when it is ready.

Yours sincerely,

Referee #1:

In this manuscript, Rasool et al. address the role of the plant homeo domain zinc finger protein PHF6 in neural progenitor cell proliferation and brain development. A tamoxifen-inducible cKO mouse model is used to ablate PHF6 in neural progenitor cells of the developing brain. In parallel, knockdown and overexpression of PHF6 in vitro and two mouse lines harboring point mutations in PHF6 found in patients with Börjeson-Forssman-Lehmann Syndrome are used. The different models suggest a role of PHF6 in regulating neural progenitor cell proliferation. To identify the molecular mechanism, the authors perform ChIP-seq and RNA-seq analysis. Based on this, they propose that PHF6 regulates the expression of a set of Ephrin receptor genes, contributing to the neural progenitor cell self-renewal phenotype.

Given that PHF6 is mutated in a human X-linked intellectual disability (XLID) condition, understanding the underlying cellular and molecular mechanisms is important. Yet, in the current form, the manuscript has several significant limitations that compromise the interpretation of the data and the conclusions put forward.

1) The Phf6^{-/-} KO model and the respective controls are insufficiently described and characterized. In the methods, it is stated that the KO is induced by a dose of 2 mg tamoxifen at E14.5 and the brains analyzed 24 h later. Yet, there are Phf6 KO mice at P0 and E13.5 (Figure 5). How were they generated? In Figure 5A/B, it looks like Phf6 is lost from all cell types in the tissue. How is that compatible with induction of deletion at E14.5? Moreover, previously, it has been described that a dose of 2 mg Tamoxifen in combination with the Nes::CreERT2 can induce a microcephaly phenotype (Forni, Ponzetto, J Neurosc 2006). Did the authors check for that? What was the efficiency of KO induction? What are the control mice used? The designation of "Phf6^{+/+}/Nestin-CreERT2 mice (Phf6^{Loxp/Loxp})" is confusing, as either the mice are Phf6^{+/+} or Loxp/Loxp. Which of the two is the case?

2) Throughout the paper it is unclear of where cells come from, which stages are used and how the specific deletion or knockdown is induced. One example is the RNA-seq analysis (Suppl Table 4). What was the source of the material? Moreover, it is highly problematic to use the entire brain for NSC proliferation assays as the brain consists of different structures and progenitor zones at E14.5.

3) The quality of the ChIP-seq data cannot be assessed as Input/IgG controls are lacking and no examples of ChIP-seq tracks are presented. Moreover, all data sets should be deposited in a repository and made accessible to reviewers and the scientific community. Quality controls and replicate information should be included in the supplementary material. Given that the authors have a KO strain, the specificity of the antibody in ChIP should be demonstrated. Example tracks of the (CA)_n repeat regions and Ephrin genes should be shown for Phf6 and corresponding controls to convince the reader that this is not an artifact of the repeat sequences.

4) The most reproducible phenotype across the different Phf6 models appears to be the neural progenitor proliferation effect. How is this compatible with previous work on the R342X mutant that has reported microcephaly and hydrocephaly (Ahmed, Picketts, Human Mol Genet 2021)? Where these brain phenotypes also observed in the Phf6 KO? What is known about this in the human disorder? Is the regulation of NPC proliferation by Phf6 restricted to central regions of the brain? If so, which regions are relevant to the human pathology?

5) The title is highly misleading and should be rephrased. To suggest a treatment for neurogenesis for a disorder that is diagnosed when neurogenesis is largely complete is misleading and unacceptable.

- 6) Figure 1e/f should be replaced with numerical and statistical data.
- 7) Is enrichment of microsatellites statistically significant?
- 8) Is the RNA-seq data in Figure 2a from neural progenitor cells (text) or neurons (figure legend)? How were the cells isolated?
- 9) The data for RNAP II recruitment to the TSS (line 132) should be shown.
- 10) The statement on neuron density (line 171) should be undermined by quantitative data of neuronal layer markers and discussed in the context of previous Phf6 mouse models.
- 11) From which brain region are the coronal sections in Figure 7? The images should show the entire region (e.g. cortical column, if that is the example). Only upon quantification of sufficient n and statistical analysis can conclusions be drawn.

Referee #2:

PHF6 is a transcriptional regulator whose mutation cause a X-linked intellectual disability (XLID). In this manuscript Rasool et al. describe the role of PHF6 in controlling the expression of several members of the Eph receptor (Eph R) family and in modulating neural progenitor cells self renewal during mouse brain development. The authors performed a wide array of analyses, using different gain and loss of function in vitro and in vivo models. First, they performed genome wide ChIP-seq analyses as well as transcriptomic analyses to identify direct target genes of PHF6 in the developing mouse brain. They show that PHF6 acts as a DNA binding protein that regulates gene expression. Through these analyses they identify Eph R as potential targets which they validate using western blot, RT-PCR, luciferase assays and ChIP-PCR. In addition, Rasool et al. show that PHF6 loss of function leads to an increased proliferation and self renewal of neural progenitor cells while restoring Eph R expression in PHF6 depleted cells decreases proliferation and self renewal. The authors conclude that Eph R may be therapeutic targets for XLID disorders with impaired neurogenesis.

Overall this is an interesting study which contributes to a better understanding of the role of PHF6 in the developing brain and how it may contribute to XLID. However, there are a number of weaknesses in the manuscript that lessens confidence in the study.

Main comments

- 1) Different types of experiments were performed at different stages (E14, E17, P0) and/or on different models (PHF6 knock down, PHF6 lox/Cre KO, PHF6 R342X and C99F mutants) which renders comparisons between datasets difficult. For instance, ChIP seq was performed at E17 on cortex tissue while RNA-Seq was performed on E14.5 neurospheres. The cell composition of E17 cortex and E14.5 neurospheres are very different, it is thus difficult to conclude on direct targets from these experiments since changes in gene expression could reflect changes in cell composition. Also, on the RNA-Seq data, were Sox2 and Nestin identified as DEG ?
- 2) The data is presented in a somewhat odd order, with some of the control experiments presented late. For instance, IF data on PHF6^{-/-} is presented in figure 3a, but the validation of the line (loss of PHF6 expression) is presented in figure 5.
- 3) The study reports 2 sets of data that contradict already published ones yet the authors do not comment or even cite some of the previously published results.
 - a. On Figure 3j, k the authors conclude that PHF6^{-/-} brains have reduced neuron numbers yet in (Ahmed et al. Hum Mol Genet 2021), the authors showed that neuron numbers were unchanged in PHF6 R342X mutants. To be convincing, this data has to be supported by IF staining for neuronal markers. In fact it would be important to back up the in vitro data on progenitors with in vivo data. The expectation is that PAX6⁺ or SOX2⁺ progenitors should be increased in the PHF6^{-/-} developing neocortex.
 - b. On figure 6a, the authors show that overexpression of EphA4 in neural progenitors leads to a decrease in stem cell frequency yet in (North et al. Development 2009), the author showed the opposite. Also, in figure 6n-o, a control condition (WT cells) is needed to conclude that re-expressing EphA4 rescues the mutation in PHF6.
- 4) In a similar vein, other ChIP-seq data and RNA seq data have been published for PHF6 but are not discussed.
- 5) In the introduction and discussion, in the paragraph on neurogenesis and EphR, the authors refer only to publications on adult neurogenesis, while their study deals with embryonic neurogenesis. They should use appropriate references, closest to their model.
- 6) Detailed information on statistical analyses, quantification methods and replicates are lacking. Description of the siRNA procedure should be provided.
- 7) Labeling of some figure panels is misleading, for instance PHF6^{-/-} is not correct if only male embryos/animals were used (it should be PHF6Y^{-/-}). Exact genotype should be provided for all samples on each figure panel.
- 8) The start of the Discussion on PHF6 DNA binding is odd since this is not the main point of the study. In fact the entire manuscript reads like two (or three) different studies have been collated into a single manuscript that is not completely fluid.
- 9) The text should be carefully edited, some sentences are incomprehensible because of short cuts. For instance lines 55 to 58.

Also, some of the references are not formatted properly.

10) The title (and discussion) reads like an overstatement on Eph R since their role on neurogenesis was already known and their therapeutic potential is not strongly supported from the data presented here. Also, « mapping » does not mean anything in the context of the title.

Referee #3:

The main scientific question of this article is whether and how PHF6 regulates early cortical development in mammal brain, which is essentially answered via investigations into the molecular mechanisms and biological functions of PHF6. The authors found that PHF6 was crucial to regulate embryonic neural stem cells (eNSCs) in terms of their self-renewal and cell fate specifications, which was impaired in Börjeson-Forsman-Lehmann syndrome (BFLS). Main findings of this manuscript include 1) PHF6 can bind with DNA directly as an activator or repressor; 2) PHF6 is crucial for eNSC proliferation and differentiation in brain development; 3) PHF6 regulates eNSC functions through downstream EphA proteins. The authors designed precise experiments to show manipulating EphR, particularly EphA4, was sufficient to rescue the abnormal neural development induced by PHF6 dysfunction. These results highlight the necessity of EphR in PHF6 neurogenic regulatory pathway. However, the authors did not attach detailed information of the RNA-seq experiments that guided them to focus on EphR research.

Overall, the design of experiments is clear and meet the basic requirements of molecular biology community, especially the screening of PHF6 targets and the discovery of downstream Ephrin receptors (EphRs). However, the manuscript lacks valid data interpretation and proper writing. Some evidence is not solid enough to support all the theories, and more subtle descriptions of key experiments and data are needed. The suggestions in detail are as follows:

1. Line 76: "deregulation" is supposed to describe the entire loss of function for a gene, whereas Phf6 mutations may lead to multiple types of regulatory disruptions. I suggest using "dysregulation" to include more complete possibilities.
2. Line 80-85: EphRs are well studied proteins in the field of neural development. Since the experiments are established on embryonic NSCs, a brief introduction how EphRs, especially EphA family, regulate prenatal neural development is recommended instead of your current summary of their roles in adult brains.
3. Line 98-100: According to the introduction, PHF6 is likely to function at early stages of corticogenesis, so I am confused why the study chose E17-18 embryos, a relatively late stage, for ChIP-seq analysis. Could the authors explain the reasons of the selection?
4. Line 111-114: It is a bit difficult to recognize the words "nervous system development", "neurogenesis" and "neuron differentiation" in word-cloud plots. Could the authors use more distinguishable plots such as dotplots, gseaplots or emaplots to show the enrichment of GO terms?
5. Line 121-122: The authors seem to forget indicating the material used for mRNA-seq, which is also not mentioned in the method part. I guess the authors meant to conduct RNA-seq of the cell cultures 5 days after isolation. Please add the detailed information of samples.
6. Line 127-130: Similarly, the authors did not conspicuously indicate the method used to identify Pol II occupancy.
7. Line 150-152: The authors do not explain what materials are used for the immunoblotting assay. Is it cultured eNSCs at a certain stage or spheres formed by eNSCs?
8. Line 153: The word "3D cultures" is confusing. Is it a star method applied in the research? If so, please introduce the characteristics of the "3D cultures" at the beginning of this paragraph.
9. Line 187: I noticed the authors regarded Nestin and Sox2 as signs of eNSC stemness more than once. As Nestin could be expressed in early astrocytes while Sox2 is more expressed from activated neuron precursors than neural stem cells, it is controversial if the authors claim the total mRNA expression of these two genes in cell cultures completely reflects the stemness of eNSCs. Could the authors use markers like Hopx or Ascl1 to label the stemness instead?
10. Line 188-190: Figures 3l-m are not great ways to represent the significant difference between the mutant and control data, especially Figure 3l. Bar plots comprised of both mutant and control groups with a significant star label on the top could help emphasize the significant increase. Also, please attach the method applied to calculate the significance in the legend.
11. Line 199-202: The authors do not indicate any detailed information in RNA-seq analysis. Could the authors add the genotypes, the number, the age and the brain region of mice sampled for RNA extraction? And it is more helpful if the authors could explain the statistical processes to screen PHF6 downstream targets and the criteria leading the authors to concentrate on EphR family.
12. Line 208-210: Some EphRs, such as EphA7 and EphB1, exhibits stage-enriched expression patterns. Thus, it is not proper to claim all EphR genes are expressed from E10 to P4. Since many cell clusters mixed up and overlap on each other, the presentation of Figure 4a and 4g is also hard to understand.
13. Line 213-216: It is confusing that the authors used the unit "post-conceptual weeks (pcw)" for the x-axis of the supplementary Figure 2. Mouse pups are usually born within 3-4 pcw, and the unit "pcw" is generally used for human fetal samples. Did the authors write a wrong legend here?
14. Line 254: Could the authors explain why P0 R342X mice were used? What is the importance of EphR and PHF6 at this developmental stage?
15. Line 283: The x-axis of figure 6g lacks a title. Alternatively in Figure 6g, the blue straight line representing the pLVX. GFP

- group seems not to fit the blue scatters well. Could the authors double-check the plotting or explain why the control group in 6g displays increased self-renewal compared to the controls in 6a-d?
16. Line 288: According to Figure 5d and 5f, WT eNSCs express a baseline endogenous EphA4 naturally; however, WT eNSCs in Figure 6i seem not to express any endogenous EphA4 protein. Could the authors explain any reasons of such inconsistency?
17. Line 290: The authors should use the full name "stem cell frequency" instead of solely an abbreviation "SCF" when it firstly appears in the article.
18. Line 291-293: Could the authors add an interpretation or conclusion of Figure 5k-l? Also, could the authors explain the deficiency of EphA4 protein in the control group of Figure 5k?
19. Line 294-296: I suggest to add a non-mutated WT group on the basis of current rescue experiments to examine the power of both rescue manipulations.
20. Line 303-305: If the authors tried to investigate the alterations of eNSC fate specification in PHF6 mutated mice, defining the stemness of eNSCs via SOX2 and NESTIN protein level is not enough to represent a wide insight into cell fates. Glial markers (e.g. GFAP, OLIG2, etc.) as well as immature neuron markers (e.g. DCX, STMN1/2, etc.) are required together with stemness markers to show the change of eNSC fate determinations.
21. Line 306-312: Could the authors also count the percentage of SOX2+/TBR2+ merged cells and put that data as a bar in Figure 7c-d?

Point by point response

EMBOR-2023-57640-T

We would like to thank the reviewers for their valuable time to review this manuscript, for the positive comments as well as insightful suggestions. We have addressed each point by providing the required information on both methods and statistical analysis. We also provide new data in the revised manuscript to address the remaining reviewers' comments as outlined in this point-by-point-response.

Referee #1:

In this manuscript, Rasool et al. address the role of the plant homeo domain zinc finger protein PHF6 in neural progenitor cell proliferation and brain development. A tamoxifen-inducible cKO mouse model is used to ablate PHF6 in neural progenitor cells of the developing brain. In parallel, knockdown and overexpression of PHF6 in vitro and two mouse lines harboring point mutations in PHF6 found in patients with Börjeson-Forssman-Lehmann Syndrome are used. The different models suggest a role of PHF6 in regulating neural progenitor cell proliferation. To identify the molecular mechanism, the authors perform ChIP-seq and RNA-seq analysis. Based on this, they propose that PHF6 regulates the expression of a set of Ephrin receptor genes, contributing to the neural progenitor cell self-renewal phenotype.

Given that PHF6 is mutated in a human X-linked intellectual disability (XLID) condition, understanding the underlying cellular and molecular mechanisms is important. Yet, in the current form, the manuscript has several significant limitations that compromise the interpretation of the data and the conclusions put forward.

1) The *Phf6*^{-/-} KO model and the respective controls are insufficiently described and characterized. In the methods, it is stated that the KO is induced by a dose of 2 mg tamoxifen at E14.5 and the brains analyzed 24 h later. Yet, there are *Phf6* KO mice at P0 and E13.5 (Figure 5). How were they generated? In Figure 5A/B, it looks like *Phf6* is lost from all cell types in the tissue. How is that compatible with induction of deletion at E14.5? Moreover, previously, it has been described that a dose of 2 mg Tamoxifen in combination with the *Nes::CreERT2* can induce a microcephaly phenotype (Forni, Ponzetto, J Neurosc 2006). Did the authors check for that? What was the efficiency of KO induction? What are the control mice used? The designation of "*Phf6*^{+/+}/*Nestin-CreERT2* mice (*Phf6*^{Loxp/Loxp})" is confusing, as either the mice are *Phf6*^{+/+} or *Loxp/Loxp*. Which of the two is the case?

We agree with the reviewers that the methodology for *Phf6* knockout models were insufficiently described in the previous version of the manuscript. In the revised manuscript, we have added details to provide clarifications in the methods section (**page 19-21 of revised manuscript**). Briefly, we employed two different methods in our study to generate: a) *Phf6* knockout mice using *Nestin-Cre*, and b) inducible *Phf6* knockout mice using the *Nestin-CreERT2* strain. For clarification, in the revised manuscript, all the experiments conducted with *Nestin-Cre* model is moved to the supplementary materials and all the results with inducible *Nestin-CreERT2* model are included in the main figures. Furthermore, we have replaced all the abbreviation with full names on each panel. With respect to the question on Fig 5a/b (**Fig EV3 in the revised manuscript**), we would like to clarify that the *Nestin-Cre* knockout model was employed wherein the *Phf6* gene was deleted from the mouse central and peripheral nervous system from E11.5, which is the onset of *Nestin* gene expression¹. Thus, *Phf6* is excised from any cells that

have expressed Nestin at some point during their lifespan including neuro-progenitors in the SVZ prior to their differentiation and migration to their final position within the cortex.

As for the tamoxifen inducible model, unlike (Forni, Ponzetto, J Neurosc 2006), we did not observe a microcephaly phenotype. This could be explained due to different approaches used in each study, mainly time and duration of induction: a) we performed a single administration of tamoxifen (TMX) at ~E14, at which point the major brain structures and the brain's basic organization is complete, with exception of ongoing growth and refinement of neural connections, b) at the time points employed herein, upper layers of the cortex are already formed, c) we collected the brains after 24-48 hours post-TMX, as opposed to the mentioned publication by Forni, Ponzetto, J Neurosc 2006² where they administered TMX at E10.5 (the time at which the neural tube is in the process of closing) and the embryos were not harvested by the researchers until E19.5 (9 days later).

The efficiency of KO induction in the Nestin-CreERT2 strain is ~80-90% as revealed by gene expression (previous Fig 5c, **updated Fig 5A**), and protein expression analysis (previous Fig 5d, **updated Fig 5B**).

Regarding nomenclature, we now clarify the strains as *Phf6*^{-Y} / *Nestin-Cre*⁺ or *Phf6*^{-Y} / *Nestin-CreERT2*⁺ for KO male mice, and *Phf6*^{loxP/Y} / *Nestin-Cre*⁻ or *Phf6*^{loxP/Y} / *Nestin-CreERT2*⁻ for male WT controls.

2) Throughout the paper it is unclear of where cells come from, which stages are used and how the specific deletion or knockdown is induced. One example is the RNA-seq analysis (Suppl Table 4). What was the source of the material? Moreover, it is highly problematic to use the entire brain for NSC proliferation assays as the brain consists of different structures and progenitor zones at E14.5.

We would like to clarify that we did not use the entire brain for NSC proliferation assays. We isolated and established a pure homogenous eNSC culture as per established protocols³⁻⁵ prior to conducting the assay. The eNSCs were isolated and cultured from ~E14 and cultured in NSC media (containing B27 supplement, β FGF, and mEGF) that allows for growth of stem cells (Published in Burbani & Jahani-Asl, 2022)⁵. After 4-7 days post-culture, the cells were split and passaged to maintain the pure stem cell population and discard any debris. These eNSCs were then used for downstream assays.

As for RNA-seq data, cortical progenitors were also established from the E14 cortex and were maintained in culture for 5 days to expand the homogeneous population prior to submitting samples for mRNA-Seq. This detail has been added to the Methods section (**page 29 in the revised manuscript**).

3) The quality of the ChIP-seq data cannot be assessed as Input/IgG controls are lacking and no examples of ChIP-seq tracks are presented. Moreover, all data sets should be deposited in a repository and made accessible to reviewers and the scientific community. Quality controls and replicate information should be included in the supplementary material. Given that the authors have a KO strain, the specificity of the antibody in ChIP should be demonstrated. Example tracks

of the (CA)_n repeat regions and Ephrin genes should be shown for Phf6 and corresponding controls to convince the reader that this is not an artifact of the repeat sequences.

As requested by the reviewer we provide new data to show the specificity of the antibody using *Phf6* KO as an additional control (**New Fig EV5I-J**).

Additionally, we now provide cross-correlation metrics for our ChIP-seq data, showing high signal enrichment at specific peaks (**New Fig EV1A**). We have also included example ChIP-seq tracks for PHF6 pull-down as well as IgG control for several (CA)_n repeat regions and *EphR* genes of interest (**New Fig EV1B**). We have also deposited all the raw data from Figures 1 and 2 in the GEO (series accession number for both datasets is **GSE247838**) (**page 30 in the revised manuscript**).

4) The most reproducible phenotype across the different *Phf6* models appears to be the neural progenitor proliferation effect. How is this compatible with previous work on the R342X mutant that has reported microcephaly and hydrocephaly (Ahmed, Picketts, Human Mol Genet 2021)? Where these brain phenotypes also observed in the *Phf6* KO? What is known about this in the human disorder? Is there regulation of NPC proliferation by Phf6 restricted to certain regions of the brain? If so, which regions are relevant to the human pathology?

The main focus of our study is the analysis of NSCs in the brain of developing embryos which is consistently deregulated in the *Phf6* conditional KO mouse models and different BFLS models. The underlying mechanisms of hydrocephaly and microcephaly and percentage occurrence of these phenotypes in Ahmed et al 2021 remained unexplored and a subject for future investigation. One potential model to explain such phenotypes is that they could result from *Phf6* point mutations in other cell types (i.e., mutation in ependymal cells, or cells that express CSF). A second alternative explanation is that eNSC deregulation and impact on neural density in part may contribute to such phenotype. Our analysis of *Phf6*^Y/*Nestin-Cre*⁺ or in the *Phf6*^Y/*Nestin-CreERT2*⁺ mice revealed no hydrocephaly or microcephaly, supporting model 1 since the R342X mice was generated using the mouse embryonic stem cells, so all cells in the mouse will end up having the mutation. The *Phf6* KO mice, however, only have deletion of *Phf6* in *Nestin* expressing cells.

As requested by the reviewer, we analyzed *Phf6*^Y/*Nestin-Cre*⁺ mouse brains and observed no brain abnormalities at birth. *Phf6*^Y/*Nestin-Cre*⁺ mice appeared normal at birth with average body weight and brain size (panel a-b). However, we found that *Phf6*^Y/*Nestin-Cre*⁺ male mice showed modest reduced body weight at P10 and P35 compared to *Phf6*^{loxP/Y}/*Nestin-Cre*⁻ control littermates (panel b).

Figure for referee with unpublished data and its description has been removed upon request by the authors.

Finally, PHF6's regulation of neural progenitor cell proliferation is restricted to certain brain regions that houses neural progenitor cells (mainly ventricular zone)⁶⁻¹⁰. Stem cells from the SVZ are destined to become mature neurons of the cortical layers and we have previously shown a PHF6 impact on cortical neuron development (Zhang et al 2013)¹¹. We have also shown that abnormalities in the cortical structures leads to neuronal hyperexcitability (Cheng et al 2018)¹², which can be attributed to the seizure phenotype seen in patients. Although there is no data available characterizing the developing brain of human BFLS patients, alterations in fate specification of eNSCs within the SVZ is expected to be the underlying cause of XLID syndromes, with defects in developing cortex, as is the case in other forms of intellectual disability¹³⁻¹⁶.

5) The title is highly misleading and should be rephrased. To suggest a treatment for neurogenesis for a disorder that is diagnosed when neurogenesis is largely complete is misleading and unacceptable.

Promoting adult neurogenesis in SVZ and SGZ areas, where quiescent NSCs reside, has great potential in regenerative medicine. Our data suggests that EphR may serve as a promising target to expand the NSC pool. However, as requested by the reviewer, we have revised the title to 'Transcriptional reprogramming of neural stem cell by a PHF6/EphR signalling pathway'.

6) Figure 1e/f should be replaced with numerical and statistical data.

We have now included Supplementary Tables containing detailed numerical and statistical data for the GO terms summarized in Fig 1E/F (**New Table EV5**). Fig 1g provides an additional numerical representation of the same data that underlie Fig 1E/F.

7) Is enrichment of microsatellites statistically significant?

Among the top 1000 PHF6 peaks, 609 overlap a (CA)_n repeat on either DNA strand. In comparison, we observed an overlap of only 67 between (CA)_n repeats and shuffled peak coordinates (Fisher's exact test $P < 2.2e-16$). These results are now described in the manuscript (**page 5**). The **New Table EV2** also includes the information about different repeat elements that overlap the top 1000 PHF6 peaks.

8) Is the RNA-seq data in Figure 2a from neural progenitor cells (text) or neurons (figure legend)? How were the cells isolated?

We apologize for the typo. This is corrected to neural progenitors, as we described in our response to point #2.

9) The data for RNAP II recruitment to the TSS (line 132) should be shown.

We thank the reviewer for pointing out this omission. The Pol II occupancy data was obtained from GEO (accession number GSM2442441)¹⁷. The bedGraph file representing Pol II occupancy was directly downloaded from GEO, converted to bigWig, and overlaid on gene TSS coordinates using bwtool¹⁸.

This information is now included in the revised manuscript (**page 29**).

10) The statement on neuron density (line 171) should be undermined by quantitative data of neuronal layer markers and discussed in the context of previous *Phf6* mouse models.

As requested by the reviewer, we provide quantitative data with cortical layer markers (**New Fig EV3E-G**). we conducted additional analysis using cortical layer markers at P0 of the *Phf6^{loxP/Y} / Nestin-Cre⁻* and *Phf6^{-Y} / Nestin-Cre⁺* brains (**New Fig EV3E-G**). We utilized three layer-markers to co-label SATB2+ neurons in layer II-VI, CTIP2+ neurons in layer V, and TBR1+ neurons in layer VI to assess the role of PHF6 in maintaining the proper neuronal lamination in corticogenesis. Quantification of these data revealed that although, *Phf6* deletion in the CNS did not significantly affect the number of SATB2+, CTIP2+ and TBR2+ neurons, a shift of SATB2+/neurons away from the top of cerebral cortex plate was observed in *Phf6^{-Y} / Nestin-Cre⁺* mice. To quantify the migration patterns of SATB2+ and neurons in the cerebral cortex influenced by loss of *Phf6*, a grid consisting of 10 equivalent bins was applied to the image of P0 cerebral cortex to equally divide the cortical wall spanning from the basal of ventricle zone to the pial surface into ten bins. The ten bins were marked sequentially from the top to the bottom, namely, bin 1 covers the most superficial (i.e., outer) layer and bin 10 covers the deepest layer. Neurons within each bin were counted and a significant decline in SATB2+ neurons in bin 1 of the cerebral cortex was observed in *Phf6^{-Y} / Nestin-Cre⁺* mice (**New Fig EV3E-G**) suggesting impairment in the ability of SATB2+ neurons to migrate to superficial layers of the developing cerebral cortex in *Phf6* KO mice. This finding is consistent with the attenuation of neuron density (**New Fig EV3C**) and suggests that PHF6 is involved in regulating the process of radial neuronal migration during the establishment of cortical lamination.

11) From which brain region are the coronal sections in Figure 7? The images should show the entire region (e.g. cortical column, if that is the example). Only upon quantification of sufficient n and statistical analysis can conclusions be drawn.

The images that were quantified for Fig 7 are taken from the walls of the ventricular zone and were repeated with n=3 (biological replicates) with a two-tailed unpaired student t-test applied for analyzing significance. As requested by the reviewer, we have included the VZ in the revised manuscript (**Fig 7F**).

Referee #2:

PHF6 is a transcriptional regulator whose mutation cause a X-linked intellectual disability (XLID). In this manuscript Rasool et al. describe the role of PHF6 in controlling the expression of several members of the Eph receptor (Eph R) family and in modulating neural progenitor cells self renewal during mouse brain development. The authors performed a wide array of analyses, using different gain and loss of function in vitro and in vivo models. First, they performed genome wide ChIP-seq analyses as well as transcriptomic analyses to identify direct target genes of PHF6 in the developing mouse brain. They show that PHF6 acts as a DNA binding protein that regulates gene expression. Through these analyses they identify Eph R as potential targets which they validate using western blot, RT-PCR, luciferase assays and ChIP-PCR. In addition, Rasool et al. show that PHF6 loss of function leads to an increased proliferation and self renewal

of neural progenitor cells while restoring Eph R expression in PHF6 depleted cells decreases proliferation and self renewal. The authors conclude that Eph R may be therapeutic targets for XLID disorders with impaired neurogenesis.

Overall this is an interesting study which contributes to a better understanding of the role of PHF6 in the developing brain and how it may contribute to XLID. However, there are a number of weaknesses in the manuscript that lessens confidence in the study.

Main comments

1) Different types of experiments were performed at different stages (E14, E17, P0) and/or on different models (PHF6 knock down, PHF6 lox/Cre KO, PHF6 R342X and C99F mutants) which renders comparisons between datasets difficult. For instance, ChIP seq was performed at E17 on cortex tissue while RNA-Seq was performed on E14.5 neurospheres. The cell composition of E17 cortex and E14.5 neurospheres are very different, it is thus difficult to conclude on direct targets from these experiments since changes in gene expression could reflect changes in cell composition.

Also, on the RNA-Seq data, were Sox2 and Nestin identified as DEG ?

We have previously established that PHF6 is highly expressed in the developing brain and its expression is significantly attenuated at birth \sim P2-4^{12,19}. Therefore, the timepoints examined between E14 and P0 are carefully selected to establish the relationship between PHF6 and EphR throughout such timelines in different mouse models. Furthermore, analysis of scRNA-Seq data reveals the strong correlation of PHF6 and EphR in the developing embryo at these timepoints (**Fig 4**). With respect to ChIP-Seq and RNA-Seq data, the cortical progenitors were isolated at \sim E14 and maintained in culture for 5 days to match the ChIP timing of the developing cortex (E17-E19). ChIP-Seq and RNA-Seq are used as candidate screens in our study and all data is validated multiple times with various models using low throughput assays. We would also like to mention that the bulk RNA-seq data was among the very first screens that we conducted for this study, long before we had generated and had access to any mouse models. This experiment was conducted using *Phf6*- siRNA in wild type mouse cortical progenitors, similar to the ChIP assay timing in wild type mice. However, we did observe a trend in the downregulation of SOX2 at 10% and the stem cell marker ALDH1A1 at 60%, although due to high variability among the three biological replicates, they did not reach statistical significance. We therefore applied low throughput assays to validate several stem cell markers (**Fig 3C,K-L, 7A-B, EV3H**).

2) The data is presented in a somewhat odd order, with some of the control experiments presented late. For instance, IF data on PHF6^{-/-} is presented in figure 3a, but the validation of the line (loss of PHF6 expression) is presented in figure 5.

We would like to clarify the order of experiments: We conducted our knockdown experiments using an siRNA approach in pilot studies prior to generating the mutant and KO mice. Fig 3A represents the change in eNSC spheres upon siRNA knockdown of *Phf6* in WT eNSCs. Following completion of this dataset we then generated and characterized the *Phf6*^{-Y} / *Nestin-Cre*⁺ KO mice, and the BFLS mice. In summary, Figure 3 is focused on eNSC self-renewal and proliferation, whereas Figure 5 is on analysis of PHF6 and EphR expression and establishes the binding of PHF6 to *EphR* gene regulatory region. We have now included validation of PHF6 expression via immunoblotting for Fig 3 (**New Fig 3D**).

3) The study reports 2 sets of data that contradict already published ones yet the authors do not comment or even cite some of the previously published results.

a. On Figure 3j, k the authors conclude that PHF6^{-/-} brains have reduced neuron numbers yet in (Ahmed et al. Hum Mol Genet 2021), the authors showed that neuron numbers were unchanged in PHF6 R342X mutants. To be convincing, this data has to be supported by IF staining for neuronal markers. In fact it would be important to back up the *in vitro* data on progenitors with *in vivo* data. The expectation is that PAX6⁺ or SOX2⁺ progenitors should be increased in the PHF6^{-/-} developing neocortex.

b. On figure 6a, the authors show that overexpression of EphA4 in neural progenitors leads to a decrease in stem cell frequency yet in (North et al. Development 2009), the author showed the opposite. Also, in figure 6n-o, a control condition (WT cells) is needed to conclude that re-expressing EphA4 rescues the mutation in PHF6.

As requested by the reviewer, we provide IF staining for neuronal markers (**New Fig EV3E-G**). We find significant decline in SATB2⁺ neurons which supports our finding on neuronal density (please also see our response to question #10 from reviewer 1). We also provide *in vivo* data that show the same trend for SOX2⁺ cells in the *Phf6* KO developing neocortex (**New Fig EV3D**)

With regards to the previous publication by North et al. 2009²⁰, the authors report that overexpression of *EphA4* in neural progenitor cells in the cortex was shown to cause a decrease in stem cell frequency²⁰, and specifically show that this is through ephrin-B1-initiated forward signalling. Unlike our study, they did not focus on eNSCs in the SVZ. Another more recently published article shows *EphA4* KD increased proliferation of hippocampal precursor cells (Zhao et al., 2019)²¹. These differences can be due to cell/region specific differences, and the specific ligand-receptor interactions. This can be potentially explained by mode of EphR, having varying expression levels and functions in a cell-type specific manner, which also depends on the mode of signalling based on the availability of the corresponding ligand at specific developmental time points. In further support, it is becoming clear that EphRs are differentially expressed on various cell types within the brain and play different roles on stem cell proliferation or differentiation (Jing et al., 2012)²². To address the reviewers concern on the differences in results between our manuscript and that of North et al., 2009, we have now referenced this study in the discussion section of our revised manuscript. In summary, the focus of our study is fate specification of NSCs in the SVZ in BFLS models which remained unexplored prior to our study reported herein.

As requested by the reviewer, we include control condition (WT cells) in our **New Figure 6M-P**, to conclude that re-expressing EphA4 rescues the mutation in R342X.

4) In a similar vein, other ChIP-seq data and RNA seq data have been published for PHF6 but are not discussed.

To our knowledge, we are the first to conduct the studies in the developing cortex. However as requested by the reviewer, we reference other studies in the revised discussion.

5) In the introduction and discussion, in the paragraph on neurogenesis and EphR, the authors

refer only to publications on adult neurogenesis, while their study deals with embryonic neurogenesis. They should use appropriate references, closest to their model.

As requested by the reviewer, we have now revised both Introduction and Discussion and we have made it more focused towards eNSCs.

6) Detailed information on statistical analyses, quantification methods and replicates are lacking. Description of the siRNA procedure should be provided.

As requested by the reviewer, we have now included detailed information outlined on **page 27-28** (siRNA methods), and **page 29-30** (quantification and statistical analysis).

7) Labeling of some figure panels is misleading, for instance PHF6^{-/-} is not correct if only male embryos/animals were used (it should be PHF6Y^{-/-}). Exact genotype should be provided for all samples on each figure panel.

We have revised the Method section (see response to point 1 by reviewer 1, and **page 19-20** in the revised manuscript) and have re-labelled our main figures to denote *Phf6*^{-Y} / *Nestin-CreERT2*⁺ for KO male mice, and *Phf6*^{loxP/Y} / *Nestin-CreERT2* for WT controls. Supplementary figures that were generated with the Nestin-Cre strain are now labelled as *Phf6*^{-Y} / *Nestin-Cre*⁺ for KO male mice, and *Phf6*^{loxP/Y} / *Nestin-Cre*⁻ for WT controls (**New Fig EV3A-G**, previously Fig 3 j-k & Fig 5 a-b).

8) The start of the discussion on PHF6 DNA binding is odd since this is not the main point of the study. In fact, the entire manuscript reads like two (or three) different studies have been collated into a single manuscript that is not completely fluid.

As requested by the reviewer we have revised the discussion to make it more focused.

9) The text should be carefully edited, some sentences are incomprehensible because of short cuts. For instance, lines 55 to 58. Also, some of the references are not formatted properly.

As requested by the reviewer we have revised and edited the text and references and have revised formatting.

10) The title (and discussion) reads like an overstatement on EphR since their role on neurogenesis was already known and their therapeutic potential is not strongly supported from the data presented here. Also, « mapping » does not mean anything in the context of the title.

We are the first to show the significance of EphR in BFLS and as a direct transcriptional target of PHF6. Our data suggest that manipulation of EphA4 and EphA7 in altering NSC fate and expanding the stem cell pool has important implications in regenerative medicine, however, as requested by the reviewer, we have revised the title and discussion.

Referee #3:

The main scientific question of this article is whether and how PHF6 regulates early cortical development in mammal brain, which is essentially answered via investigations into the molecular mechanisms and biological functions of PHF6. The authors found that PHF6 was crucial to regulate embryonic neural stem cells (eNSCs) in terms of their self-renewal and cell fate specifications, which was impaired in Börjeson-Forssman-Lehmann syndrome (BFLS). Main findings of this manuscript include 1) PHF6 can bind with DNA directly as an activator or repressor; 2) PHF6 is crucial for eNSC proliferation and differentiation in brain development; 3) PHF6 regulates eNSC functions through downstream EphA proteins. The authors designed precise experiments to show manipulating EphR, particularly EphA4, was sufficient to rescue the abnormal neural development induced by PHF6 dysfunction. These results highlight the necessity of EphR in PHF6 neurogenic regulatory pathway. However, the authors did not attach detailed information of the RNA-seq experiments that guided them to focus on EphR research.

Overall, the design of experiments is clear and meet the basic requirements of molecular biology community, especially the screening of PHF6 targets and the discovery of downstream Ephrin receptors (EphRs). However, the manuscript lacks valid data interpretation and proper writing. Some evidence is not solid enough to support all the theories, and more subtle descriptions of key experiments and data are needed. The suggestions in detail are as follows:

1. Line 76: "deregulation" is supposed to describe the entire loss of function for a gene, whereas Phf6 mutations may lead to multiple types of regulatory disruptions. I suggest using "dysregulation" to include more complete possibilities.

As requested by the reviewer we have changed the terminology to dysregulation

2. Line 80-85: EphRs are well studied proteins in the field of neural development. Since the experiments are established on embryonic NSCs, a brief introduction how EphRs, especially EphA family, regulate prenatal neural development is recommended instead of your current summary of their roles in adult brains.

As requested, we have revised the section to include a summary of literature on eNSC as available in the literature (**page 4 in the revised manuscript**)

3. Line 98-100: According to the introduction, PHF6 is likely to function at early stages of corticogenesis, so I am confused why the study chose E17-18 embryos, a relatively late stage, for ChIP-seq analysis. Could the authors explain the reasons of the selection?

The peak expression of PHF6 is in developing brain from E12 to P2-4^{12,19}. E17-18 was selected as it is mid time point in expression.

4. Line 111-114: It is a bit difficult to recognize the words "nervous system development", "neurogenesis" and "neuron differentiation" in word-cloud plots. Could the authors use more distinguishable plots such as dotplots, gseaplots or emaplots to show the enrichment of GO terms?

The reason we chose word clouds is that there are many GO terms that are significantly enriched for PHF6 binding sites and/or (CA)_n repeats, which makes it difficult to visualize the significant terms as individual items. To make it possible for the readers to explore these GO terms, we have now added a **New Table EV5** containing the list of all significant hits and their associated statistics.

5. Line 121-122: The authors seem to forget indicating the material used for mRNA-seq, which is also not mentioned in the method part. I guess the authors meant to conduct RNA-seq of the cell cultures 5 days after isolation. Please add the detailed information of samples.

As requested by the reviewer, we have now added the details to the Methods (**page 29 of the revised manuscript**).

6. Line 127-130: Similarly, the authors did not conspicuously indicate the method used to identify Pol II occupancy.

We thank the reviewer for pointing out this omission. We have now added the description of the methods used to identify Pol II occupancy (**page 29**): Pol II occupancy data were obtained from GEO (accession number GSM2442441)¹⁷. The bedGraph file representing Pol II occupancy was directly downloaded from GEO, converted to bigWig, and overlaid on gene TSS coordinates using bwtool¹⁸.

7. Line 150-152: The authors do not explain what materials are used for the immunoblotting assay. Is it cultured eNSCs at a certain stage or spheres formed by eNSCs?

To address the reviewers comment we have clarified the materials used for each immunoblotting in the results text of the revised manuscript.

8. Line 153: The word "3D cultures" is confusing. Is it a star method applied in the research? If so, please introduce the characteristics of the "3D cultures" at the beginning of this paragraph.

The term 3D cultures here describe neurosphere cultures in suspension (free-floating), rather than adherent. We have revised the terminology to neurosphere cultures in the revised manuscript to avoid confusion.

9. Line 187: I noticed the authors regarded Nestin and Sox2 as signs of eNSC stemness more than once. As Nestin could be expressed in early astrocytes while Sox2 is more expressed from activated neuron precursors than neural stem cells, it is controversial if the authors claim the total mRNA expression of these two genes in cell cultures completely reflects the stemness of eNSCs. Could the authors use markers like Hopx or Ascl1 to label the stemness instead?

As requested by the reviewers we provide analysis of *Hopx* gene expression via RTqPCR (**New Fig EV3H**), and ASCL1 protein expression via Western blot in the E14 brains of R342X (**New Fig EV3I**).

10. Line 188-190: Figures 3l-m are not great ways to represent the significant difference between

the mutant and control data, especially Figure 3l. Bar plots comprised of both mutant and control groups with a significant star label on the top could help emphasize the significant increase. Also, please attach the method applied to calculate the significance in the legend.

We have now added the details with a significant star label on the top and have included the method applied to calculate the significance in the legend.

11. Line 199-202: The authors do not indicate any detailed information in RNA-seq analysis. Could the authors add the genotypes, the number, the age and the brain region of mice sampled for RNA extraction? And it is more helpful if the authors could explain the statistical processes to screen PHF6 downstream targets and the criteria leading the authors to concentrate on EphR family.

As requested by the reviewer, we have now added detail information on **pages 28-29**. In short, PHF6 ChIP-seq peaks were identified using MACS (version 1.4)^{23,24} with a permissive p-value threshold of 0.001, using "--nomodel" option. Fragment size was specified using "--shiftsize" argument, with the fragment length obtained by cross-correlation analysis using phantompeakqualtools²⁵. Peak-TSS distances were calculated using bedtools²⁶ only for peaks that passed p-value threshold of 10^{-5} , with TSS coordinates obtained from GENCODE²⁷ (release M9). Gene-level read counts were obtained by HTSeq²⁸, using gene annotations from GENCODE (release M9). Genes with a minimum of 150 reads in at least one sample were retained. Gene set analysis was then performed using ConsensusPathDB²⁹. The final list of downstream targets was narrowed down to focus on receptors and kinases in which the *EphR* genes were the top candidates in the developing brain.

12. Line 208-210: Some EphRs, such as EphA7 and EphB1, exhibits stage-enriched expression patterns. Thus, it is not proper to claim all EphR genes are expressed from E10 to P4. Since many cell clusters mixed up and overlap on each other, the presentation of Figure 4a and 4g is also hard to understand.

Although *EphA7/B1* may have stage-enriched expression that may be higher at some points in development than others, they are still detectable and generally expressed throughout embryonic development, and we show this via querying a public RNA-seq dataset (Di Bella et al., 2021)³⁰, exhibiting their expression across a number of cell types ranging from E10-P4. This dataset provides a robust representation of EphR gene expression dynamics during embryonic development. We have prepared additional visual representation of the data in Fig 4 (**New Fig EV4A-E**). Here, we can conclude that *Phf6* and *EphA4* are expressed in migrating neurons and intermediate progenitors across all time points (E10 to P4). Additionally, *EphA7/EphB1* show stage-enriched expression as they have higher expression from E18 to P4 (**Fig EV4A-E**).

13. Line 213-216: It is confusing that the authors used the unit "post-conceptual weeks (pcw)" for the x-axis of the supplementary Figure 2. Mouse pups are usually born within 3-4 pcw, and the unit "pcw" is generally used for human fetal samples. Did the authors write a wrong legend here?

We apologize for the error in the legend. This analysis used the bulk RNA-seq profiling from BrainSpan³¹, which collected postmortem human brain specimens to generate an atlas of the developing human brain. We have updated the legend to reflect that this data is from human origin.

14. Line 254: Could the authors explain why P0 R342X mice were used? What is the importance of EphR and PHF6 at this developmental stage?

PHF6 and EphR are co-expressed throughout the developing embryo up to birth at P2-4^{12,19} where their expression significantly declines. We used ~E14, E17, and P0 for our experiments to represent the range of developmental processes.

15. Line 283: The x-axis of figure 6g lacks a title. Alternatively in Figure 6g, the blue straight line representing the pLVX. GFP group seems not to fit the blue scatters well. Could the authors double-check the plotting or explain why the control group in 6g displays increased self-renewal compared to the controls in 6a-d?

We apologize for the typo and plotting error. We have now added a title for the Y axis and corrected the plotting error for the blue scatter line of the GFP group. The differences in self-renewal between the control groups in Fig 6G and Fig 6A-D can be attributed to the varying experimental design used for each assay. In Fig 6G, we employed a lentiviral plasmid, whereas in Fig 6A-D we employed electroporation of siRNA. This may contribute to cellular responses which could cause differences in self-renewal outcomes in the control groups.

16. Line 288: According to Figure 5d and 5f, WT eNSCs express a baseline endogenous EphA4 naturally; however, WT eNSCs in Figure 6i seem not to express any endogenous EphA4 protein. Could the authors explain any reasons of such inconsistency?

We would like to clarify that the EPHA4 protein expression in Figures 5 and 6 were examined across different sample types. Specifically, EPHA4 protein expression in *Phf6* KO samples was done on protein lysates from eNSCs, whereas in the R342X model the protein lysates were from the whole brain which may include multiple cell types expressing varying levels of EPHA4. In previous Fig 6i (**New Fig 6K**), we specifically assessed EPHA4 protein expression in WT eNSCs. Furthermore, we have updated this figure and provide a better blot for GFP bands in the control vector and the *Epha7* overexpressing cells (**New Fig 6L**). As such, we have updated the format of Fig 6I,K. Additionally, in Fig 4G we show EphA4 expression is lower in immature neurons than it is in more mature cell types.

17. Line 290: The authors should use the full name "stem cell frequency" instead of solely an abbreviation "SCF" when it firstly appears in the article.

As requested by the reviewer in the revised manuscript we have used the full name "Stem cell frequency".

18. Line 291-293: Could the authors add an interpretation or conclusion of Figure 5k-l? Also, could the authors explain the deficiency of EphA4 protein in the control group of Figure 5k?

This is probably referring to Fig 6K-L, as there is no panel K-L in Fig 5. Here, we are showing efficient electroporation of *EphA4* and *EphA7*-GFP tagged plasmids in eNSCs cultured from R342X mutant E14 embryos. As stated in our response to point 16 by reviewer 3, the deficiency of endogenous EPHA4 in the control samples could be due to the sample type, in this case eNSCs, which don't normally express high levels of EPHA4 as shown in cell-type expression correlation (Fig 4). Furthermore, we have generated a new blot for Fig 6L to better show the GFP bands in the control vector and the *EphA7* over expression plasmid more clearly than our previous blots, and we have updated the format of Fig 6k to match.

19. Line 294-296: I suggest to add a non-mutated WT group on the basis of current rescue experiments to examine the power of both rescue manipulations.

As requested by the reviewer, we have added the additional control and have revised the plot (**New Fig 6M-P**).

20. Line 303-305: If the authors tried to investigate the alterations of eNSC fate specification in PHF6 mutated mice, defining the stemness of eNSCs via SOX2 and NESTIN protein level is not enough to represent a wide insight into cell fates. Glial markers (e.g. GFAP, OLIG2, etc.) as well as immature neuron markers (e.g. DCX, STMN1/2, etc.) are required together with stemness markers to show the change of eNSC fate determinations.

As requested by the reviewer, we have provided data with these markers in the revised manuscript (**New Fig EV3I**).

21. Line 306-312: Could the authors also count the percentage of SOX2+/TBR2+ merged cells and put that data as a bar in Figure 7c-d?

As requested by the reviewers we have added the data (**New Fig 7E**).

References:

1. Tronche, F., *et al.* Disruption of the glucocorticoid receptor gene in the nervous system results in reduced anxiety. *Nature genetics* **23**, 99-103 (1999).
2. Forni, P.E., *et al.* High levels of Cre expression in neuronal progenitors cause defects in brain development leading to microencephaly and hydrocephaly. *Journal of Neuroscience* **26**, 9593-9602 (2006).
3. Nasser, M., *et al.* Transplantation of embryonic neural stem cells and differentiated cells in a controlled cortical impact (CCI) model of adult mouse somatosensory cortex. *Frontiers in Neurology* **9**, 895 (2018).
4. Azari, H., Rahman, M., Sharififar, S. & Reynolds, B.A. Isolation and expansion of the adult mouse neural stem cells using the neurosphere assay. *JoVE (Journal of Visualized Experiments)*, e2393 (2010).
5. Burban, A. & Jahani-Asl, A. Isolation of Mouse Embryonic Neural Stem Cells and Characterization of Neural Stem Markers by Flow Cytometry. in *Neuronal Cell Death: Methods and Protocols* 297-308 (Springer, 2022).

6. Ayoub, A.E., *et al.* Transcriptional programs in transient embryonic zones of the cerebral cortex defined by high-resolution mRNA sequencing. *Proceedings of the National Academy of Sciences* **108**, 14950-14955 (2011).
7. Bani-Yaghoub, M., *et al.* Role of Sox2 in the development of the mouse neocortex. *Developmental biology* **295**, 52-66 (2006).
8. Ferri, A.L., *et al.* Sox2 deficiency causes neurodegeneration and impaired neurogenesis in the adult mouse brain. (2004).
9. Hutton, S.R. & Pevny, L.H. SOX2 expression levels distinguish between neural progenitor populations of the developing dorsal telencephalon. *Developmental biology* **352**, 40-47 (2011).
10. Zhang, Z., *et al.* Histone methylations define neural stem/progenitor cell subtypes in the mouse subventricular zone. *Molecular neurobiology* **57**, 997-1008 (2020).
11. Zhang, C., *et al.* The X-linked intellectual disability protein PHF6 associates with the PAF1 complex and regulates neuronal migration in the mammalian brain. *Neuron* **78**, 986-993 (2013).
12. Cheng, C., *et al.* Characterization of a Mouse Model of Borjeson-Forssman-Lehmann Syndrome. *Cell Rep* **25**, 1404-1414.e1406 (2018).
13. Li, H., *et al.* Disruption of TCF4 regulatory networks leads to abnormal cortical development and mental disabilities. *Molecular psychiatry* **24**, 1235-1246 (2019).
14. Li, Y. & Zhao, X. Concise review: fragile X proteins in stem cell maintenance and differentiation. *Stem Cells* **32**, 1724-1733 (2014).
15. Moore, J.M., *et al.* Laf4/Aff3, a gene involved in intellectual disability, is required for cellular migration in the mouse cerebral cortex. *PLoS One* **9**, e105933 (2014).
16. Matsumoto, A., *et al.* LIN7A depletion disrupts cerebral cortex development, contributing to intellectual disability in 12q21-deletion syndrome. *PLoS One* **9**, e92695 (2014).
17. Liu, J., Wu, X., Zhang, H., Pfeifer, G.P. & Lu, Q. Dynamics of RNA polymerase II pausing and bivalent histone H3 methylation during neuronal differentiation in brain development. *Cell reports* **20**, 1307-1318 (2017).
18. Pohl, A. & Beato, M. bwtool: a tool for bigWig files. *Bioinformatics* **30**, 1618-1619 (2014).
19. Voss, A.K., *et al.* Protein and gene expression analysis of Phf6, the gene mutated in the Borjeson-Forssman-Lehmann Syndrome of intellectual disability and obesity. *Gene Expr Patterns* **7**, 858-871 (2007).
20. North, H.A., *et al.* Promotion of proliferation in the developing cerebral cortex by EphA4 forward signaling. *Development* **136**, 2467-2476 (2009).
21. Zhao, J., *et al.* EphA4 regulates hippocampal neural precursor proliferation in the adult mouse brain by d-serine modulation of N-Methyl-d-Aspartate receptor signaling. *Cerebral Cortex* **29**, 4381-4397 (2019).
22. Jing, X., *et al.* Ephrin-A1-mediated dopaminergic neurogenesis and angiogenesis in a rat model of Parkinson's disease. *PLoS one* **7**, e32019 (2012).
23. Feng, J., Liu, T., Qin, B., Zhang, Y. & Liu, X.S. Identifying ChIP-seq enrichment using MACS. *Nat Protoc* **7**, 1728-1740 (2012).
24. Zhang, Y., *et al.* Model-based analysis of ChIP-Seq (MACS). *Genome Biol* **9**, R137 (2008).

25. Landt, S.G., *et al.* CHIP-seq guidelines and practices of the ENCODE and modENCODE consortia. *Genome Res* **22**, 1813-1831 (2012).
26. Quinlan, A.R. & Hall, I.M. BEDTools: a flexible suite of utilities for comparing genomic features. *Bioinformatics* **26**, 841-842 (2010).
27. Frankish, A., *et al.* GENCODE reference annotation for the human and mouse genomes. *Nucleic Acids Res* **47**, D766-D773 (2019).
28. Anders, S., Pyl, P.T. & Huber, W. HTSeq--a Python framework to work with high-throughput sequencing data. *Bioinformatics* **31**, 166-169 (2015).
29. Kamburov, A., *et al.* ConsensusPathDB: toward a more complete picture of cell biology. *Nucleic Acids Res* **39**, D712-717 (2011).
30. Di Bella, D.J., *et al.* Author Correction: Molecular logic of cellular diversification in the mouse cerebral cortex. *Nature* **596**, E11 (2021).
31. BrainSpan. BrainSpan: Atlas of the Developing Human Brain. (2013).

Dear Dr. Jahani-Asl,

Thank you for the submission of your revised manuscript. We have now received the enclosed reports from the referees. All referees still have a few more minor suggestions that I would like you to address and incorporate before we can proceed with the official acceptance of your manuscript. Please co-submit a detailed point-by-point response to all last concerns with your final ms.

A few editorial requests will also need to be addressed:

- Please reduce the number of keywords to 5.
- Please correct the conflict of interest subheading to "Disclosure and Competing Interest Statement"
- Please resolve Ahmad Sharanak in ms file vs. Ahmad Sharanek in our online ms submission system.
- Please remove the author credits from the ms file. All credits need to be entered upon online ms submission.
- Please correct the EMBO reports reference format that can be found in EndNote: et al needs to be used after 10 author names, DOIs are only needed for preprints and datasets that have not been published yet.
- Table EV1-EV6 are all datasets and should be renamed and uploaded as such: Dataset EV1, etc. Table EV7 can be renamed to Table EV1. All EV Table legends need to be removed from the ms file, and the callouts in the ms file need to be updated accordingly.
- The synopsis image is created with BioRender and this needs to be acknowledged in the methods section (and only there) as follows:
"Graphics:
(some of the... OR Figure #... OR synopsis) Graphics were created with BioRender.com."
- The legends of the uploaded figures need to be removed from the figure files.
- The labels of the EV figure legends at the end of the ms file need to be corrected: "Figure EV1" instead of "Expanded View Figure 1"
- Our routine image analysis of accepted ms revealed possible reuses of images: 1. Figure 3D and Figure 5B GAPDH
2. Figure 7A and Figure EV 5B PHF6
The source data also seem to be the same. Please explain.
- Please note that a separate 'Data Information' section is required in the legends of figures 3; 5; 6; 7; EV2; EV3; EV5.
- Please indicate the statistical test used for data analysis in the legends of figures 2a, c
- Please note that information related to n is missing in the legends of figures 2a, c
- Please note that the data citation Data ref: (BrainSpan, 2013) refers to Allen Brain Atlas BrainSpan dataset and not experimental data. Please correct.
[Verify that data citations refer to deposited experimental data (not to journal articles or UniProt/Ensembl sequences etc.).
Callouts in text must be preceded by 'Data ref:'. Data references in the Reference list should have a [DATASET] tag and a URL link to the dataset. Verify that data citation callouts in text have matching entries in the Reference list and vice versa.]
- Some text on the synopsis image is too small at the final image size of 550 pixels x 383 pixels. The image is also a little too complex for this final size. Please send us a new, simplified image at the correct size on which all text is readable.
- EMBO press papers are accompanied online by A) a short (1-2 sentences) summary of the findings and their significance, B) 2-3 bullet points highlighting key results. Please send us this information along with the final manuscript.

Referee #1:

The revised manuscript provides greatly improved data on the role of the plant homeo domain zinc finger Phf6. In particular, the mouse models are now sufficiently described and the corresponding data clearly labeled. In addition, the authors have added supporting data and statistical analysis for several experiments, which corroborates their findings. Overall, the present manuscript has significantly improved and the findings are of great interest, especially since the authors have used a whole battery of different approaches and mouse models to demonstrate the role of Phf6 in regulating neural stem cell proliferation. Phf6 binding was investigated genome-wide, together with gene expression changes, leading to the identification of the EphR receptor family as important down-stream targets. There are some important minor comments that still remain to be addressed (see below). Upon addressing the remaining points, I recommend the manuscript for publication.

Additional points that remain to be addressed:

- 1) Abstract, "genome-wide binding of PHF6 ..., most of which overlap with (CA)_n-microsatellites". To my understanding, it is not the majority of peaks that overlap with microsatellites (Figure 1). If this is indeed the case, it should be shown in percentages in Figure 1.
- 2) While the data on neural stem cell proliferation is very strong, the claim on neurogenesis defects (abstract, line 333) and neuron density (lines 174, 305, 312) is not well supported. The Nissl stain is difficult to interpret. The more detailed analysis of total cells and neuronal subtypes in the cortical plate (figure 7 and EV3) do not support a neurogenesis defect nor reduced neuron numbers. If the authors want to make a claim on "density", they need to show corresponding convincing quantifications.
- 3) The datasets related to Phf6 genome-wide binding and transcriptional analysis should include both raw data and processed data files for immediate exploration and re-use by the community. GEO GSE247838 currently appears to only include raw data. In addition, bed and bigwig files should be made available.
- 4) Line 51, not clear why the presence of a NoLS suggests a role in transcription regulation?
- 5) Figure 1E should be replaced with a graph representing numerical data, including fold enrichment, number of genes and p value.
- 6) In figure 2, as fold change of differentially expressed genes, a cut-off of 2^{0.25} (less than 1.2-fold) was chosen, which is highly unconventional. Are such small changes even meaningful?
- 7) Similarly, an FDR of 0.02 is unconventional. And different FDRs appear to be used throughout figure 2?!
- 8) For the different types of analysis in figure 2, please provide the number of genes that change and highlight a few meaningful gene names in the volcano blots.
- 9) Line 202, "EphR receptors as top GO terms", I cannot find the corresponding data in the figure.
- 10) Figure 4, scRNA-seq data, please annotate the cell types in the blots. There are too many colors to match visually.
- 11) Lines 228 and 230, should also refer to Figure EV1.
- 12) Lines 318 and following, use appropriate nomenclature and replace "top" and "bottom" of cortex with "basal" and "apical".
- 13) In figures 7 and EV3, the axis of cell type quantification should be labeled more specifically. For example, what does "Percent of Sox2 cells" mean? Percent of Sox2 positive cells out of total cells in the entire cortical wall, or specific zones?

Referee #2:

In this revised manuscript the authors addressed most of my concerns. However, although the in vitro data is very convincing, I still have concerns about the in vivo data :

- 1) The authors added new images showing immunostaining for layer specific antibodies and immunostaining for Sox2 in Phf6 floxed mice on Fig EV3. While in the text they mention quantifications, these quantified data are not presented. Specifically, the authors wrote Lines 316-319 : « Quantification of these data revealed that although, Phf6 deletion in the CNS did not significantly affect the number of SATB2+, CTIP2+ and TBR2+ neurons, a shift of Hoechst+ neurons/SATB2+ neurons

towards the top of cerebral cortex plate was observed in Phf6-/- / Nestin-Cre+ mice.

2) The description of this data is odd, the authors refer to Hoechst+ neurons while Hoechst is a universal nuclear marker.

3) Lastly, the authors mention a shift of SATB2+ neurons towards the top of the cerebral cortex whereas line 325 they conclude to a significant decline in SATB2+ neurons in bin 1 (the most superficial).

Also, the title of figure 7 is misleading, Tbr2 is not a marker of newborn neurons, it is a marker of intermediate progenitors.

Lastly, in light of the data presented, the concluding statement (below) should be modified since no quantified data on a change in neuron numbers is provided:

lines 331-334 : Together, we report that PHF6 alters the mechanisms that regulate cell fate in the developing brain, and that loss-of-function of PHF6 in BFLS results in an imbalance in the number of uncommitted stem cells and newly born neurons which potentially may describe impaired circuit connectivity in BFLS patients.

Referee #3:

The current version of manuscript is much more accomplished than the original one. The authors solve the molecular mechanisms of how Phf6 mutants cause neurogenic defects in BFLS. The discovery of Phf6-EphA regulatory pathway connects a rare disease to a conventional neurogenic molecule, which largely accelerates the development of BFLS treatment. The illustration of evidence is clear and convincing. The article will be more readable if the authors could pay attention to the interpretation and demonstration of raw data. In summary, the manuscript has met the basic quality for publication and I have only minor points on the manuscript.

Line 123-126: The authors discussed two downregulated GO terms related to nervous system development as Fig 2B shows. However, Fig 2B also exhibits the upregulation of cation channel activities, which highly relates to the maturation of neurons. Could the authors describe these findings and raise a possible explanation? The interpretation is recommended to put in the section between Line 169-179, where the authors find the importance of Phf6 in neuronal differentiation.

Line 144: The context of this sentence is emphasizing the role of PHF6 in cell proliferation or self-renewal rather than cell fate decision. Could the authors correct the small mistake?

Line 198-201: The authors are supposed to describe the details of the RNA-seq analysis directly in the main text, e.g., the sample groups and detailed strategies of screening.

Line 274: Before the section of the rescue function of EphA4/7 in R342X mice, several rows of brief introduction about previous research on regulatory functions of EphA4/7 to neural stem cells in neural development, like what the authors mention in the Discussion section, are recommended. It helps emphasize the importance of EphA4/7 and inspire readers to understand the whole regulatory mechanism.

Line 305-312: The proportional alteration of Sox+ cells and Tbr+ cells in R342X model is well-described. However, any further interpretations are lacking so that it is hard for readers to understand the biological meaning of the changed cell proportion. A conclusion or demonstration of the phenomenon is required at the end of this part.

Line 369: It is not proper to use "dysregulation" to illustrate the normal biological function of PHF6.

Table EV6: It is not reader-friendly that the authors only provide the Ensembl ID of each gene. Please attach gene symbols in addition to Ensembl ID in each table.

Point-by-point response

Referee #1:

The revised manuscript provides greatly improved data on the role of the plant homeo domain zinc finger Phf6. In particular, the mouse models are now sufficiently described and the corresponding data clearly labeled. In addition, the authors have added supporting data and statistical analysis for several experiments, which corroborates their findings. Overall, the present manuscript has significantly improved and the findings are of great interest, especially since the authors have used a whole battery of different approaches and mouse models to demonstrate the role of Phf6 in regulating neural stem cell proliferation. Phf6 binding was investigated genome-wide, together with gene expression changes, leading to the identification of the EphR receptor family as important down-stream targets. There are some important minor comments that still remain to be addressed (see below). Upon addressing the remaining points, I recommend the manuscript for publication.

We would like to thank the reviewer for the invaluable time to review our revised manuscript, and stating that the findings are of great interest, and that the manuscript provides greatly improved data, recommending the manuscript for publication following addressing the remaining minor points. Below, we address the last set of comments.

Additional points that remain to be addressed:

1) Abstract, "genome-wide binding of PHF6 ..., most of which overlap with (CA)_n-microsatellites". To my understanding, it is not the majority of peaks that overlap with microsatellites (Figure 1). If this is indeed the case, it should be shown in percentages in Figure 1.

The statement in Abstract was describing de novo motif analysis that was run on top 1000 PHF6 peaks in which CA repeats were found in 609 peak summits on either DNA strand. We agree with the reviewer that this is not referring to the percent of the total peaks. For clarity, we have revised the Abstract and these results are now presented in more detail in the "Results" section instead, accompanied by Dataset EV2.

2) While the data on neural stem cell proliferation is very strong, the claim on neurogenesis defects (abstract, line 333) and neuron density (lines 174, 305, 312) is not well supported. The Nissl stain is difficult to interpret. The more detailed analysis of total cells and neuronal subtypes in the cortical plate (figure 7 and EV3) do not support a neurogenesis defect nor reduced neuron numbers. If the authors want to make a claim on "density", they need to show corresponding convincing quantifications.

As requested by the reviewer, we have revised the abstract and have made it more focused on stem cell phenotype. Having said that, we believe our results suggests a neurogenesis defect as neurogenesis is comprised of proliferation and fate specification of NSCs, migration of newborn neurons, and maturation of these neurons: a) Neural stem cells (NSCs) are the cell population at the apex of the neurogenic processes, and we show that they are consistently altered in different mouse models harbouring *Phf6* deletion and BFLS patient mutations. b) This is supported by a decrease in protein expression of mature cell-type markers including oligodendrocytes (OLIG2), astrocytes (GFAP), as well as progenitor cells (ASCL1) (Fig EV3I), in addition to the

Point-by-point response

quantification/imaging analysis of progenitors (TBR2+ cells in Fig 7) that we had provided. c) The figure EV3 on neuronal subtype quantification in the cortical plate (at defined P0) was initially provided in response to reviewers' concerns on how neural migration could be affected. These experiments show impairment in the ability of SATB2+ neurons to migrate to superficial layers of the developing cerebral cortex in *Phf6* KO mice. This finding is consistent with the attenuation of neuron density and suggests that PHF6 may be involved in regulating the process of radial neuronal migration during the establishment of cortical lamination. We agree though that characterization of NSCs has been a major component of this study and thus we have revised the Abstract as requested.

3) The datasets related to *Phf6* genome-wide binding and transcriptional analysis should include both raw data and processed data files for immediate exploration and re-use by the community. GEO GSE247838 currently appears to only include raw data. In addition, bed and bigwig files should be made available.

As requested by the reviewer, we have uploaded both raw data and processed data files, and bed and bigwig files. Of note, the bed files and count (tab) files were already available in GEO. To show the processed files and not only the compressed aggregate "GSE247838_RAW.tar" file, the reviewers should click on "(custom)" under "Download". The bed files are shown in the SubSeries expanded file list.

4) Line 51, not clear why the presence of a NoLS suggests a role in transcription regulation?

Previous literature supports a hypothesis in which PHF6 functions as a transcriptional regulator via its PHD domain, a process regulated by the nucleosome remodeling and deacetylation complex recruited to the genomic target site by the NoLS region of PHF6. As well, RNA-dependent localization of PHF6 has been proposed in rRNA transcription and enrichment of PHF6 across rDNA-coding sequence (Liu *et al*, 2014; Todd *et al*, 2016). However, in the revised manuscript, we have made the "Introduction" more concise and have included literature as related to the subject of our study.

5) Figure 1E should be replaced with a graph representing numerical data, including fold enrichment, number of genes and p value.

As requested by the reviewer, we have replaced Figure 1E. Our new graph (New Fig 1E) shows fold enrichment, number of genes, and p-value.

6) In figure 2, as fold change of differentially expressed genes, a cut-off of $2^{0.25}$ (less than 1.2-fold) was chosen, which is highly unconventional. Are such small changes even meaningful?

The rationale behind selecting a relatively moderate log fold-change cut-off is that, when many genes involved in the same or related biological processes show a coordinated change in expression, even relatively small changes can have significant cellular effects. In fact, the 1.2-fold change we have selected here (which corresponds to 20% increase or decrease in expression) is coincidentally the same as what was proposed by Subramanian *et al.*, (Subramanian *et al*, 2005), and Mootha *et al.*, (Mootha *et al*, 2003), in their papers describing

Point-by-point response

gene set enrichment analysis: “An increase of 20% in all genes encoding members of a metabolic pathway may dramatically alter the flux through the pathway...”. Similarly, we use this cut-off primarily to perform gene set analysis in downstream analyses described in the manuscript, in order to study coordinated pathway-level expression changes induced by PHF6 modulation.

7) Similarly, an FDR of 0.02 is unconventional. And different FDRs appear to be used throughout figure 2?!

While we understand that FDR cut-offs such as 0.05 are more conventional, we would like to point out that an FDR of 0.02 is more restrictive. Therefore, the results shown in Fig. 2 would still be identified as significant even if we opted to use a more relaxed, yet uniform cut-off, such as 0.05. We opted to use more restrictive FDR cut-offs to focus on a more restrictive set of candidates (hypotheses) with lower probability of false discovery.

8) For the different types of analysis in figure 2, please provide the number of genes that change and highlight a few meaningful gene names in the volcano blots.

As requested by the reviewer, we now provide revised figures 2A and 2C, showing the number of genes that have changed, and highlighting meaningful EphR gene names in 2A.

9) Line 202, "EphR receptors as top GO terms", I cannot find the corresponding data in the figure.

The GO term here refers to neurogenesis in Fig. 1F. Analyzing the candidate genes under the neurogenesis category highlighted EphRs as top candidates. For clarity, we have revised the text in lines 195-205 of the revised manuscript as follows:

“To identify downstream effectors of PHF6 function in the regulation of neurogenesis, we first analyzed the candidate target genes with their expression significantly deregulated based on the RNA-seq analysis with particular focus on druggable targets (e.g., Receptors, Kinases). These analyses revealed a host of candidate genes that could serve as PHF6 targets to regulate neurogenesis (Dataset EV6). We focused on members of the ephrin receptors (EphRs) family (*EphA4/7*, and *EphB1/2*) given that EphRs are the largest family of RTKs highly expressed in the developing brain (Barquilla & Pasquale, 2015; Darling & Lamb, 2019; Lisabeth *et al*, 2013). Importantly, EphRs have been shown to play different roles in regulating neuronal development (Aoki *et al*, 2004; del Valle *et al*, 2011; Stuckmann *et al*, 2001; Wilkinson, 2014). Prior to validation of EphRs as viable targets of PHF6 in the context of BFLS, we conducted additional gene expression analysis using public databases...”.

10) Figure 4, scRNA-seq data, please annotate the cell types in the blots. There are too many colors to match visually.

As requested by the reviewer, we have now added annotation to label the cell types (New Figure 4H).

11) Lines 228 and 230, should also refer to Figure EV1.

Point-by-point response

As requested by the reviewer, we have inserted Figure EV1 in the text (new lines 225-229) as follows:

“We next set out to investigate if the identified *EphRs* are direct PHF6 targets. Our ChIP-seq data revealed robust and significant binding of PHF6 to the promoter of *EphA4* with a p-value of $1.8E-08$ (**Dataset EV1, Fig EV1**). ChIP-seq data also revealed peaks associated with the TSS of *EphA7* and *EphB1* (**Dataset EV1, Fig EV1**) ...”.

12) Lines 318 and following, use appropriate nomenclature and replace "top" and "bottom" of cortex with "basal" and "apical".

As requested by the reviewer, we have changed the terminologies “top” and “bottom”, with “basal” and “apical” in new lines 319-332 as follow:

“Our results revealed a shift of SATB2+ neurons away from the apical cerebral cortex plate ..., ... The ten bins were marked sequentially from apical to basal...”.

13) In figures 7 and EV3, the axis of cell type quantification should be label more specifically. For example, what does "Percent of Sox2 cells" mean? Percent of Sox2 positive cells out of total cells in the entire cortical wall, or specific zones?

We have clarified on these points in the figure labelling. In Figure 7, this refers to the percentage of positive cells in areas of the ventricular zone (VZ), whereas in Figure EV3, this refers to percentage of cells in the cortical wall shown.

Referee #2:

In this revised manuscript the authors addressed most of my concerns. However, although the in vitro data is very convincing, I still have concerns about the in vivo data:

We thank the reviewer for taking the time to review our revised manuscript and for finding that we have addressed most of the concerns. Below, we address the remaining concerns.

1) The authors added new images showing immunostaining for layer specific antibodies and immunostaining for Sox2 in Phf6 floxed mice on Fig EV3. While in the text they mention quantifications, these quantified data are not presented. Specifically, the authors wrote Lines 316-319 : « Quantification of these data revealed that although, Phf6 deletion in the CNS did not significantly affect the number of SATB2+, CTIP2+ and TBR2+ neurons, a shift of Hoechst+ neurons/SATB2+ neurons towards the top of cerebral cortex plate was observed in Phf6-/Y / Nestin-Cre+ mice.

We would like to clarify that the text is referring to the quantification of data for different layer neurons (Figure EV3E-F). For clarity, we have edited the text as follows in the revised manuscript on lines 315-332:

Point-by-point response

“We thus set out to analyze the impact of *Phf6* deletion on cortical layer neurons via subjecting *Phf6^{loxP/Y} / Nestin-Cre⁻* and *Phf6^{-Y} / Nestin-Cre⁺* brain sections at P0 to immunohistochemical analysis using antibodies to SATB2⁺, CTIP2⁺, and TBR1⁺ to quantify neuronal numbers in cortical layers II-VI, layer V, and layer VI, respectively (Fig EV3E-F). Our results revealed a shift of SATB2⁺ neurons away from the apical cerebral cortex plate in *Phf6^{-Y} / Nestin-Cre⁺* mice, with no significant changes in the number of SATB2⁺, CTIP2⁺, and TBR1⁺ neurons. To quantify the migration patterns of SATB2⁺ neurons in the cerebral cortex influenced by loss of *Phf6*, a grid consisting of 10 equivalent bins was applied to the image of P0 cerebral cortex to equally divide the cortical wall spanning from the basal of ventricle zone to the pial surface into ten bins...”

In addition, we include quantification of SOX2 in BFLS (Fig 7C, F), and as was requested we also provided supplemental SOX2 images from *Phf6* KO mice, suggesting the same trend (Fig EV3D).

2) The description of this data is odd, the authors refer to Hoechst+ neurons while Hoechst is a universal nuclear marker.

We thank the reviewer for pointing out the typo. We edited the text as follows in the revised manuscript (new lines 315-332):

“We thus set out to analyze the impact of *Phf6* deletion on cortical layer neurons via subjecting *Phf6^{loxP/Y} / Nestin-Cre⁻* and *Phf6^{-Y} / Nestin-Cre⁺* brain sections at P0 to immunohistochemical analysis using antibodies to SATB2⁺, CTIP2⁺, and TBR1⁺ to quantify neuronal numbers in cortical layers II-VI, layer V, and layer VI, respectively (Fig EV3E-F). Our results revealed a shift of SATB2⁺ neurons away from the apical cerebral cortex plate in *Phf6^{-Y} / Nestin-Cre⁺* mice, with no significant changes in the number of SATB2⁺, CTIP2⁺, and TBR1⁺ neurons. To quantify the migration patterns of SATB2⁺ neurons in the cerebral cortex influenced by loss of *Phf6*, a grid consisting of 10 equivalent bins was applied to the image of P0 cerebral cortex to equally divide the cortical wall spanning from the basal of ventricle zone to the pial surface into ten bins. The ten bins were marked sequentially from apical to basal, with bin 1 covering the most superficial (i.e., apical) layer, and bin 10 covering the deepest (i.e., basal) layer. Neurons within each bin were counted and a significant decline in SATB2⁺ neurons in bin 1 of the cerebral cortex was observed in *Phf6^{-Y} / Nestin-Cre⁺* mice (Fig EV3E-F), suggesting impairment in the ability of SATB2⁺ neurons to migrate to superficial layers of the developing cerebral cortex in *Phf6* KO mice. This finding is consistent with the attenuation of neuron density (Fig EV3C) and suggests that PHF6 is involved in regulating the process of radial neuronal migration during the establishment of cortical lamination...”

3) Lastly, the authors mention a shift of SATB2+ neurons towards the top of the cerebral cortex whereas line 325 they conclude to a significant decline in SATB2+ neurons in bin 1 (the most superficial).

We thank the reviewer for pointing out this error, this is now corrected in the revised manuscript as follows (new lines 319-320):

“Our results revealed a shift of SATB2⁺ neurons **away from** the apical cerebral cortex...”

Point-by-point response

Also, the title of figure 7 is misleading, Tbr2 is not a marker of newborn neurons, it is a marker of intermediate progenitors.

Lastly, in light of the data presented, the concluding statement (below) should be modified since no quantified data on a change in neuron numbers is provided:

lines 331-334 : Together, we report that PHF6 alters the mechanisms that regulate cell fate in the developing brain, and that loss-of-function of PHF6 in BFLS results in an imbalance in the number of uncommitted stem cells and newly born neurons which potentially may describe impaired circuit connectivity in BFLS patients.

We agree with the reviewer that Tbr2 is an intermediate progenitor marker, however, Tbr2+ intermediate progenitors arise from radial glial cells and will then differentiate into Tbr1+ mature neurons of the cortical plate (Elsen *et al*, 2021; Englund *et al*, 2005), Thus Tbr2 is also considered a neural progenitor marker (Khacho *et al*, 2016; Qiu *et al*, 2010; Wang *et al*, 2011), which is why we had previously referred to Tbr2 as an early neuron marker. We have updated the concluding statement changing it to progenitors (new lines 333-335):

“Together, we report that PHF6 alters the mechanisms that regulate NSC fate in the developing brain, and that loss-of-function of PHF6 in BFLS results in an imbalance in the number of uncommitted stem cells and neural progenitors which may contribute to BFLS pathogenesis.”.

Referee #3:

The current version of manuscript is much more accomplished than the original one. The authors solve the molecular mechanisms of how Phf6 mutants cause neurogenic defects in BFLS. The discovery of Phf6-EphA regulatory pathway connects a rare disease to a conventional neurogenic molecule, which largely accelerates the development of BFLS treatment. The illustration of evidence is clear and convincing. The article will be more readable if the authors could pay attention to the interpretation and demonstration of raw data. In summary, the manuscript has met the basic quality for publication and I have only minor points on the manuscript.

We thank the reviewer for the many positive comments and stating that we solve the molecular mechanisms of how Phf6 mutants cause neurogenic defects in BFLS and that the discovery of Phf6-EphA regulatory pathway connects a rare disease to a conventional neurogenic molecule, which largely accelerates the development of BFLS treatment. Below we address the remaining minor points.

Line 123-126: The authors discussed two downregulated GO terms related to nervous system development as Fig 2B shows. However, Fig 2B also exhibits the upregulation of cation channel activities, which highly relates to the maturation of neurons. Could the authors describe these findings and raise a possible explanation? The interpretation is recommended to put in the section between Line 169-179, where the authors find the importance of Phf6 in neuronal differentiation.

Point-by-point response

We have now highlighted the upregulation of cation channel activity in the discussion section (new lines 391-396) of the revised manuscript:

“Cation channels are vital for action potential generation and propagation, synaptic transmission, and overall neuronal communication and functioning (Chen & Lui, 2019). The upregulation of cation channel activities might represent a compensatory mechanism to enhance neuronal function or to accelerate certain aspects of neuronal maturation given the developmental delays observed in BFLS.”.

Line 144: The context of this sentence is emphasizing the role of PHF6 in cell proliferation or self-renewal rather than cell fate decision. Could the authors correct the small mistake?

As requested by the reviewer, we have revised the sentence and have replaced cell fate with cell proliferation or self-renewal in the paragraph as follow in lines 143-149:

“These findings led us to investigate whether PHF6 regulates cell proliferation or self-renewal. To begin with, we subjected...”

Line 198-201: The authors are supposed to describe the details of the RNA-seq analysis directly in the main text, e.g., the sample groups and detailed strategies of screening.

The details of the RNA-seq analysis are included directly in the main text in line 120-127, when it is first mentioned with the results of Figure 2:

“Next, we profiled the genome-wide pattern of gene deregulation by analysis of *Phf6* knockdown (KD) and control cortical progenitors following their isolation at embryonic day 14 (E14) and expansion for 5 days in culture...”.

Line 274: Before the section of the rescue function of EphA4/7 in R342X mice, several rows of brief introduction about previous research on regulatory functions of EphA4/7 to neural stem cells in neural development, like what the authors mention in the Discussion section, are recommended. It helps emphasize the importance of EphA4/7 and inspire readers to understand the whole regulatory mechanism.

As requested by the reviewer, we have now added a brief introduction to emphasize the importance of EphA4/A7 in lines 262-264 of the revised manuscript, making it more comprehensive:

“EphA4 and EphA7 are involved in NSC regulation and neural development. EphA4 has been studied in axon guidance and neural circuit formation, whereas EphA7 is shown to play a key role in apoptosis and cortical patterning (Depaepe *et al*, 2005; Kania & Klein, 2016; Klein, 2012)...”.

Line 305-312: The proportional alteration of Sox+ cells and Tbr+ cells in R342X model is well-described. However, any further interpretations are lacking so that it is hard for readers to understand the biological meaning of the changed cell proportion. A conclusion or demonstration of the phenomenon is required at the end of this part.

Point-by-point response

As requested by the reviewer, we have now added further interpretation of the alterations observed in SOX2+ and TBR2+ cells in the R342X model as follows (new lines 311-313):

“The changes in the proportion of SOX2+ and TBR2+ cells suggest altered cell populations manifesting a disproportionate number of neural stem versus progenitor cells in BFLS, which may contribute to disease pathogenesis...”.

Line 369: It is not proper to use "dysregulation" to illustrate the normal biological function of PHF6.

As requested by the reviewer, we have revised the sentence as follows (new lines 369-372):

“...consistent with our findings in stem cell regulation, other groups have also reported a role for PHF6 in cell differentiation (Pawar *et al*, 2021) and lineage specification (Soto-Feliciano *et al*, 2017) in leukemia myeloid cell models.”.

Table EV6: It is not reader-friendly that the authors only provide the Ensembl ID of each gene. Please attach gene symbols in addition to Ensembl ID in each table.

As requested by the reviewer, we have now included a column for gene symbols in the new Dataset EV6.

References:

- Aoki M, Yamashita T, Tohyama M (2004) EphA receptors direct the differentiation of mammalian neural precursor cells through a mitogen-activated protein kinase-dependent pathway. *Journal of Biological Chemistry* 279: 32643-32650
- Barquilla A, Pasquale EB (2015) Eph receptors and ephrins: therapeutic opportunities. *Annual review of pharmacology and toxicology* 55: 465-487
- Chen I, Lui F (2019) Neuroanatomy, neuron action potential.
- Darling TK, Lamb TJ (2019) Emerging roles for Eph receptors and ephrin ligands in immunity. *Frontiers in immunology* 10: 1473
- del Valle K, Theus MH, Bethea JR, Liebl DJ, Ricard J (2011) Neural progenitors proliferation is inhibited by EphB3 in the developing subventricular zone. *International journal of developmental neuroscience* 29: 9-14
- Depaepe V, Suarez-Gonzalez N, Dufour A, Passante L, Gorski JA, Jones KR, Ledent C, Vanderhaeghen P (2005) Ephrin signalling controls brain size by regulating apoptosis of neural progenitors. *Nature* 435: 1244-1250
- Gerdes J, Schwab U, Lemke H, Stein H (1983) Production of a mouse monoclonal antibody reactive with a human nuclear antigen associated with cell proliferation. *International journal of cancer* 31: 13-20
- Kania A, Klein R (2016) Mechanisms of ephrin–Eph signalling in development, physiology and disease. *Nature reviews Molecular cell biology* 17: 240-256
- Khacho M, Clark A, Svoboda DS, Azzi J, MacLaurin JG, Meghaizel C, Sesaki H, Lagace DC, Germain M, Harper M-E (2016) Mitochondrial dynamics impacts stem cell identity and fate decisions by regulating a nuclear transcriptional program. *Cell stem cell* 19: 232-247
- Klein R (2012) Eph/ephrin signalling during development. *Development* 139: 4105-4109
- Lisabeth EM, Falivelli G, Pasquale EB (2013) Eph receptor signaling and ephrins. *Cold Spring Harbor perspectives in biology* 5: a009159
- Liu Z, Li F, Ruan K, Zhang J, Mei Y, Wu J, Shi Y (2014) Structural and functional insights into the human Börjeson-Forssman-Lehmann syndrome-associated protein PHF6. *Journal of Biological Chemistry* 289: 10069-10083
- Mootha VK, Lindgren CM, Eriksson K-F, Subramanian A, Sihag S, Lehar J, Puigserver P, Carlsson E, Ridderstråle M, Laurila E (2003) PGC-1 α -responsive genes involved in oxidative phosphorylation are coordinately downregulated in human diabetes. *Nature genetics* 34: 267-273
- Pawar A, Somers P, Verner R, Antony C, George SS, Pimkin M, Paralkar VR (2021) PHF6 Positively Regulates Transcription of Myeloid Differentiation Genes By Binding at Enhancer Regions. *Blood* 138: 3303
- Qiu R, Wang J, Tsark W, Lu Q (2010) Essential role of PDZ-RGS3 in the maintenance of neural progenitor cells. *Stem cells* 28: 1602-1610
- Scholzen T, Gerdes J (2000) The Ki-67 protein: from the known and the unknown. *Journal of cellular physiology* 182: 311-322
- Soto-Feliciano YM, Bartlebaugh JME, Liu Y, Sanchez-Rivera FJ, Bhutkar A, Weintraub AS, Buenrostro JD, Cheng CS, Regev A, Jacks TE *et al* (2017) PHF6 regulates phenotypic plasticity through chromatin organization within lineage-specific genes. *Genes Dev* 31: 973-989
- Stuckmann I, Weigmann A, Shevchenko A, Mann M, Huttner WB (2001) Ephrin B1 is expressed on neuroepithelial cells in correlation with neocortical neurogenesis. *Journal of Neuroscience* 21: 2726-2737

Point-by-point response

Subramanian A, Tamayo P, Mootha VK, Mukherjee S, Ebert BL, Gillette MA, Paulovich A, Pomeroy SL, Golub TR, Lander ES (2005) Gene set enrichment analysis: a knowledge-based approach for interpreting genome-wide expression profiles. *Proceedings of the National Academy of Sciences* 102: 15545-15550

Todd MA, Huh MS, Picketts DJ (2016) The sub-nucleolar localization of PHF6 defines its role in rDNA transcription and early processing events. *Eur J Hum Genet* 24: 1453-1459

Wang J, Zhang H, Young AG, Qiu R, Argalian S, Li X, Wu X, Lemke G, Lu Q (2011) Transcriptome analysis of neural progenitor cells by a genetic dual reporter strategy. *Stem cells* 29: 1589-1600

Wilkinson DG (2014) Regulation of cell differentiation by Eph receptor and ephrin signaling. *Cell adhesion & migration* 8: 339-348

Dr. Arezu Jahani-Asl
University of Ottawa
Cellular and Molecular Medicine
451 Smyth Road
Ottawa, Ontario K1H 8M5
Canada

Dear Dr. Jahani-Asl,

I am very pleased to accept your manuscript for publication in the next available issue of EMBO reports. Thank you for your contribution to our journal.

Yours sincerely,
